# Fine mapping spatiotemporal mechanisms of genetic variants underlying cardiac traits and disease

Matteo D'Antonio ©[1,2,3,9] ✉, Jennifer P. Nguyen[2,4,9], Timothy D. Arthur[2,5], iPSCORE Consortium*, Hiroko Matsui[3], Agnieszka D'Antonio-Chronowska[1] & Kelly A. Frazer ©[1,3] ✉

The causal variants and genes underlying thousands of cardiac GWAS signals have yet to be identified. Here, we leverage spatiotemporal information on 966 RNA-seq cardiac samples and perform an expression quantitative trait locus (eQTL) analysis detecting eQTLs considering both eGenes and eIsoforms. We identify 2,578 eQTLs associated with a specific developmental stage-, tissue- and/or cell type. Colocalization between eQTL and GWAS signals of five cardiac traits identified variants with high posterior probabilities for being causal in 210 GWAS loci. Pulse pressure GWAS loci are enriched for colocalization with fetal- and smooth muscle- eQTLs; pulse rate with adult- and cardiac muscle- eQTLs; and atrial fibrillation with cardiac muscle- eQTLs. Fine mapping identifies 79 credible sets with five or fewer SNPs, of which 15 were associated with spatiotemporal eQTLs. Our study shows that many cardiac GWAS variants impact traits and disease in a developmental stage-, tissue- and/or cell type-specific fashion.

Genome-wide association studies (GWAS) have identified thousands of loci associated with cardiac traits and diseases[1], but for the vast majority of these associations, the underlying causal variants and genes have not yet been delineated. Many approaches have been developed to identify causal variants and genes, including annotating GWAS loci with their closest gene[1], prioritizing variants that overlap regulatory elements active in cardiac tissues[2,3], and integrating GWAS variants with adult cardiac eQTL (expression quantitative trait loci) signals[4]. However, these approaches have been limited in part due to the fact that regulatory elements can regulate the expression of multiple genes over hundreds of kilobases[5,6]; and regulatory variants frequently work in a spatiotemporal context not captured in previous eQTL analyses[7–11]. For some GWAS traits and diseases, efforts have been made to fine map each GWAS locus and identify potential associations between genetic variation, gene expression, and disease[12–14];

however, the cardiac field still lacks a resource for conducting in depth annotations of GWAS loci. Here, we integrate GWAS signals for multiple cardiac traits with gene expression data for different cardiac developmental stages, tissues, and cell types across hundreds of individuals and demonstrate the power of this comprehensive resource for fine mapping causal regulatory variants and understanding the molecular mechanisms underlying cardiovascular GWAS traits and disease.

Gene expression has long been known to be regulated in a spatial (organ, tissue, or cell type) and temporal (fetal-like and adult) specific manner[15–22], indicating that cardiac regulatory variants function, and hence affect, the expression of a gene and its associated cardiac traits and disease, across a range of developmental stages and in different cellular contexts[15–17,23–26]. With the development of cell type deconvolution techniques[27,28], bulk RNA-seq enables the characterization of cell

[1]Department of Pediatrics, University of California San Diego, La Jolla, CA 92093, USA. [2]Division of Biomedical Informatics, University of California, San Diego, La Jolla, CA 92093, USA. [3]Institute of Genomic Medicine, University of California San Diego, 9500 Gilman Dr, La Jolla, CA 92093, USA. [4]Bioinformatics and Systems Biology Graduate Program, University of California, San Diego, La Jolla, CA 92093, USA. [5]Biomedical Sciences Graduate Program, University of California, San Diego, La Jolla, CA 92093, USA. [9]These authors contributed equally: Matteo D'Antonio, Jennifer P. Nguyen. *A list of authors and their affiliations appears at the end of the paper. ✉e-mail: madantonio@ucsd.edu; kafrazer@health.ucsd.edu

type-specific gene expression as well as the expression of both genes and associated isoforms[24,29,30]. Furthermore, the GTEx consortium has generated bulk RNA-seq from hundreds of adult cardiac samples from multiple tissue types and whole genome-sequenced the donors[25]. Although several fetal-associated factors, such as low birth weight, maternal preeclampsia, under- and malnutrition and oxidative stress in utero, have been associated with increased risk of developing cardio-vascular disease as adults[31,32], large numbers of human fetal cardiac specimens are not readily available, and functional genetic variation in fetal heart cell types remains largely unstudied. To overcome ethical and availability issues associated with the use of fetal samples, iPSCORE has developed strategies to employ induced pluripotent stem cell (iPSC) derived cardiovascular precursor cells (iPSC-CVPCs) from hundreds of whole-genome sequenced individuals to study early developmental cardiac traits and disease[29,33–35]. We and others have shown that iPSC-CVPCs display epigenomic and transcriptomic properties similar to that of fetal cardiac cells[29,36–41], and their differentiation results in both cardiomyocytes and epicardium-derived cells[33]. We have also previously deconvoluted the proportions of cell types in ~1000 bulk RNA-seq samples from fetal-like iPSC-CVPCs[33], adult healthy heart and arteria[42], and heart failure[43], and demonstrated the power of this approach for examining the reactivation of fetal-specific genes and isoforms during heart failure[29].

In this study, we show that the combined deconvoluted fetal-like iPSCORE and adult GTEx[29] cardiac expression datasets enables the genome-wide mapping of regulatory variant functions in a spatial-temporal-specific manner. We performed an expression quantitative trait locus (eQTL) analysis on 966 deconvoluted bulk RNA-seq samples, including adult (atrium, ventricle, aorta, and coronary artery) and fetal-like (iPSC-CVPCs) tissues, and identified cardiac eQTL signals associated with 11,692 eGenes and 7165 eIsoforms. Less than half of the eIsoforms shared the same eQTL signal with their associated gene, indicating that molecular mechanisms underlying the associations of genes and their isoforms with cardiac traits and disease are different in many cases. By leveraging information about the source and cellular composition of each sample, we were able to detect more than 2578 eQTLs that function in a spatiotemporal manner. We exploited the cardiac eQTLs to investigate the molecular underpinnings of five cardiac traits and diseases from the UK BioBank and found that many of the cardiac GWAS signals colocalized with eQTL signals that function in a specific cardiac stage, organ, tissue, and/or cell type context. We also observed that three of the cardiac traits were enriched for eQTLs that function in specific spatiotemporal contexts, including pulse pressure with fetal-like, arteria, and smooth muscle cell eQTLs; pulse rate with adult, heart, and atrial eQTLs; and atrial fibrillation with left ventricle and cardiac muscle cell eQTLs. We used the colocalized eQTL and GWAS signals for fine mapping, which allowed us to identify a potential causal variant for 210 cardiac GWAS loci. Overall, our study serves as a comprehensive resource mapping cardiac regulatory variants that function in spatiotemporal context-specific manners to alter gene expression and affect cardiac traits, thereby providing potential molecular mechanisms underlying the associations of hundreds of GWAS loci with cardiac traits and disease.

## Results
### Combined eQTL analysis
To investigate associations between genetic variation and cardiac gene expression we obtained RNA-seq for 180 fetal-like iPSC-CVPCs (derived from 149 iPSC lines) from 139 individuals included in the iPSCORE collection[33] and integrated with RNA-seq data for 786 adult cardiac tissues, including atrial appendage, left ventricle, aorta, and coronary artery from 352 individuals included in the GTEx Consortium[42] (Supplementary Data 1). To map the regulatory effects of genetic variants on these fetal-like and adult cardiac tissues, we performed a combined eQTL analysis on all 966 RNA-seq samples using a linear mixed model

(LMM) including a kinship matrix to account for genetic relatedness between samples. Among 19,586 expressed autosomal genes (TPM ≥ 1 in at least 10% samples), we identified at least one eQTL for 11,692 genes (eGenes, 59.7% of tested genes, Fig. 1a, Supplementary Data 2, Summary Statistics reported in "Figshare [https://doi.org/10.6084/m9.figshare.c.5594121]"[44]). An eGene may have multiple different eQTL signals, each associated with a different underlying causal variant, which can be detected by regressing out the genotype of the lead variant of the primary and subsequent conditional eQTLs. We obtained conditional eQTLs for 4394 eGenes (37.6% of all eGenes), including 1315 with two conditional eQTLs, 395 with three, 160 with four, and 74 with five (1.54 independent eQTL signals per gene, 18,030 eQTL signals overall, Fig. 1a), which is in line with what has recently been reported by GTEx[42]. We also examined 37,032 autosomal isoforms (corresponding to 10,337 genes; at least two isoforms/gene with usage ≥10% in at least 10% samples) and identified 7165 with at least one eQTL (eIsoforms, 19.3% of all tested isoforms, corresponding to 3847 genes), including 988 with one or more conditional eQTL (8489 eQTL signals overall Fig. 1a). We identified fewer eIsoforms than eGenes likely because of decreased power in detecting eQTL associations, caused by a more stringent multiple testing correction, as we tested twice as many isoforms than genes.

### Mechanisms underlying eQTLs for genes and isoforms
We investigated the extent to which underlying genetic mechanisms differ between eQTL signals for gene expression and isoform usage. Of the 3847 genes associated with the 7165 eIsoforms, 2909 were themselves eGenes (corresponding to 5744 eIsoforms), while 938 were not eGenes (corresponding to 1421 eIsoforms). For the 2909 eGenes with eIsoforms, to examine if the same genetic variants were associated with both gene expression and isoform use, we performed a colocalization analysis[45] (a Bayesian approach that provides posterior probabilities of signals for two traits at one locus (trait 1 = eIsoform; trait 2 = eGene) for five hypotheses: H0) neither trait is associated; H1) only trait 1 is associated; H2) only trait 2 is associated; H3) both traits are associated, but with different underlying causal variants; and H4) both traits are associated with the same underlying causal variants). For 3221 eIsoforms (56.2% of the 5744 associated with eGenes) the most likely hypothesis was 4, indicating that the eIsoform and eGene eQTL signals share the same causal variant (Fig. 1b, Supplementary Data 3). For example, *B4GALT7* and its isoform (ENST00000029410.10_2) shared a common eQTL signal, with their lead variant (rs28473516) having posterior probability of association (PPA) > 99% of being causal for both gene expression and isoform use (Fig. 1c). On the other hand, 1674 (29.1%) of the eIsoforms had a different signal than their associated eGene. For example, the lead variant for *RNH1* (rs61876335) was located ~6 kb downstream of the 3′ end of the gene, whereas the lead variant for its isoform (rs17584, ENST00000397604.7_2), which is not in LD with rs61876335 ($r^2 = 0.019$ in EUR individuals), was a synonymous coding variant (Fig. 1d). For the remaining 849 (14.7%) eIsoforms, the eQTL signals for the eIsoform (hypothesis 1), for the eGene (hypothesis 2) or for both (hypothesis 0) were likely underpowered. These results show that at least 29.1% of eIsoforms had a different signal than their associated eGene, suggesting that different mechanisms underlie the association of genetic variation with gene expression and isoform use.

To test if eQTLs for genes and isoforms have different properties, we investigated their overlap with gene bodies and promoters. We observed that gene eQTLs are more likely to occur in intergenic regions than isoform eQTLs ($p \approx 0$), whereas isoform eQTLs are more likely to overlap gene bodies, including introns ($p = 1.9 \times 10^{-87}$), UTRs ($p = 7.8 \times 10^{-88}$), splice donor sites ($p = 9.0 \times 10^{-19}$), splice acceptor sites ($p = 1.1 \times 10^{-31}$) and exons ($p = 1.1 \times 10^{-47}$, Fig. 1e), indicating that isoform eQTLs are more likely to influence transcript stability than regulatory elements. For example, we found that the lead eVariant (rs11589479)

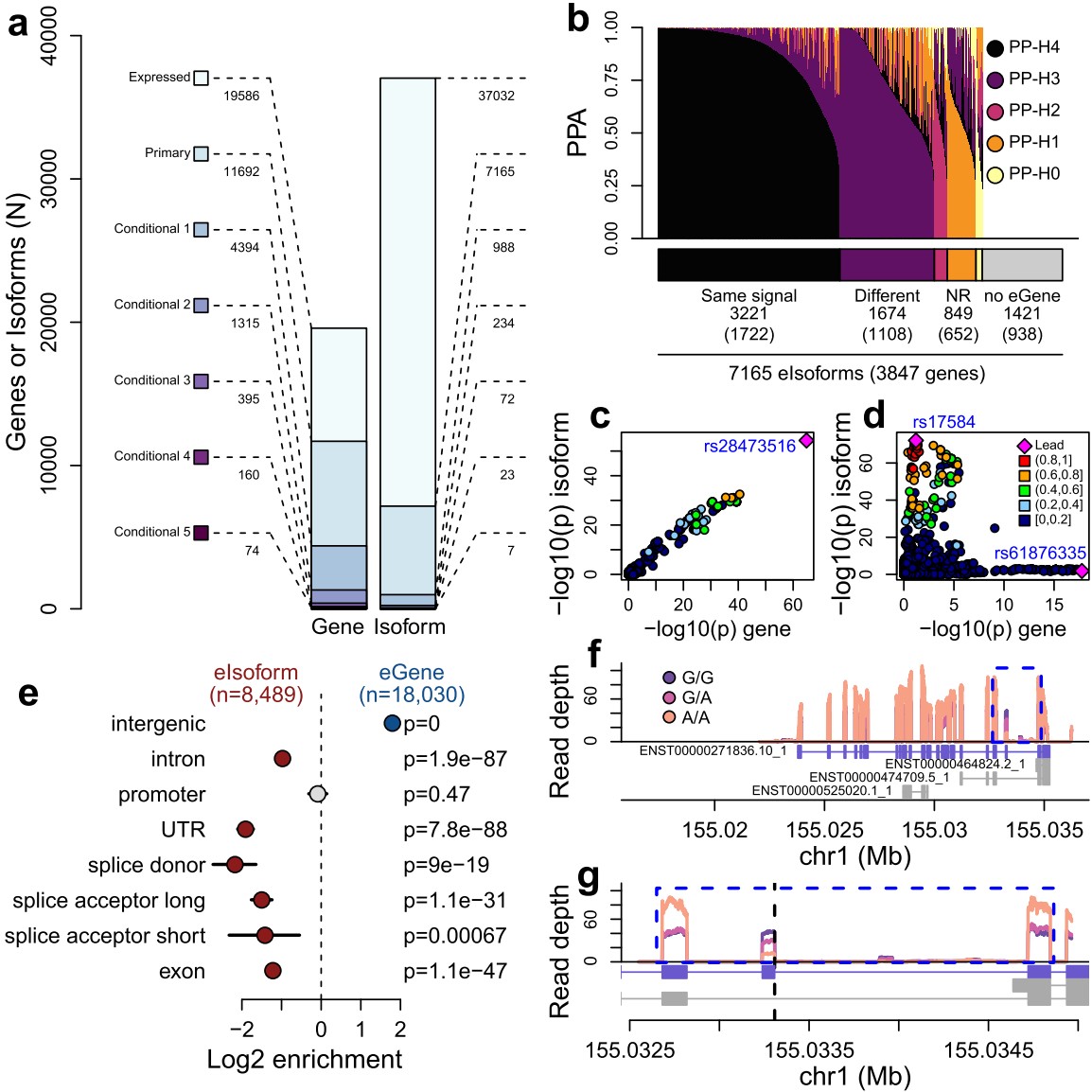

**Fig. 1 | Gene and isoform eQTLs. a** Barplot showing the number of eQTLs for eGenes (left) and eIsoforms (right). Colors represent the number of eGenes with primary and conditional eQTLs (up to five conditional signals). **b** Barplot showing the distribution of PPA for each of the five colocalization hypotheses. All the colocalizations associated with hypothesis 0, 1, and 2 were likely underpowered, thus they were labeled as "not resolved (NR)". The 1421 eIsoforms whose associated gene did not have an eQTL were labeled as "no eGene". **c, d** Examples of (**c**) eQTL signal for an eGene (*B4GALT7*) that colocalizes with PPA = 1 with the eQTL signal of one associated eIsoform and (**d**) eQTL signal for an eGene (*RNH1*) that does not colocalize with the eQTL signal of one associated eIsoform. In each plot, X axis represents the −log₁₀ (eQTL p-value) for the associations between the genotype of each tested variant and gene expression, whereas the Y axis shows the −log₁₀ (eQTL p-value) for the associations between the genotype of each tested variant and isoform use. **e** Enrichment of eGenes compared with eIsoforms for overlapping intergenic regions, introns, promoters, UTRs, splice donor sites, splice acceptor sites (short = the first 5 nucleotides upstream of the splice site; long = the first 100 bp) and exons. *P*-values were calculated using Fisher's exact test. Points (blue = enriched for eGenes; red = enriched for eIsoforms; gray = not significant) represent log₂ enrichment and horizontal lines represent 95% confidence intervals calculated using the *fisher.test* function in R. **f, g** Median normalized read depth signal of *ADAM15* gene expression levels in iPSC-CVPCs. Different colors represent the genotypes of the lead eVariant for isoform ENST00000271836.10_1 (rs11589479, G > A). The blue rectangle in (**f**) is enlarged in (**g**). rs11589479 overlaps the splice donor site for exon 19 and its position is shown as a vertical dashed line in (**g**). The plots show that the exon whose splice site is affected by rs11589479 becomes expressed at lower levels when the variant is heterozygous or homozygous alternative, as it disrupts the splice site.

for an *ADAM15* isoform (ENST00000271836.10_1: the most common isoform of *ADAM15*) is a G > A substitution that likely disrupts the splice site for exon 19 (Fig. 1f, g). *ADAM15* encodes a disintegrin and metalloprotease involved in cell-cell and cell-matrix interactions and has an established role in inflammation and angiogenesis[46]. Of note, rs11589479 did not colocalize with the primary eQTL signal for *ADAM15* expression. We observed that *ADAM15* exon 19 was expressed at lower levels in samples carrying heterozygous or homozygous alternative alleles for rs11589479 and that the overall expression of exon 19 was

reduced by ~80% in homozygous alternative samples compared with the surrounding exons, whereas in samples carrying homozygous reference alleles for rs11589479 the expression of exon 19 was comparable with the surrounding exons. These data show that the molecular mechanisms underlying the associations of genes and their isoforms with cardiac traits and disease will be different in many cases, with gene expression being likely associated with regulatory variants at promoters and enhancers and isoform usage associated with variants that affect post-transcriptional modifications.

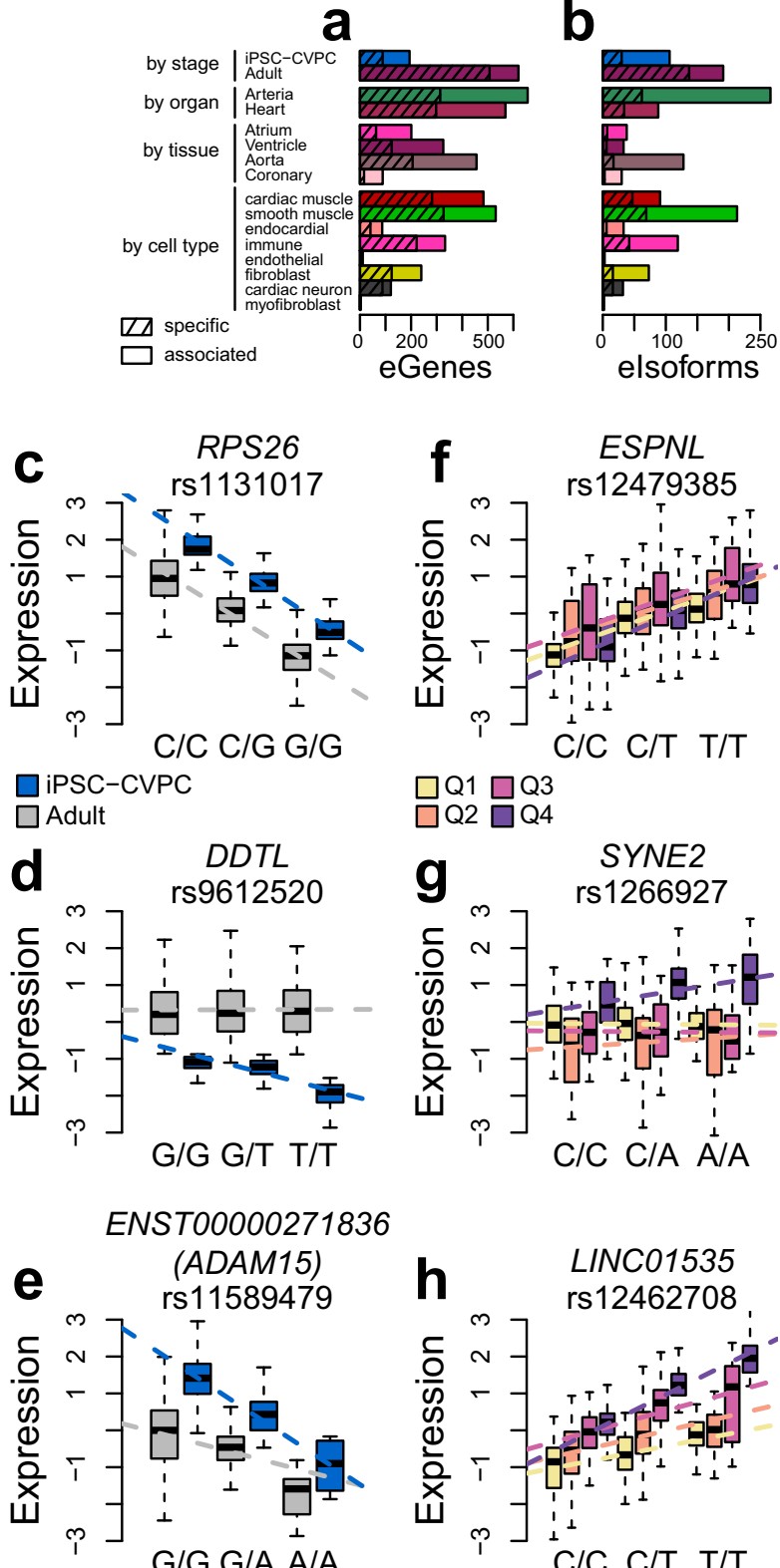

## Mapping spatiotemporal cardiovascular eQTLs

To identify eQTLs for eGenes and eIsoforms that function in a spatiotemporal manner, we used an interaction term to determine if the lead eVariants identified in the combined analysis were associated with samples at one stage but not the other (iPSC-CVPC or adult) or samples annotated as a specific organ (arteria and heart) or tissue (atrium, ventricle, aorta, and coronary artery) or if they were associated with specific cell types (cardiac muscle, smooth muscle, endothelial, fibroblast, immune, cardiac neuron, endocardial, and myofibroblast)[47]. From this analysis, we classified eQTL signals as shared, specific, or associated (Fig. 2a–e, Supplementary Figs. 1, 2, Supplementary Data 4). For example, for *RPS26* we did not observe a significant interaction between the genotype of the lead eQTL signal variant (rs1131017) and stage ($\beta = 0.002$, $p = 0.21$, Fig. 2c), therefore we annotated this eQTL as

**Fig. 2 | Stage, organ, tissue, and cell type eQTLs. a**, **b** Barplots showing the number of **a** eGenes and **b** eIsoforms associated with: cardiac stage (iPSC-CVPC or adult); organ (arteria or heart); tissue (atrial appendage, left ventricle, aorta or coronary artery); and cell type. Non-hatched bar sections represent eQTLs that are associated with the indicated stage, organ, tissue or cell type, and hatched sections represent specific eQTLs. **c–e** Examples of three association types between eQTLs and cardiac stage. For each eGene, boxplots ($n = 966$ samples) describe the normalized expression in iPSC-CVPCs (blue) and all other samples (i.e., adult cardiac samples; gray), grouped by genotype. The panels show examples of: **c** an eGene whose eQTL is shared across both cardiac stages; **d** an iPSC-CVPC-specific eQTL: the association between genotype and gene expression is only present in iPSC-CVPCs; and **e** iPSC-CVPC-associated eQTL: while the genotype is associated with gene expression in both iPSC-CVPCs and the adult samples, the eQTL is significantly

stronger in iPSC-CVPCs. All five possible association types are shown in Supplementary Fig. 1. The boxplots were built as follows: upper and lower edges represent the 25th and 75th percentiles and the middle line the median, vertical bars represent the distance from the 25th (or 75th) percentile minus (or plus) 1.5 times the interquartile range. **f–h** Examples of associations between eQTLs and cell types. For each eGene, boxplots ($n = 966$ samples) describe the normalized expression divided into four quartiles according to their cardiac muscle proportion (yellow = low; purple = high), grouped by genotype. The panels show examples of: **f** an eGene whose eQTL is shared across cell types; **g** a cardiac muscle-specific eQTL: the association between genotype and gene expression is only present in the top quartiles; and **h** cardiac muscle-associated eQTL: while the genotype is associated with gene expression in all quartiles, the eQTL is significantly stronger in the top quartiles. All five possible association types are shown in Supplementary Fig. 3.

"shared" between both stages. Conversely, the eQTL signals for *DDTL* and for *ADAM15* transcript ENST00000271836.10_1 showed a significant interaction between genotype (rs9612520 and rs11589479, respectively) and stage ($\beta = -0.90$, $p = 2.0 \times 10^{-42}$ for *DDTL*; $\beta = -1.09$, $p = 2.6 \times 10^{-7}$ for *ADAM15*, Fig. 2d, e), suggesting that their expression is differentially associated with genotype between iPSC-CVPC and adult heart. However, these two eQTL signals show substantial differences. The genotype of rs9612520 is not associated with *DDTL* in adult cardiac samples ($\beta = 0$, $p = 0.90$), suggesting that this spatiotemporal eQTL signal is iPSC-CVPC-specific, whereas the spatiotemporal eQTL signal for *ADAM15* is significant in both iPSC-CVPC ($\beta = -2.32$, $p = 2.7 \times 10^{-22}$) and adult heart ($\beta = -0.92$, $p = 8.0 \times 10^{-14}$), when tested independently. Since the signal in iPSC-CVPC is stronger than adult, we labeled this spatiotemporal eQTL as "iPSC-CVPC-associated". We found 814 stage-eQTL signals (combined -specific and -associated) for eGenes and 297 for eIsoforms (Bonferroni-corrected $p < 0.05$). Of these, the majority (620 eGenes and 191 eIsoforms) had adult-specific eQTL signals; 105 eGenes and 76 eIsoforms had associated eQTLs (i.e., in both stages but with significantly different effect sizes); and 89 eGenes and 30 eIsoforms had iPSC-CVPC-specific eQTL signals. We also observed organ- and tissue-specific or -associated eQTL signals for 1,246 eGenes and 350 for eIsoforms (Fig. 2a, b, Supplementary Fig. 2). Since the same spatiotemporal eQTL signal may be associated with both stage and tissue or organ, in total our interaction eQTL approach identified 1665 eQTL signals for eGenes and 565 for eIsoforms associated with cardiac stage, organ and/or tissue.

To assess the associations between eQTLs and cell types, we divided the 966 RNA-seq samples into quartiles according to their deconvoluted cell type populations, and for each cell type, tested the interaction between genotype and cell type proportions and then compared the top and bottom quartiles (Fig. 2f–h, Supplementary Fig. 3). We identified cases, such as *ESPNL*, where the interaction was not significant ($\beta = -0.003$, $p = 1.0$, Fig. 2f), suggesting that these eQTLs were shared across cell types. For eQTL signals with a significant interaction, we compared the top and bottom quartile and annotated the spatiotemporal eQTL as "cell type-specific" if only the expression levels in the samples included in the top quartile were associated with the genotype (for example, *SYNE2* and cardiac muscle, Fig. 2g), and as "cell type-associated" if both the top and bottom quartile were significantly associated, but the signal in the top quartile was stronger (for example, *LINC01535*, Fig. 2h) Using this method, we found 1191 cell type-specific or -associated eQTL signals for eGenes and 364 for eIsoforms (Fig. 2a, b). To validate our approach for identifying cell type eQTLs (combined -specific and -associated), we tested the overlap between cardiac muscle eQTLs with regulatory elements in nine cardiac cell types obtained from a single nuclei ATAC-seq (snATAC-seq) study on adult cardiac cells[48]. We observed that the cardiac muscle- eQTLs were more likely than expected to overlap regulatory elements enriched for being active in atrial and ventricular cardiomyocytes ($p = 4.8 \times 10^{-14}$ and $p = 8.4 \times 10^{-9}$, respectively, Supplementary Fig. 4, Supplementary Data 5) and were less likely to overlap regulatory elements active in

other cell types, including macrophages ($p = 8.1 \times 10^{-16}$), fibroblasts ($p = 6.4 \times 10^{-10}$) and adipocytes ($p = 6.4 \times 10^{-3}$). Furthermore, we observed that cardiac muscle-, heart- and left ventricle-specific eQTLs were strongly correlated with the set of myocyte eQTLs that were described by GTEx[24] (Supplementary Fig. 5).

Overall, these data show that integrating stage, organ, tissue, and cell type information with genotype and gene expression, we were able to determine the spatiotemporal context of 2578 eQTL signals (Supplementary Data 4).

## Multigenic eQTL signals are enriched for being spatiotemporal regulated

Enhancers can regulate the expression of more than one gene[6], therefore we investigated how often eGenes in close proximity share the same eQTL signal. Using colocalization, we found that 2778 eQTL signals were shared between two or more eGenes or eIsoforms from different genes ("multigenic eQTLs", PPA > 0.8, range = 2–9 genes; mean = $2.21 \pm 0.62$, Supplementary Data 6). We next investigated whether these multigenic eQTL signals (i.e., associated with multiple eGenes) are enriched for being spatiotemporal regulated (i.e., associated with a cardiac stage-, organ-, tissue- or cell type). We found a significant positive association between the number of eGenes that share the same eQTL signal and the likelihood of their eQTL to be spatiotemporal regulated: adult (combined -specific and -associated) ($p = 1.6 \times 10^{-17}$, linear regression, Fig. 3a), organ (arteria: $p = 9.9 \times 10^{-8}$; and heart: $p = 5.5 \times 10^{-4}$), tissue (ventricle: $p = 0.023$; and aorta: $p = 4.8 \times 10^{-3}$) and/or cell type (cardiac muscle: $p = 1.1 \times 10^{-3}$; smooth muscle: $p = 9.5 \times 10^{-4}$; immune cells: $p = 3.1 \times 10^{-7}$; and fibroblasts: $p = 3.3 \times 10^{-12}$). The difference in the enrichment between iPSC-CVPC and adult heart could suggest that fetal eQTLs are less likely to be multigenic than adult eQTLs. Furthermore, we observed that eGenes that share the same eQTL signals are significantly more likely than expected to be associated with the same stage, organ, tissue, or cell type (Fig. 3b). These results indicate that multigenic eVariants are enriched for being in regulatory elements that function in a temporal (in the adult, rather than in the fetal-like heart) and spatial (organ, tissue or cell type) specific manner.

## Cardiac genes and their paired antisense RNA enriched for sharing eQTL signals

Since antisense RNAs are a particular class of non-coding RNAs with regulatory function[49,50], we hypothesized that they may share eQTL signals with their associated gene. In total, we tested 163 pairs of genes and their associated antisense RNAs, and observed that 47 pairs shared the same eQTL signal, which was significantly more than expected (odds ratio = 17.1, $p = 2.8 \times 10^{-37}$, Fisher's exact test; see Supplementary Note 1, Supplementary Figs. 6, 7, 8). For example, rs7589901, located -350 bp downstream of *PAX8-AS1* TSS, was the lead eQTL for both *PAX8* and *PAX8-AS1* (Fig. 3c–e). *PAX8* plays a pivotal role in cardiomyocyte development, growth, and senescence[51], while *PAX8-AS1* knockdown facilitates cell growth[52]. We also observed cases where the signal for an

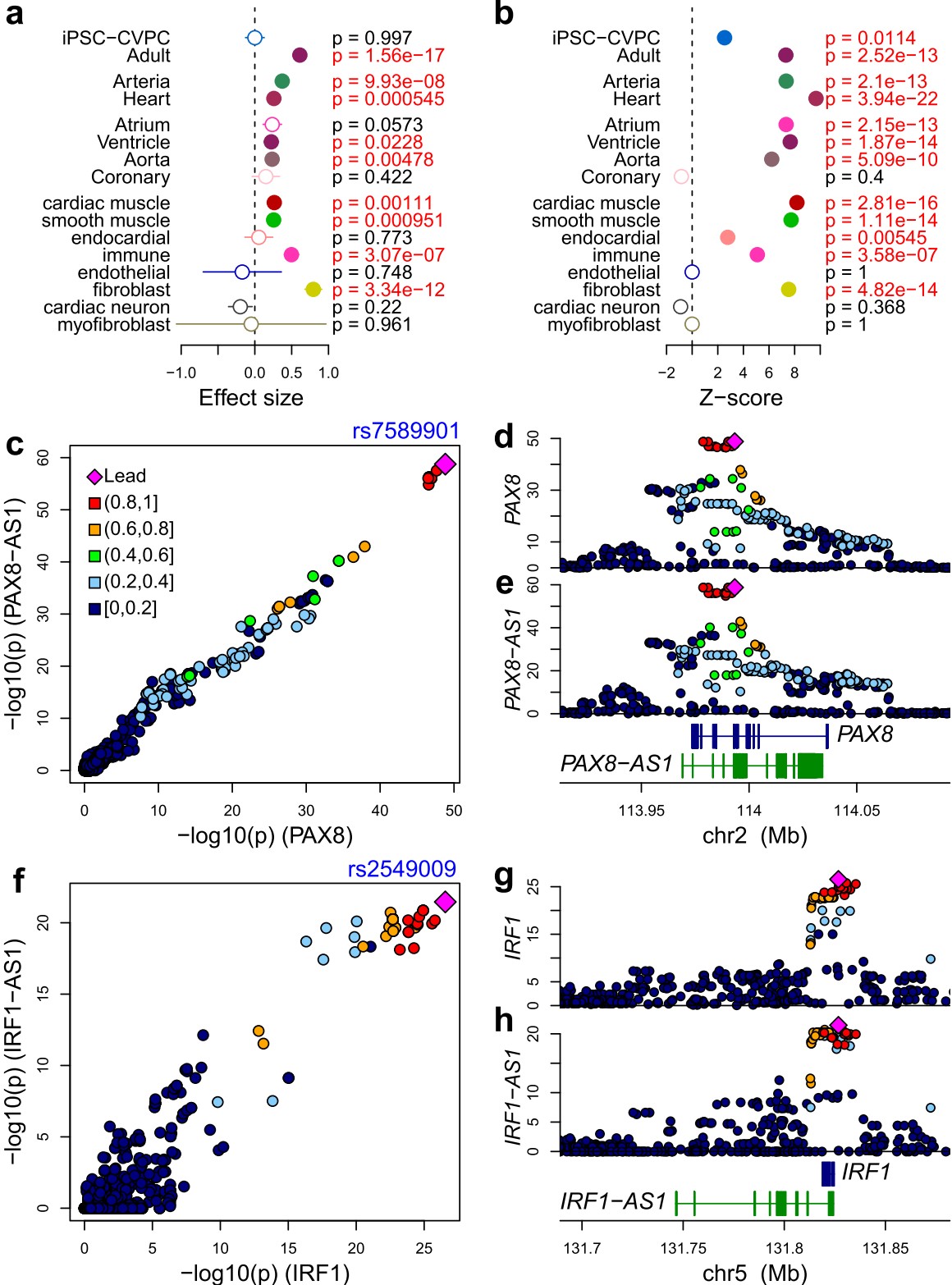

**Fig. 3 | eQTL signals associated with multiple genes. a** Enrichment of eGenes that share the same eQTL signal with other eGenes or eIsoforms for having stage-, organ-, tissue- and cell type- eQTLs measured by linear regression analysis (n = 2778). Dots represent effect size and segments represent standard errors. Effect sizes, standard errors and p-values were measured using the *lm* function in R. **b** eGenes that share the same eQTL signals are enriched for being associated with same stage, organ, tissue, or cell type calculated using a permutation test. Dots represent Z-scores. **c**–**h** eQTL signals shared between a gene and its associated antisense RNA: **c**–**e** *PAX8* and *PAX8-AS1*; **f**–**h** *IRF1* and *IRF1-AS1*. **c**, **f** scatterplots showing the −log₁₀ (eQTL p-value) for the gene (X axis; **c**: *PAX8*; **f**: *IRF1*) and for the antisense RNA (Y axis; **c**: *PAX8-AS1*; **f**: *IRF1-AS1*). **d**, **e**, **g**, **h** eQTL signal for the gene (in blue, panels **d** and **g**) and for the antisense RNA (in green, panels **e** and **h**).

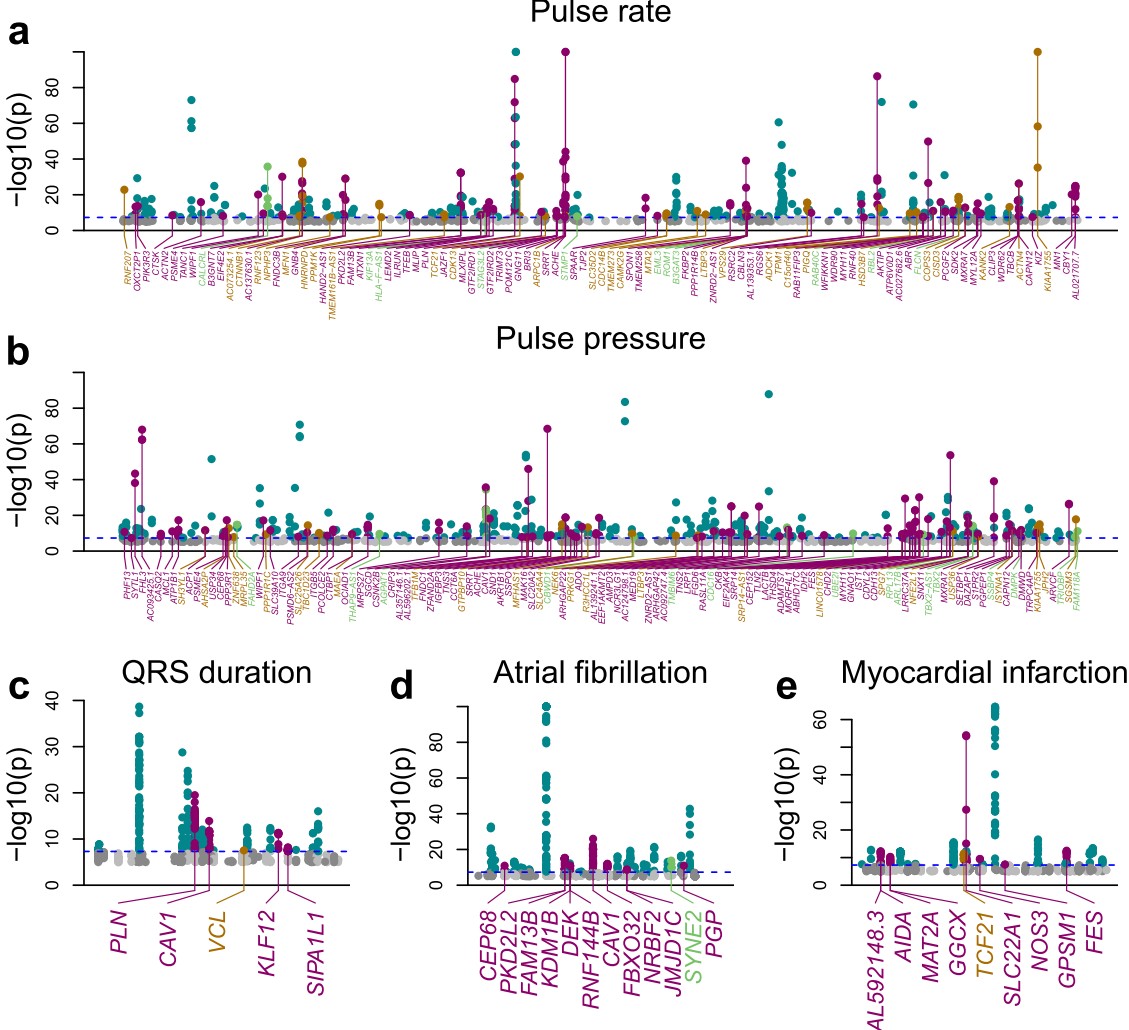

**Fig. 4 | Manhattan plots showing the GWAS signals that colocalize with eQTLs.**
**a–e** Manhattan plots showing the GWAS signals for five cardiac traits. Among all genome-wide significant SNPs, the GWAS signals that colocalize with eQTLs for eGenes are highlighted in purple, with eQTLs for eIsoforms (light brown), with eQTL signals for both eGenes and eIsoforms (green), and those that do not colocalize with eQTLs are shown in turquoise. Horizontal dashed blue line represents the genome-wide significance threshold (GWAS $p = 5 \times 10^{-8}$).

eIsoform colocalized with the expression of its associated antisense RNA, such as rs2549009, ENST00000472045.1_2 (*IRF1*), and *IRF1-AS1* (Fig. 3f–h). *IRF1* is involved in cardiac remodeling and its overexpression is associated with cardiac hypertrophy[53] and rs2549009 is in LD ($R^2 = 0.764$) with rs7734334, a diastolic blood pressure-associated SNP[54], suggesting a likely role of the gene pair *IRF1/IRF1-AS1* in driving cardiac remodeling in response to high blood pressure. These results show that cardiac genes and their paired antisense RNAs are enriched for sharing eQTL signals.

**Colocalization identifies potential molecular mechanisms underlying GWAS signals**
To examine the extent to which the causal variants underlying cardiac eQTLs are associated with cardiac traits and disease, we performed a colocalization test[45] between eQTL signals for eGenes and eIsoforms and the GWAS signals for pulse rate, QRS duration, pulse pressure, atrial fibrillation, and myocardial infarction, all obtained from the UK BioBank. We focused on 1444 eGenes and 919 eIsoforms that overlapped or were in close proximity (<500 kb) with genome-wide significant GWAS signals; and to account for the potential presence of multiple independent causal variants at the same locus, we performed colocalization using both primary and conditional eQTL signals. We

found that the eQTLs for 206 eGenes and 125 eIsoforms (including 296 primary and 35 conditional eQTLs) colocalized with high PPA with at least one GWAS signal (PPA ≥ 0.8, Fig. 4, Supplementary Data 7). Since multiple eGenes may share the same eQTL signal and certain eQTL signals may be associated with both gene expression and isoform usage, we identified 210 independent GWAS signals associated with one or more eQTLs, including 145 that colocalized with a single eQTL and 65 with multiple eQTL signals (range: 2–9). The vast majority of eQTL-GWAS signal colocalizations were associated with pulse pressure (106 signals) and pulse rate (83), whereas QRS interval, atrial fibrillation, and myocardial infraction were associated with fewer than 10 signals each (five, nine, and seven, respectively). This finding is consistent with the fact that pulse rate and pulse pressure are known to have hundreds of genome-wide significant loci, whereas less than 100 GWAS loci have been associated with atrial fibrillation- or myocardial infarction[2,55–58].

Since associations between genetic variation and complex traits can function in a spatiotemporal manner[24], we investigated what fraction of the eQTL signals colocalizing with cardiac GWAS signals were associated with developmental stage, organ, tissue, or cell type. Of the 206 eGenes and 125 eIsoforms with a high PPA with at least one GWAS signal (PPA ≥ 0.8, 210 GWAS loci in total), we observed that 51

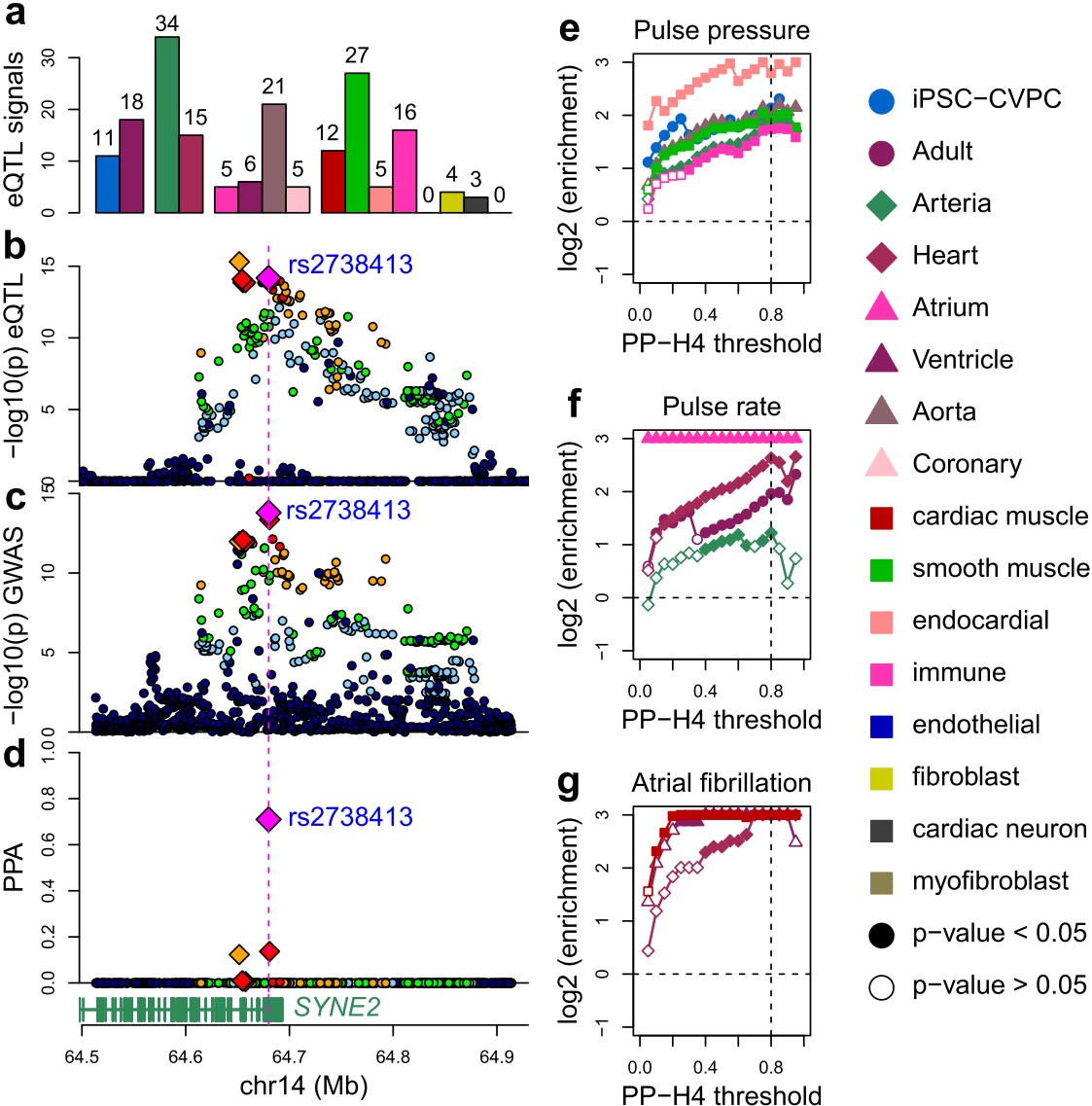

**Fig. 5 | Colocalization between spatiotemporal eQTLs and cardiac GWAS.**
**a** Barplots showing the number of spatiotemporal eQTL signals that colocalize with cardiac GWAS signals. We observed 51 eGenes and 21 eIsoforms that were associated with one or more spatiotemporal categories colocalize with GWAS traits. Colors represent cardiac stage, organ, tissue, and cell type as described in the legend on the right. **b–d** Plots showing (**b**) −log₁₀(*p*-value) for cardiac muscle- eQTL signal for *SYNE2*, **c** the GWAS signal for atrial fibrillation and **d** the PPA of each variant in the colocalization. The lead variant (i.e., the variant with highest PPA of being causal for both the eQTL and GWAS signals) is shown as a magenta diamond: all non-lead variants with PPA > 0.01 are shown as smaller red diamonds. The variant with the strongest *p*-value in the eQTL signal is shown as an orange diamond. **e–g** Line plots showing the enrichment of stage-, organ-, tissue- and cell-type- eQTLs in various GWAS traits: **e** pulse pressure; **f** pulse rate; and **g** atrial fibrillation (ICD10 code: I48). Enrichment is plotted as the log (odds ratio) (Y axis) over all PP-H4 thresholds (0.05 to 0.95, in 0.05 increments) of the eQTL signal colocalizing (0 = not colocalizing; 1 = completely colocalizing) with the GWAS signal (X axis). Only contexts with FDR-corrected *p*-value <0.01 at PP-H4 = 0.8 are shown. All associations for the five traits are shown in Supplementary Figs. 10, 11, and 12.

(24.8%) and 21 (16.8%), respectively, had stage, organ, tissue, or cell type eQTLs (PPA > 0.8, Fig. 5a, Supplementary Data 7), suggesting that these loci are highly context-dependent. For example, the cardiac muscle- eQTL signal for *SYNE2* (Figs. 2g, 5b–d), a gene that encodes for protein included in the nesprin family that links organelles and nuclear lamina to the actin cytoskeleton[59], colocalized with an atrial fibrillation GWAS signal (PPA = 98.3%). Overall, these data show that about one-quarter of the eQTL signals that colocalize with cardiac GWAS traits function in a spatiotemporal manner.

**Cardiac traits enriched for spatiotemporal eQTLs that function in specific contexts**
To characterize the associations between each trait and eVariants that function in specific spatiotemporal contexts, we performed an enrichment analysis[30]. Significant associations were observed for three

of the five traits: pulse pressure, pulse rate, and atrial fibrillation (Supplementary Figs. 9, 10, 14, Supplementary Data 8). Pulse pressure was enriched for fetal-like iPSC-CVPC-, arteria-, aorta-, smooth muscle-, endocardial- and immune- eQTLs (Fig. 5e); pulse rate was associated with adult-, heart-, arteria- (lower extent), and atrium- eQTLs (Fig. 5f); and atrial fibrillation with heart-, left ventricle- and cardiac muscle- eQTLs (Fig. 5g). While most of these spatiotemporal enrichments make sense for the given trait, the enrichment of pulse pressure for fetal-like iPSC-CVPC-eQTLs was surprising (Supplementary Fig. 11). The eQTL signals underlying this enrichment were for five eGenes (*AKR1B1*, *CBWD1*, *RPL13*, *THAP9-AS1*, and *TBX2-AS1*) and three eIsoforms (*DMPK*, *RPL13*, and *PRKG1*), of which two were iPSC-CVPC-specific (*AKR1B1* and *TBX2-AS1*), indicating that the variants at these loci associated with pulse pressure in the adult could exert their function during cardiac development. Conversely, seven adult- eGenes (*ACHE*, *EIF4E2*, *FLCN*,

*RAB40C*, *RBL2*, *SPON1*, and *WFIKKN1*) and three adult- eIsoforms (*AC073254.1*, *B3GAT3*, and *RBL2*), of which six were adult-specific (*ACHE*, *EIF4E2*, *RAB40C*, *RBL2*, *SPON1*, and *AC073254.1*), colocalized with pulse rate, resulting in the enrichment of pulse rate for adult-eQTLs (Supplementary Fig. 12). The association between pulse pressure and immune cell- eQTLs was also an unexpected observation. Multiple immune cell- eGenes that colocalized with pulse pressure are involved in immune response and inflammation, including *ASAP2*, *SH3YL1*, *ARVCF*, and *ATP1B1*[60–63], and changes in their expression may affect the mechanisms that trigger the inflammatory response and its effects on blood pressure. To validate enrichments of the spatiotemporal eQTLs and GWAS signals, we determined the tissue context of each eQTL signal using an independent method, multi-variate adaptive shrinkage (mash)[64], and observed very similar enrichments (see Supplementary Note 2, Supplementary Figs. 13, 14, Supplementary Data 9, 10). In summary, by integrating context-associated eQTLs with GWAS, we were able to find that the SNPs associated with three of the five cardiac traits are enriched for being functional in specific spatiotemporal contexts, including associations between pulse pressure and fetal-like iPSC-CVPC-eQTLs and between pulse rate and adult-eQTLs.

## Fine mapping using colocalization data identifies putative causal variants for hundreds of loci

To identify putative causal variants underlying the shared GWAS and eQTL signals we used the colocalization data to conduct genetic fine mapping[14]. For each of the 210 independent GWAS signals and each of their colocalizing eGenes and eIsoforms (331 GWAS-eQTL combinations) we computed 99% credible sets (defined as the SNPs whose sum of PPAs is >99%). In the 65 cases where multiple eGenes or eIsoforms colocalized with the same GWAS signal, we retained only the credible set with the smallest number of SNPs and, in cases of multiple credible sets with the same number of SNPs, we retained the one having the lead SNP with highest PPA. Across the five cardiac GWAS traits, we found that most credible sets (113, 53.8%) included 10 or fewer SNPs, including 28 (13.3%) having one single causal variant and 51 (24.3%) between two and five (Fig. 6a–e, Supplementary Data 11). Only two credible sets, both associated with QRS duration, included more than 100 SNPs and were considered as not resolved. Fifteen of the 79 credible sets with five or fewer SNPs were associated with spatiotemporal eQTL signals, suggesting that the association between these genetic variants and cardiac traits likely occurs at specific developmental stages, tissues, or cell types. Interestingly, fine mapping resulted in credible sets with five or fewer SNPs for three of the pulse pressure GWAS signals that colocalized with fetal-like iPSC-CVPC-eQTLs (for two eGenes: *AGPAT1* and *CBWD1*; and one eIsoform: *RPL13*, Fig. 5e, Supplementary Fig. 11). To compare the results from fine mapping by colocalization with eQTLs with standard genetic fine mapping, for each of the 210 loci we performed a standard genetic fine mapping using the GWAS signals alone[14,45], and observed that the colocalization approach resulted in smaller credible set sizes ($p = 2.07 \times 10^{-7}$, paired t-test, Supplementary Fig. 15) and stronger posterior probabilities for the lead variants ($p = 1.15 \times 10^{-13}$). These results show that genetic fine mapping GWAS loci by colocalization with eQTL signals reduces the number of candidate causal variants to only a handful in the majority of loci.

To examine the functional effects of the putative causal variants in the 210 credible sets, we performed two enrichment analyses. First, we obtained Roadmap Epigenome Data for fetal heart, adult right atrium, adult left ventricle, and adult right ventricle and observed that variants with high PPA are more likely to reside in active chromatin states, such as enhancer regions in fetal heart ($q = 2.29 \times 10^{-28}$) and transcriptional start sites in adult heart (right atrium: $q = 3.34 \times 10^{-31}$; left ventricle: $q = 2.75 \times 10^{-38}$; and right ventricle: $q = 3.9 \times 10^{-45}$, Supplementary Fig. 16). Second, we tested the association between the strength of the PPA for a putative causal SNP and the predicted impact of the SNP on disease using DeepSea[65,66]. Using linear regression, we observed that

SNPs with higher PPAs are more likely to impact disease compared to SNPs with lower PPAs ($p = 6.08 \times 10^{-14}$, Supplementary Fig. 17). These results show that the variants with high PPA are more likely to have functional impacts, confirming their putative causal role for the associated cardiac traits.

We examined if the causal variants identified by our colocalization analysis correspond to lead index GWAS SNPs that have previously been described. Specifically, we investigated whether they were reported as associated with the same trait in the GWAS Catalog[1]. We intersected the SNPs with the highest PPA in each of the 210 credible sets with 4,772 trait-associated index SNPs in the GWAS catalog[1], and found that 43 (20.5%) have been previously reported as index SNP for the same trait (Fig. 6f, Supplementary Data 11). For the other 167 signals, we tested their LD with SNPs reported in the GWAS catalog and found that only for 13 signals (6.2%), the lead variant was in high LD ($R^2 > 0.8$) and for 25 (11.9%) the lead variant was in low to moderate LD ($0.2 \leq R^2 \leq 0.8$), indicating that the causal variant is likely not the reported index SNP but rather in LD with it. For the remaining 129 (61.4%) "novel" signals, the putative causal SNP that we identified was in a locus not associated with the same trait in the GWAS catalog. These results show that colocalizing eQTLs with GWAS signals from the UK BioBank identifies putative causal variants associated with cardiac traits at hundreds of loci, of which the majority are novel.

## Insights from fine mapping into the molecular underpinnings of GWAS signals

In some cases, our analysis provides a spatiotemporal context for previously identified putative causal variants. For instance, we found that the SNP (rs74181299) with the highest PPA for atrial fibrillation and the cardiac muscle-eQTL for *CEP68* (PPA ≈ 1, Fig. 6g–i), is in moderate LD ($R^2 = 0.75$) with the previously identified index variant rs2723064 (PPA ≈ 0, 4.2 kb upstream)[2,67]. We also found that the SNP (rs2738413) with the highest PPA for atrial fibrillation and the cardiac muscle- eQTL in the intron of *SYNE2* (Figs. 2g, 5b–d) had been previously identified as an index variant[2,4], but proposed to affect expression of the estrogen receptor *ESR2*[4], which is located downstream of *SYNE2*, as 17β-estradiol has arrhythmogenic effects on cardiomyocytes[68]. Since *SYNE2* encodes a nesprin protein, which affects the mechanical properties of the actin cytoskeleton[69,70], and *ESR2* is not expressed in cardiac samples, our results suggest that the most likely mechanism underlying the association between rs2738413 and atrial fibrillation involves changes in the expression of *SYNE2* in cardiac muscle cells. Similarly, the SNP (rs11000060) with the highest PPA for pulse pressure and the fetal-like-eQTL for *PRKG1* (isoform ENST00000643582.1_1, PPA = 0.38, Fig. 5e, Supplementary Fig. 11, Fig. 6j–l) had been previously identified as the index variant[71].

On the other hand, many of the fine-mapped GWAS signals were novel (i.e., not in GWAS catalog). For instance, the SNP (rs1708618) with the highest PPA for pulse rate and the eQTL for an *FLCN* isoform (ENST00000389168.6_2) (PPA ≈ 1, Fig. 6m–o) had not previously been described in cardiac-related GWAS. However, it has been shown that loss of *FLCN* in the heart results in excess energy and upregulated mitochondrial metabolism, suggesting an important role of this gene in cardiac homeostasis[72]. The GWAS SNP (rs12724121) with the highest PPA for pulse rate and the eQTL for *ACTN2* (Fig. 6p–r) is another example of a novel finding. *ACTN2* encodes alpha-actinin-2, a major component of the sarcomere Z-disc expressed in cardiac and skeletal muscle cells. Rare missense mutations in *ACTN2* result in ventricular fibrillation, cardiomyopathy, and sudden death[73,74], and a distal variant (rs535411) associated with heart failure in a GWAS[75] was in low $R^2$ but high D′ with rs12724121 (D′ = 1; $R^2 = 0.035$, 28.5 kb upstream), suggesting that a potential synthetic association[76,77] exists between these two variants. Our results indicate that rs12724121 may regulate pulse rate (i.e., a proxy for heart rate) through the altered expression of *ACTN2* and, since elevated resting heart rate may cause heart failure[78], our

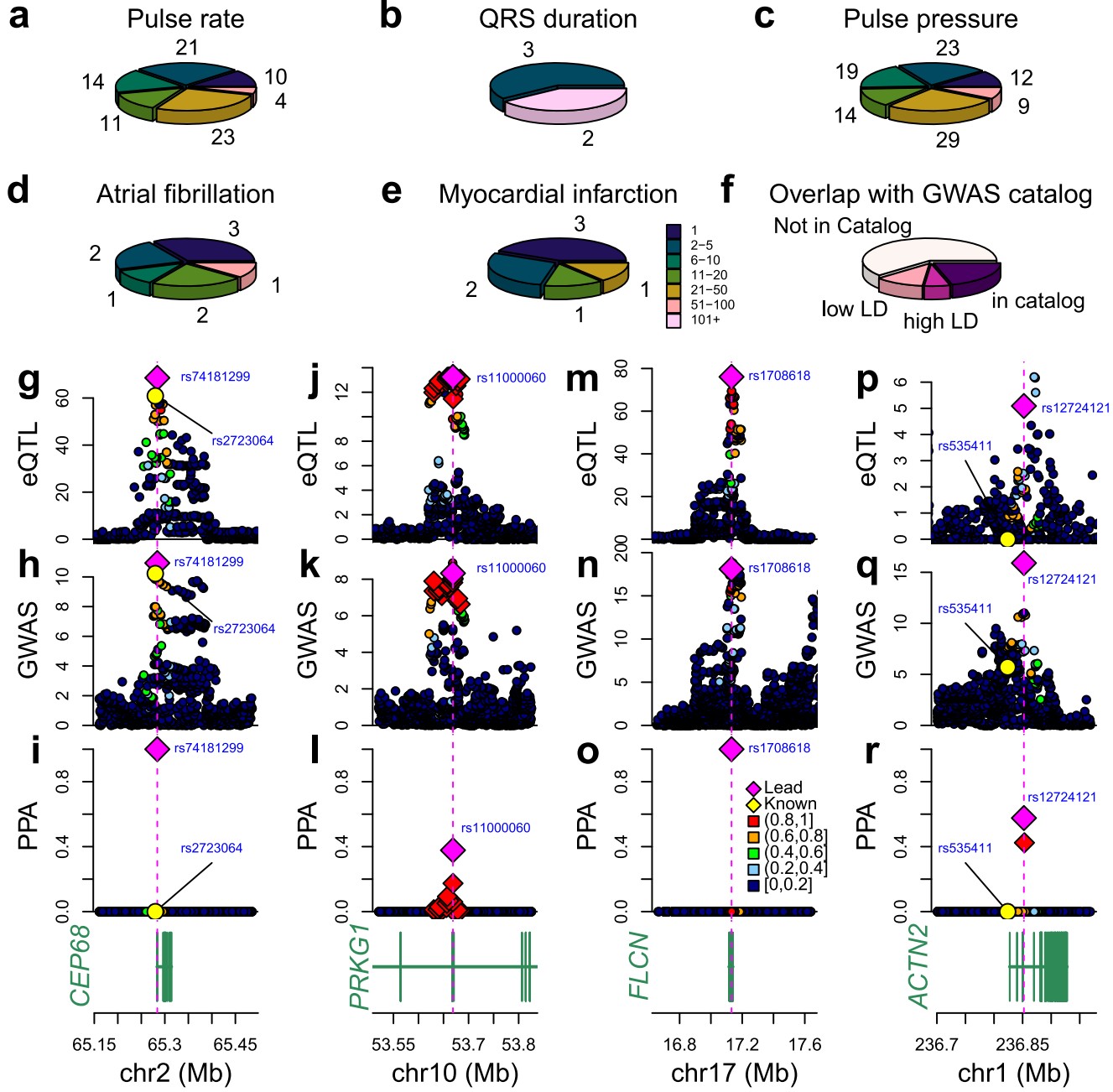

**Fig. 6 | Fine mapping of stage and cell type- eQTLs colocalizing with cardiac traits.** **a**–**e** Pie charts showing for each cardiac trait the distribution of SNPs in 99% credible sets. The colors indicate the number of SNPs and the numbers around the perimeter indicate the number of credible sets. **f** Pie chart showing the overlap between the lead SNP at each GWAS signal in the five cardiac traits and the index SNP for the GWAS signal in the GWAS catalog for the same trait. SNPs in high LD have $R^2 > 0.8$ and SNPs in low LD have $0.2 \leq R^2 \leq 0.8$. **g**–**o** Plots showing $-\log_{10}(p$-value) eQTL signals (top row; panels **g**, **j**, **m**, **p**), the GWAS signals (middle row; panels **h**, **k**, **n**, **q**) and the PPA of each variant in the colocalization (bottom row; panels **i**, **l**, **o**, **r**) at four loci: (**g**–**i**) *CEP68* expression and atrial fibrillation; **j**–**l** *PRKG1* (isoform ENST00000643582.1_1) expression and pulse pressure; **m**–**o** *FLCN* expression and pulse rate; and **p**–**r** *ACTN2* expression and pulse rate.

findings may explain the previously observed GWAS association between this locus and heart failure. Of note, these observations show the importance of investigating LD in terms of D' since synthetic associations[76,77] between variants can exist at GWAS loci.

In summary, our fine mapping analysis identified the likely causal SNPs in 81 previously characterized GWAS loci and 129 novel GWAS loci, demonstrating the power of this approach to uncover insights into the biology of cardiac traits and disease.

## Discussion

We have conducted one of the largest eQTL analyses of human cardiac samples considering both eGenes and eIsoforms and taking spatiotemporal context into consideration. We combined 180 fetal-like cardiac samples (iPSC-CVPC) with 786 adult cardiac samples from multiple tissues, including atrium, ventricle, aorta, and coronary artery. We showed that 44.9% of eIsoforms were either not associated with an eGene or had a different eQTL signal than their associated eGene; and that eGenes were enriched for being associated with regulatory variants at promoters and enhancers while eIsoforms were enriched for being associated with variants that affect post-transcriptional modifications. These results were consistent with previous studies, which described an enrichment in regulatory elements for variants affecting gene expression[25,79] and transcribed regions for variants affecting alternative splicing[25,80]. Taking spatiotemporal

information about the cardiac samples into account has provided us with the unique opportunity to identify more than 2,500 eQTL signals dependent on developmental stage, organ, tissue, and/or cell type. Using these data, we show that regulatory variants underlying eQTL signals shared between multiple eGenes are more likely to function in a spatiotemporal manner; and that eGenes which share the same eQTL signals tend to display the same spatiotemporal regulation. Our findings also indicate that fetal-like- eQTLs are less likely to be associated with multiple eGenes than adult eQTLs, which suggests that gene expression could be more individually regulated in early cardiac development. We demonstrate that, when paired genes and antisense transcripts are both eGenes, they are enriched for sharing the same eQTL signal, suggesting that a tight regulation of protein synthesis is required. Overall, we have generated an invaluable resource comprised of cardiac eQTL signals for 11,692 eGenes and 7165 eIsoforms, which can be used to understand the molecular mechanisms underlying the association of genetic variants with cardiac traits and disease.

Recent studies showed that certain GWAS loci are associated with tissue- and cell-type-specific regulatory elements and eQTLs[4,15,16,24,30]. Furthermore, GWAS of multiple cardiac traits, such as atrial fibrillation and PR interval, identified loci associated with embryonic development-associated genes and genes whose mutations are known to cause serious heart defects, such as *TTN*, *GATA4*, *MYH6*, *NKX2-5*, *PITX2*, and *TBX5*[2-4]. However, the extent to which regulatory variants that function in a spatiotemporal manner underlie cardiac trait GWAS signals was unknown. We showed that many eQTL signals that colocalize with cardiac GWAS traits function in a spatiotemporal manner. Furthermore, we found that three of the five traits examined in this study were enriched for eQTLs that function in specific spatiotemporal contexts. Surprisingly, pulse pressure was enriched for fetal-like-eQTLs, indicating that a subset of genetic variants associated with this adult trait exert their function during early cardiac development.

Using colocalization between eQTL and GWAS signals, we fine mapped 210 unique GWAS signals for five cardiac traits and disease. We were able to identify one single likely causal variant (with posterior probability of being causal >99%) for 28 of these GWAS signals, while for an additional 51 we were able to restrict the number of likely causal variants to fewer than five. Fifteen of these fine-mapped loci were associated with spatiotemporal eQTLs, including four that were iPSC-CVPC-eQTLs (*AGPAT1*, *CBWD1*, *RPL13*, and *PRKG1*). For 81 (38.6%) of the 210 GWAS signals, the SNP with the highest PPA in the credible was either the index SNP (43 signals) or in LD (38 signals) with the index SNP for the same trait in the GWAS catalog. For the remaining 129 (61.4%) GWAS signals, the putative causal SNP that we identified was in a locus not associated with the same trait in the GWAS catalog. These findings show that fine mapping provides the blueprint to both understand the molecular mechanisms underlying known and novel GWAS loci and to uncover insights into the biology of important cardiac traits and disease. Of note, there remains hundreds of GWAS signals that did not colocalize with eQTLs, indicating that a substantial fraction of the inherited component of human traits is not explained by regulatory variation, which is consistent with the recent findings of others[81]. Therefore, future studies on the functional characterization of genetic variants will likely require integration with other functional information, such as epigenomic data or trans-eQTLs to identify candidate causal variants[37]. Overall, we show that fine mapping using our cardiac eQTL resource identified the causal variant underlying hundreds of GWAS signals in five cardiac traits, led to an understanding of the underlying spatiotemporal context, and provided novel insights into the biology of the corresponding cardiac GWAS trait.

## Methods
### Data processing
We obtained RNA-seq and WGS data from 491 individuals from two sources: 180 iPSC-CVPC samples from 139 subjects from the iPSCORE

Collection[33,82] and 786 adult cardiac samples (227 aorta, 125 coronary artery, 196 atrial appendage and 238 left ventricle) from 352 GTEx subjects[25] (Supplementary Data 1). 23 iPSCORE subjects had more than one iPSC-CVPC sample (range: 2–6), whereas 228 GTEx individuals had samples for at least two of the four adult cardiac tissues (range: 2–4). iPSCORE iPSC-CVPCs were derived using a standardized protocol followed by lactate purification[36,38,39]. All 180 samples were collected at day 25 differentiation and represent fetal cardiac cells, as we and others have previously determined[36-39].

**RNA-seq data.** RNA-seq data for all 966 cardiac samples (180 iPSC-CVPC and 786 adults) was obtained from dbGaP (phs000924 and phs000424) and processed as follows[36,83]. Briefly, FASTQ files were aligned to the hg19 reference genome using STAR 2.5.0a[84] and Gencode V.34lift37[85] with parameters *outFilterMultimapNmax 20, --outFilterMismatchNmax 999, --alignIntronMin 20, --alignIntronMax 1000000, ---alignMates-GapMax 1000000*. We sorted the BAM files using Sambamba 0.6.7[86] and marked duplicates using biobambam2 (2.0.95) *bammarkduplicates*[87]. To quantify TPM gene expression and relative isoform abundance, we used RSEM V.1.2.20[88] with options *rsem-calculate-expression --bam --num-threads 16 --no-bam-output --seed 3272015 --estimate-rspd --paired-end --forward-prob 0*. Using this pipeline, we determined the expression of 62,492 genes and their corresponding 229,835 isoforms. Only 19,586 autosomal genes with TPM ≥ 1 in at least 10% of the 966 samples were considered as expressed and used for eQTL analysis. Likewise, 37,032 isoforms (TPM ≥ 1 and usage >10% in at least 10% of the 966 samples) from 10,337 expressed genes were used for isoform eQTL analysis.

To normalize gene and isoform expression and usage across samples, we performed quantile-normalization using *normalize.quantiles* (preprocessCore 1.50.0) and *qnorm* functions in R, in order to obtain mean expression = 0 and standard deviation = 1[29,89].

We deconvoluted cell type proportions for all 966 cardiac RNA-seq samples as follows[29]. Briefly, we determined marker genes for eight cardiac cell types (cardiac muscle, cardiac neuron, endocardial, endothelial, fibroblast, immune, myofibroblast, and smooth muscle) using single-cell RNA-seq[30,90,91] and used their average expression levels in each cell type to deconvolute cell type proportions using CIBERSORT[27].

**WGS data.** VCF files from WGS data were obtained from dbGaP (phs001325 and phs000424). We retained variants with MAF > 1% in both studies, that were in Hardy-Weinberg equilibrium ($p > 10^{-6}$), and that were within 500 kb of any expressed gene. Specifically, we expanded the coordinates of each of the 19,586 expressed genes (500 kb upstream and downstream) and extracted all variants in these regions using *bcftools V.1.9 view* with parameters *-f PASS -q 0.01:minor*. Next, we merged the resulting VCF files (*bcftools merge -m none*), normalized indels and split multiallelic variants (*bcftools norm -m*) and removed variants that were genotyped in fewer than 99% of samples (*bcftools filter -i 'F_PASS(GT!="mis") > 0.99'*). Finally, we converted the resulting VCF files to text using *bcftools query* and converted the genotypes from character strings (0/0, 0/1, and 1/1) to numeric (0, 0.5, and 1, respectively). This resulted in 4,962,200 total variants that were common between the two studies (iPSCORE and GTEx) and used for eQTL mapping.

### Combined eQTL analysis
To identify genetic variants that are associated with cardiac gene expression, we performed a joint eQTL analysis using all 966 samples. For each expressed gene, we used *bcftools query*[92] to obtain the genotypes for all the variants within 500 kb of each autosomal gene's coordinates (see above for details). To account for genetic relatedness between samples, we performed eQTL mapping using a linear mixed model (LMM) with limix v.3.0.4[47,93] (*scan* function), which incorporates

a kinship matrix as a random effect:

$$Y_i = \beta_j X_{ij} + \sum_{m=1}^{M} \gamma_m PC_{im} + \sum_{n=1}^{N} \gamma_n PEER_{in} + \sum_{p=1}^{P} \gamma_p C_{ip} + u_i + \epsilon_{ij} \quad (1)$$

Where $Y_i$ is the normalized expression value for sample $i$, $\beta_j$ is the effect size (fixed effect) of SNP $j$, $X_{ij}$ is the genotype of sample $i$ at SNP $j$, $PC_{im}$ is the value of the $m$th genotype principal component for the individual associated with sample $i$, $\gamma_m$ is the effect size of the $m$th genotype principal component, $M$ is the number of principal components used (M = 20), $PEER_{in}$ is the value of the $n$th PEER factor for sample $i$, $\gamma_n$ is the effect size of the $n$th PEER factor, $N$ is the number of PEER factors used, $C_{ip}$ is the value of the $p$th covariate for sample $i$, $\gamma_p$ is the effect size of the $p$th covariate, $P$ is the number of covariates used, $u_i$ is a vector of random effects for the individual associated with sample $i$ defined from the kinship matrix, and $\epsilon_{ij}$ is the error term for individual $i$ at SNP $j$.

**Conditional eQTL signals.** To identify additional independent eQTL associations for a gene or isoform (i.e., conditional eQTLs), we performed a stepwise regression analysis in which we re-performed eQTL analysis and included the genotype of the lead eQTL as a covariate. We repeated the analysis to discover up to five conditional associations. For each iteration, we performed the two-step procedure described above and considered conditional eQTLs with $q$-values <0.05 as significant.

**FDR correction.** To perform FDR correction, we used a two-step procedure similar to the one described in Huang et al.[94] that first corrects at the gene level and then at the genome-wide level. (1) For each gene, $p$-values were FDR-corrected using eigenMT, which takes into account the LD structure of the tested variants; and (2) across the lead variants for all the 19,586 tested genes and 37,032 isoforms, $p$-values were corrected for false discovery rate (FDR) using Benjamini-Hochberg's correction and considered only eQTLs with $q$-values < 0.05 as significant.

Summary statistics for all eQTLs are reported in "Figshare [https://doi.org/10.6084/m9.figshare.c.5594121]"[44].

**Covariates for eQTL mapping**
We performed eQTL mapping using the following covariates (Supplementary Data 1): (1) sex; (2) normalized number of RNA-seq reads; (3) % of reads mapping to autosomes or sex chromosome; (4) % of mitochondrial reads; (5) 20 genotype principal components to account for global ancestry; (6) 285 PEER factors to account for transcriptome variability; and (7) kinship matrix.

**Genotype principal component analysis.** We performed principal component analysis (PCA) on WGS variants to determine the global ancestry of each individual in this study[29]. Briefly, we used the genotypes of 1,634,010 SNPs that had allele frequencies between 30 and 60% in the 1000 Genomes Phase 3 Project and genotyped in both iPSCORE and GTEx. We merged the VCF files from 1000 Genomes, iPSCORE, and GTEx, and performed PCA using the *pca* function in plink 1.90b3x[95]. We showed that iPSCORE and GTEx individuals overlap their associated 1000 Genomes superpopulations, and that iPSCORE includes more individuals of East Asian ancestry than GTEx. The top 20 genotype principal components used to account for global ancestry are available for all 491 individuals (139 iPSCORE and 352 GTEx) in the expanded Supplementary Data 1 in "Figshare [https://doi.org/10.6084/m9.figshare.c.5594121]".

**PEER factors.** We sought to determine the optimal number of PEER factors[96] for obtaining the maximum number of eQTLs. We initially calculated PEER factors on the 10,000 expressed genes

with the largest variance across all samples using the R package peer 1.0. To limit biases due to the expression levels of each gene, we divided the 19,586 expressed genes into ten deciles based on their average TPM, and selected 20 random genes for each decile, for a total of 200 genes. We next performed eQTL analysis on each of the 200 genes using the following numbers of PEER factors: 5, 10, 20, and increments of 20 between 20 and 300. These eQTL analyses of the 200 genes to determine the optimal number of PEER factors was performed using the same covariates as for the combined eQTL analysis: (1) sex; (2) normalized number of RNA-seq reads; (3) % of reads mapping to autosomes or sex chromosome; (4) % of mitochondrial reads; (5) 20 genotype principal components to account for global ancestry; (6) the selected number of PEER factors to account for transcriptome variability; and (7) kinship matrix. We observed the maximum number of eQTLs using 280 PEER factors. To further refine the optimal number of PEER factors, we performed a second eQTL analysis using 270, 275, 285, and 290 PEER factors, which determined that 285 PEER factors were optimal (Supplementary Fig. 18A). For isoform eQTLs, we used 80 PEER factors as a compromise between obtaining the largest number of eQTLs and computational burden. To confirm that the PEER factors were accounting for batch effects between the iPSCORE and GTEx samples, we examined the distribution of the samples in PEER space and the correlation of the top 10 PEER factors with other covariates (Supplementary Fig. 18B). We observed that the PEER factors were distinctly correlated with iPSCORE and GTEx datasets, as well as other technical covariates (Supplementary Fig. 18C). These analyses show that batch effects were properly corrected for between the iPSCORE and GTEx datasets.

**Kinship matrix.** The random effects term captures relatedness between samples and are described in a kinship matrix. To construct the kinship matrix, we used the kinship function in plink 1.90b3x[97] using the same set of 1,634,010 SNPs employed in the genotype PCA.

**Testing enrichment of eQTL signals in different genomic annotations**
To test the enrichment of eQTLs in intergenic regions, introns, promoters, UTRs, splice donor sites, splice acceptor sites and exons, we obtained fine-mapped variants for all eQTL signals for both eGenes and eIsoforms.

**Determining the 99% credible sets for each eQTL signal.** To define a credible set of candidate causal variants for each eQTL association, we performed genetic fine mapping using the *finemap.abf* function in the *coloc 5.1.0.1* R package[45]. This Bayesian method converts $p$-values of all the variants tested for a specific gene to posterior probabilities of association (PPA).

**Testing enrichment of eQTL signals in different genomic annotations.** We obtained fine-mapped variants for all eQTL signals for both eGenes and eIsoforms with PPA > 0.01. We determined the overlap of each fine mapped variant and each genomic annotation using *bedtools intersect* and compared the proportions of variants overlapping each annotation between eGenes and eIsoforms using Fisher's Exact Test. Promoters were defined as the 2000 bp upstream of the transcription start site. Exons and UTRs were obtained from *Gencode V.34lift37*. For the intronic variants, we calculated their distance from the closest exon to determine their overlap with splice sites.

**Colocalization between eQTL signals**
For each of the 5744 eIsoforms whose associated genes had eQTLs, we performed colocalization using the *coloc.abf* function from the *coloc*

package in R[45]. This Bayesian method uses p-values of the variants tested for two traits (eIsoform and eGene) to calculate the posterior probabilities of association (PPA) for each of the five hypotheses at a specific locus: (1) H0: neither trait has a significant association at the tested locus; (2) H1: only the first trait is associated; (3) H2: only the second trait is associated; (4) H3: both traits are associated but the underlying variants are different; and (5) H4: both traits are associated and share the same underlying variants. Since multiple eQTL signals may be present for both eIsoforms and eGenes (primary and conditional eQTLs), for each eIsoform/eGene pair, we considered only the colocalization with highest PP-H4.

To determine whether two eGenes or eIsoforms from different genes shared the same eQTL signal, we used the same colocalization approach as described above.

To determine whether paired genes and antisense RNAs were enriched for sharing an eQTL signal, we obtained the gene symbols of all eGenes from Gencode. We identified 163 gene/antisense pairs where both the gene and its antisense RNA were eGenes and whose gene symbols were "Gene A" and "Gene-A-AS1", respectively. We found that 47 of these pairs shared the same eQTL signal (PP-H4 > 0.8) and tested whether the proportion of colocalized eQTL was greater than expected between any pair of eGenes that were within 500 kb of each other (146,472 pairs) using a Fisher's Exact Test.

To determine if eQTLs associated with gene/antisense pairs that act in the same versus opposite directions have differential enrichments, we obtained all fine-mapped variants with PPA > 0.01 (from colocalization) and annotated each variant by its location relative to both the gene and antisense (intergenic, introns, promoters, UTRs, splice donor sites, splice acceptor sites, and exons). We then used a Fisher's Exact Test to calculate the enrichment by comparing the proportion of SNPs within the genomic region between correlated and anti-correlated gene/antisense pairs. P-values were adjusted using Benjamini-Hochberg's method. Enrichments with FDR < 0.05 were considered significant. This analysis was repeated for the comparison between pairs that shared the same eQTL signal (PP-H4 > 0.8) and pairs that had distinct signals (PP-H3 > 0.8). For the gene/antisense pairs with distinct signals (PP-H3 > 0.8), we used all fine mapped variants (from *finemap.abf*) for both genes and antisense with PPA > 0.01.

## Spatiotemporal eQTL mapping

To detect eQTL associations with cell types, tissue or developmental stage, we obtained the lead variant for each significant eQTL from the combined analysis for each eGene and eIsoform and used the same linear mixed model described above and included an interaction between genotype and context. Since each eQTL signal underlies one single causal variant, the spatiotemporal context of the lead variant is sufficient to annotate the whole eQTL signal. We considered a total of 16 different contexts: two organs (arteria and heart), four tissues (atrium, ventricle, aorta, and coronary artery), eight cell types (cardiac muscle, smooth muscle, endothelial, fibroblast, immune, cardiac neuron, endocardial, and myofibroblast), and two developmental stages (fetal-like iPSC-CVPC and adult, Supplementary Data 4). Using the *iscan* function in limix[47], we implemented the following linear mixed model, which we applied to the lead variant of each eQTL signal (primary and conditional):

$$Y_i = \beta_{1j}X_{ij} + \beta_{2k}Z_{ik} + \beta_{3jk}X_{ij}Z_{ik} + \sum_{q=1}^{Q} \gamma_q C_{iq} + u_i + \epsilon_{ij} \quad (2)$$

where $Y_i$ is the normalized expression value for sample $i$, $\beta_{1j}$ is the effect size (fixed effect) of SNP $j$, $X_{ij}$ is the genotype of sample $i$ at SNP $j$, $\beta_{2k}$ is the effect size of cell type $k$, $Z_{ik}$ is the fraction of cell type $k$ for

sample $i$, $\beta_{3jk}$ is the effect size of the interaction between genotype $X_{ij}$ and cell type $Z_{ik}$, $C_{iq}$ is the value of the $q$th covariate for sample $i$, $\gamma_q$ is the effect size of the $q$th covariate, $Q$ is the number of covariates used (as defined for the eQTL analysis described above), $u_i$ is a vector of random effects for the individual associated with sample $i$ defined from the kinship matrix, and $\epsilon_{ij}$ is the error term for individual $i$ at SNP $j$.

To perform FDR correction, we used Bonferroni's method to correct p-values across all variants tested for each context.

**Additional filtering steps to determine context-specific and context-associated eQTLs.** While the interaction LMM described above identifies context-specific eQTLs, an interaction would also be considered significant if the eQTL is active in all other samples but the context being tested (for example: if we are testing the interaction with "left ventricle", the eQTLs that are significant in all the other tissues but left ventricle would have a significant p-value). To avoid counting these associations as "context-specific" we included the following filtering step for all spatiotemporal eQTLs (i.e., those with a significant interaction term in the above analysis).

For binary contexts (developmental stage, tissue, and organ), we divided all samples into two sets corresponding to the two binary values (for example: (1) left ventricle; (2) all the other samples). We performed an additional standard eQTL analysis on each of the two sets. P-values were corrected across all interactions using Benjamini-Hochberg's method. Associations with adjusted p-value < 0.05 were considered significant. We considered the following four scenarios:

1. eQTL is significant only for the tested context (for example: left ventricle): the eQTL signal is considered as context-specific (left ventricle-specific);
2. eQTL is significant only for "all the other samples": the eQTL is not considered as context-specific;
3. eQTL is significant for both sets but its effect size is stronger (larger absolute value) in the tested context: the eQTL is context-associated (left ventricle-associated);
4. eQTL is significant for both sets but its effect size is stronger (larger absolute value) in "all the other samples": the eQTL is not considered as context-specific or context-associated.

For cell types, we divided all samples into quartiles and analyzed the top and bottom quartile sets. When analyzing cell types, the following four scenarios may occur:

1. eQTL is significant only for the tested cell type (i.e., the top quartile): the eQTL signal is considered cell type-specific;
2. eQTL is significant only for the bottom quartile: the eQTL is not considered as cell type-specific;
3. eQTL is significant for both the top and bottom quartiles but its effect size is stronger (larger absolute value) in the top quartile: the eQTL is cell type-associated;
4. eQTL is significant for both the top and bottom quartiles but its effect size is stronger (larger absolute value) in the bottom quartile: the eQTL is not considered as cell type-associated.

In Supplementary Figures, we show examples of eQTLs with significant interactions with stage (Supplementary Fig. 1A), tissue (Supplementary Fig. 2), and cell type (Supplementary Fig. 3). Below, we provide the description of the different methods that we used to validate the cell type associations of eQTLs.

**Validation of cell type-associated eQTLs (intersection with adult heart snATAC-seq peaks).** To validate the eQTLs associated with spatiotemporal contexts, we obtained the relative accessibility score (RAS) and cell type-specific information for 286,725 snATAC-seq peaks obtained from human adult hearts[48]. Peak coordinates were lifted over to the hg19 genome using liftOver.

Across all eQTL associations, we obtained 201,082 fine mapped variants with PPA > 0.01 (see section "Determining the 99% credible sets for each eQTL signal") and intersected their coordinates with the snATAC-seq peaks using *bedtools V.2.27.1 intersect*. We found that 18,928 variants (corresponding to 10,180 eGenes, including 367 associated with cardiac muscle) overlapped snATAC-seq peaks. If multiple variants were associated with the same gene and overlapped snATAC-seq peaks, we considered only the peak that overlapped with the variant with the highest PPA.

Using the eQTL-intersected peaks, we then tested the enrichment of cardiac muscle-associated eQTLs (corresponding to 367 eGenes) in peaks that are specific to each of the nine cell types detected in snATAC-seq, including atrial cardiomyocyte, ventricular cardiomyocyte, adipocyte, fibroblast, endothelial, smooth muscle, lymphocyte, macrophage and nervous. Specifically, we used a paired t-test comparing the RAS of peaks that are specific to a cell type compared to peaks that are specific to all other cell types. Results are shown in Supplementary Fig. 4 and available in Supplementary Data 5.

### Validation of cell type-associated eQTLs (comparison with GTEx cell-type associated eQTLs).

To validate the eQTLs associated with spatiotemporal contexts we also compared the correlation of effect sizes with cell type-associated eQTLs identified by the GTEx Consortium[24]. We downloaded myocyte interaction eQTLs (ieQTLs) for heart left ventricle from the GTEx Data Portal and calculated the Pearson correlation coefficient between effect sizes for matching eGenes using the *cor.test* function in R. We observed positive associations between GTEx ieQTLs for myocytes in heart left ventricle and cardiac muscle-specific/associated, heart-specific/associated eQTLs, and heart ventricle-specific/associated eQTLs (Supplementary Fig. 5).

### Enrichment of spatiotemporally regulated eQTLs in multigenic eQTLs.

To determine if two eGenes or eIsoforms had the same underlying eQTL signal, we used an approach similar to the one described in the section "Colocalization between eQTL signals". Briefly, we obtained all pairs of eGenes within 500 kb of each other and performed colocalization between each combination of primary and conditional eQTL signals for both eGenes. Two eGenes or eIsoforms colocalized if they had PP-H4 > 0.8. Next, we tested the association between the number of genes that share the same eQTL signal and the spatiotemporal context of the eQTL signal (coded as a binary variable: 1 = eGenes are associated with the spatiotemporal context; 0 otherwise) using linear regression. To confirm that eQTL signals associated with multiple eGenes were more likely than expected to be associated with the same spatiotemporal context and to ensure that enrichments were not due to a small number of signals shared by many eGenes (certain eGenes may have a stronger weight on the overall analysis than other), we also performed a permutation analysis. We counted the number of occurrences where the same spatiotemporal context was shared between two colocalizing eGenes and compared it with the distribution of 100 permutations of each eGene-spatiotemporal context association. We calculated Z-scores and converted them to *p*-values using the *pnorm* function in R.

### Colocalization between GWAS and eQTL signals

We obtained GWAS summary statistics for five cardiac traits from the "Pan UK BioBank repository [https://pan.ukbb.broadinstitute.org/]". All data was obtained in hg19 coordinates. We sorted and indexed all of the files using *tabix*[92] and, for each trait, extracted all genome-wide significant SNPs (meta-analysis *p*-value 5 × 10⁻⁸). We found 1444 eGenes and 919 eIsoforms overlapped or were in close proximity (<500 kb) with genome-wide significant GWAS variants. We performed colocalization between each significant eQTL signal and all genome-wide significant GWAS signals within the same locus (defined as GWAS signal and corresponding eGene or eIsoform <500 kb apart) using the

*coloc.abf* function in R[45]. Colocalization tests if two genetic association signals (such as eQTL and GWAS) have the same underlying causal variant by comparing the effect sizes of all the variants genotyped at the locus of interest. Colocalization was performed only on loci that included at least one genome-wide significant eVariant and one genome-wide significant GWAS variant. For downstream analyses, we considered a trait to colocalize with a gene if the GWAS signal colocalized (PP-H4 > 0.8) with the eGene or any of its eIsoforms, while we discarded all the colocalizations with high PP-H3. All colocalization results have been deposited to "Figshare [https://doi.org/10.6084/m9.figshare.c.5594121]"[44].

To identify all GWAS loci that colocalized with at least one eQTL, we calculated the linkage disequilibrium (LD) between all pairwise combinations of lead SNPs across all 331 colocalizations. We next identified clusters of eQTL signals using the *cluster.louvain* function in *igraph 1.3.5* that shared the same lead SNP or had lead SNPs in high LD (D′ > 0.8). This analysis resulted in 210 independent GWAS signals.

### Fine-mapping and obtaining 99% credible sets for GWAS signals.

To test the utility of cardiac eQTLs to identify potential causal variants for cardiac traits, we performed fine mapping using two different methods:

1. Standard genetic fine mapping[14], which converts the *p*-value of each variant at each of the 210 GWAS loci to PPA, without considering the colocalization with eQTLs. We used the *finemap.abf* function from the coloc package in R[45] on the GWAS summary statistics at each of the 210 loci. 99% credible sets describe the smallest set of variants that cumulatively contribute to ≥99% of the PPA in each locus.

2. We performed fine-mapping on each of the 210 GWAS signals using the *coloc.abf* function in R[45], which provides the PPA that each tested variant is causal for the GWAS and eQTL associations. For each colocalization (one or more eQTL signals may colocalize with the same GWAS locus), we constructed 99% credible sets as described above. For each of the 210 independent signals (which colocalized with one or more eQTLs), we selected the credible set that had the fewest SNPs. If two credible sets had the same number of SNPs, the set containing the lead variant with the highest PPA was chosen.

### Enrichment of context-associated eQTLs for GWAS cardiac traits.

Enrichment of context-associated eQTLs in cardiac traits was calculated using a Fisher's Exact Test at multiple PP-H4 thresholds (0–0.95; at 0.05 increments), where the contingency table consisted of two classifications: (1) if the eQTL signal was significantly context-associated (FDR < 0.05); and (2) if the eQTL signal colocalized with the GWAS trait greater than the PP-H4 threshold. *P*-values were corrected using Benjamini-Hochberg across all tests for each trait. Traits were considered to be context-associated if they had an FDR-corrected *p*-value < 0.1 at the 0.8 threshold.

### Validation of the enrichment of tissue-associated eQTLs for GWAS cardiac traits.

To validate enrichment of tissue-specific eQTLs in the cardiac GWAS signals we used the multi-variate adaptive shrinkage (mash) method[98]—see Supplementary Note 2 for details on this method. We initially performed independent tissue-specific eQTL analyses on five tissues (iPSC-CVPC, arteria, aorta, ventricle, and atrium) using a linear mixed model (limix) and the same covariates as our combined eQTL approach: sex, normalized number of RNA-seq reads, % of reads mapping to autosomes or sex chromosome, % of mitochondrial reads, 20 genotype PCs, and kinship matrix, and recalculated the number of optimal PEER factors. Gene expression was quantile-normalized across all samples within each of the five datasets, and the number of optimal PEER factors was determined using the same method described for the combined eQTL analysis: we obtained

200 random genes and performed eQTL analyses using a range of PEER factors and selected the number of PEER factors for each dataset associated with the maximum number of detected eQTLs for these 200 genes. We used the following number of PEER factors: 40 for iPSC-CVPC, 60 for atrium, 70 for ventricle, 70 for aorta, and 30 for coronary artery. The kinship matrix was included as a random effect. Conditional eQTL analysis and FDR correction was performed using the same method as our combined eQTL approach (Supplementary Data 9).

To have a direct comparison between mash and our outputs for validation, we next examined the mash results obtained using the eQTLs from the first-step of our combined approach (for eGenes, not for eIsoforms; Fig. 1a) coupled with their effect sizes and standard errors for each tissue from the single-tissue eQTL analyses as input—see Supplementary Note 2 for details. We ran mash with default parameters and obtained the local false sign rate (LFSR)[64,98]. We annotated each eQTL signal as associated with a tissue if the LFSR < 0.05 (Supplementary Data 10). We also filtered at more stringent thresholds (up to $1 \times 10^{-30}$). At the least stringent LFSR threshold (LFSR = 0.05), we observed <11,500 eQTLs in each of the four adult tissues and 8856 in iPSC-CVPCs (Supplementary Fig. 13). Using the eQTLs detected by mash for each tissue, we performed enrichment tests across the five cardiac GWAS traits, as described in the previous section (Supplementary Fig. 14).

While mash is useful to validate the tissue-associations that we identified, our method and mash provide different information: we focused on identifying eQTL signals that are tissue-specific and, to do so, we developed a very stringent approach, which likely results in a large number of false negatives (i.e., tissue-specific eQTLs that we do not classify as such); on the other hand, mash uses information from other tissues to improve power to detect eQTLs in a specific tissue. Therefore, mash is able to identify eQTLs that are "active" in a tissue because they occur in multiple other tissues, rather than describing tissue-specific eQTLs.

**Enrichment of GWAS causal variants in heart chromatin states.** Across all 210 fine-mapped GWAS loci, we obtained 2157 SNPs with PPA ≥ 0.01 and tested their enrichment in Roadmap Chromatin States for fetal heart, adult right atrium, adult left ventricle, and adult right ventricle using a linear regression model with the *lm* function in R where binary state annotation was modeled as a function of SNP PPA. Enrichment *p*-values were FDR-corrected (Benjamini-Hochberg).

**Prediction of GWAS causal variants for impacting disease.** We provided a VCF of all 2157 SNPs with PPA ≥ 0.01 in the 99% credible sets of the 210 fine-mapped GWAS loci as input into DeepSea[65,66], which is a deep-learning-based framework that predicts the chromatin effects of each SNP. One of the outputs of DeepSea (version "Beluga") is the disease impact score for each SNP, which was computed using a logistic regression model that prioritizes candidate disease-associated SNPs based on their predicted effects on transcriptional or post-transcriptional regulation. To determine whether GWAS candidate causal eQTLs are likely to impact disease, we used a linear regression model to test the association between SNP PPA and disease impact score.

**Reporting summary**
Further information on research design is available in the Nature Portfolio Reporting Summary linked to this article.

## Data availability
The genotype and transcriptomic data used in this study have been obtained from dbGaP studies "phs000424" (GTEx RNA-seq and whole-genome sequencing), "phs000924" (iPSCORE, RNA-seq), and "phs001325" (iPSCORE, whole-genome sequencing). Data for adult heart snATAC-seq peaks were obtained from Supplemental Table 5 in Hocker et al.[48]. GWAS summary statistics were obtained from the "Pan UK BioBank resource [https://pan.ukbb.broadinstitute.org/]". The GWAS "manifest file [https://docs.google.com/spreadsheets/d/1AeeADtT0U1AukIiiNyiVzVRdLYPkTbruQSk38DeutU8/edit#gid=511623409]" was downloaded from Pan UK BioBank resource. Summary statistics for all eQTLs, fine mapping, and the supporting data for figures and supplemental tables have been deposited in "Figshare".

## Code availability
All scripts developed to perform this study are available in "GitHub [https://github.com/TheMatteoLab/cardiac_eqtls]".

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

## Acknowledgements

This work was supported by a California Institute for Regenerative Medicine grant GC1R-06673-B, NSF-CMMI division award 1728497, and NIH grants HG008118, HL107442, HG011558, and 3RM1HG011558-02S1. J.P.N. and T.D.A. were supported by T15LM011271, F31DK131867, and F31HL158198.

## Author contributions

K.A.F., A.D.C., M.D. and iPSCORE consortium members conceived the study. M.D., J.P.N., T.D.A., H.M. and iPSCORE consortium members performed the analyses. H.M. performed quality check on RNA-seq samples. K.A.F., A.D.C. and iPSCORE consortium members oversaw the study. M.D., J.P.N. and K.A.F. prepared the manuscript.

## Competing interests

The authors declare no competing interests.

## Additional information

## iPSCORE Consortium

Angelo D. Arias[1], Timothy D. Arthur[2,4], Paola Benaglio[1], W. Travis Berggren[6], Victor Borja[5], Juan Carlos Izpisua Belmonte[7], Megan Cook[5], Matteo D'Antonio[1,2], Agnieszka D'Antonio-Chronowska[1], Christopher DeBoever[3], Kenneth E. Diffenderfer[6], Margaret K. R. Donovan[2,3], KathyJean Farnam[5], Kelly A. Frazer[1,5], Kyohei Fujita[1], Melvin Garcia[5], Olivier Harismendy[2], Benjamin A. Henson[5], David Jakubosky[2,4], Kristen Jepsen[5], He Li[1], Hiroko Matsui[5], Naoki Nariai[1], Jennifer P. Nguyen[2,3], Daniel T. O'Connor[8], Jonathan Okubo[5], Athanasia D. Panopoulos[7], Fengwen Rao[8], Joaquin Reyna[5], Bianca Salgado[5], Erin N. Smith[1], Josh Sohmer[5], Shawn Yost[3] & William W. Young Greenwald[3]

[6]Stem Cell Core, Salk Institute for Biological Studies, La Jolla, CA 92037, USA. [7]Gene Expression Laboratory, Salk Institute for Biological Studies, La Jolla, CA 92037, USA. [8]Department of Medicine, University of California, San Diego, La Jolla, CA 92093, USA. A full list of members and their affiliations appears in the Supplementary Information.

