## [Peer Review File · Nature Communications]

Fine mapping spatiotemporal mechanisms of genetic variants underlying cardiac traits and diseaseREVIEWER COMMENTS

Reviewer #1 (Remarks to the Author):

The manuscript "Fine mapping spatiotemporal mechanisms of genetic variants underlying cardiac traits and disease" by D'Antonio et al., characterizes for the first time genetic regulation of gene and isoform expression (eQTLs) in fetal cardiac tissues versus adult heart and artery tissues. They identify fetal, adult and cell type specific eQTLs in these tissues and use the eQTLs to investigate underlying causal regulatory mechanisms for genomic loci associated with five cardiac traits and diseases. To do this, the authors combined RNA-seq data from 966 fetal-like and adult cardiac tissues with whole genome sequencing of the donors' DNA from two studies (iPSCORE and GTEx) and used a linear regression mixed effects model to compute eQTLs. The fetal like tissues are induced pluripotent stem cell (iPSC) differentiated to cardiovascular precursor cells (iPSC-CVPCs), including cardiomyocytes and epicardium-derived cells. To identify eQTLs specific to early developmental stages versus adult cardiac (atrium, ventricle, aorta and coronary artery) tissues, or that are tissue or cell type specific the authors added an interaction term between genotype and each developmental stage, tissue or cell type (estimated with deconvolution analysis) to the regression model. The authors investigated the sharing of regulation between genes and isoforms and found that adult eQTLs are more likely to affect multiple genes than fetal eQTLs. They further performed colocalization analysis between known GWAS loci for five cardiac traits and co-occurring cardiac eQTLs and fine-mapping analysis which led to identification of eQTLs and 99% credible sets of variants likely to be causal to 210 GWAS loci. This work proposed fetal-, adult tissue-, and cell type-specific eQTLs that may contribute to cardiac traits, such as fetal- and smooth muscle-specific eQTLs for pulse pressure or cardiac muscle- eQTLs and atrial fibrillation.

While this is a commendable and important effort, I have several major comments on the methodologies chosen for the analyses presented in the study that may not be optimal for comparison of genetic regulation across different tissues and developmental stages, or for dealing with allelic heterogeneity in complex trait loci, that may affect the results and conclusions of this work.

Major comments:

- The authors combined 966 RNA-seq samples and WGS data from 180 fetal-like iPSC-CVPC samples from 139 subjects from the iPSCORE Collection and 786 adult cardiac samples from 2 heart and 2 artery tissues from 352 GTEx subjects, and performed eQTL analysis on the combined dataset using a mixed effects model to account for overlap in donors between different subsets of tissues. Tissue, organ, developmental stage and cell type specific eQTLs were assessed by adding a genotype by tissue/cell type interaction term to the model. I think there are more suitable and better powered approaches for comparing genetic regulation of gene expression between tissues, which account for differences in sample size, direction of effect of the eVariants, and allelic heterogeneity in the QTL loci and across tissues, such as the multi-variate adaptive shrinkage (mash) method (Urbut et al., Nature Genetics 2019, PMID: 30478440). In the least I would recommend performing eQTL analysis on each tissue separately and comparing results to those generated based on the combined analysis.
- Given that samples are from two different studies, it would be good to see how the samples from the two studies cluster in PCA space based on their gene expression profiles across all genes before and after PEER factor correction to make sure that batch effects are properly corrected for between the iPSCORE and GTEx samples. The authors chose to include 285 PEERs (inferred covariates of non-specific gene expression variation) in the eQTL regression model based on the largest eGene discovery of a test set of 200 genes as a function of number of PEERs (up to 300 PEERs). This number (N=285) seems very large for the given sample size. Can the authors please show the plot of number of identified eQTLs or eGenes as a function of number of PEERs to support this choice.
- Given that samples come from two different studies, it would be good to inspect the ancestral distribution of the iPSCORE samples compared to the GTEx samples based on the top genotype principal components (PCs) (e.g. plot PC1 vs PC2, PC1 vs PC3, PC2 vs PC3 for all samples colorcoding

points by study). To correct for potential differences in population background between samples and studies, in particular given that the samples are from two different studies (iPSCORE and GTEx), the authors included the top 20 genotype PCs in their eQTL regression model. A smaller number of genotype PCs might suffice (as adding more covariates to the model might lead to loss of power). I would recommend inspecting/plotting the variance explained by each PC (eigenvalue divided by the sum of all eigenvalues across all PCs, where n =total number of samples), and choosing the number of genotype PCs based on those components with the highest relative variance explained, that fall before the infection point. Can the authors present

- The authors used a set of 1,634,010 SNPs with an allele frequency between 0.3-0.6 in the 1000 Genomes Phase 3 project and genotyped in both GTEx and iPSCORE to generate genotype PCs. Did the authors perform LD pruning between these variants to ensure that the SNPs are genetically independent to each other? This usually yields on the order of 10^5 SNPs in WGS data. This is important to do prior to computing genotype PCs to properly represent the genetic variation between population backgrounds.
- Can the authors specify in the Methods section what method was used to reconstruct and quantify the isoforms? Was it using RSEM or another method?
- What normalization was performed on the number of RNA-seq reads per gene or isoform? For example, in GTEx v8 TMM normalization with EdgeR is applied followed by standard normalization.
- Did the authors include sex in eQTL regression model? It seems that they added sex only for estimating the optimal number of PEERs in the eQTL analysis. Can the authors clarify this point.
- To test enrichment of eQTLs in different functional genomic annotations, the authors inspected the overlap of the lead variant per eGene or eIsoform different genomic categories, including intergenic regions, introns, promoters, UTRs, splice sites, and exons. However, since the lead variant might not be the causal variant but rather tagging the causal variant via LD, I would recommend doing this analysis using all the significant variant-gene pairs per eGene/eIsoform or at least include the variants in LD (e.g., $r^2 > 0.8$) with the lead variant. Also, can the authors more clearly explain the enrichment test that was performed here. Another method that could be fitting to apply here is TORUS (<https://github.com/xqwen/torus>) that is a logistic regression model that estimates enrichment of eQTL variants in functional annotations.
- Can the authors please clarify how they determined developmental stage-, organ-, tissue-, and cell type-specific eQTLs. At first in the Methods section, it is stated that a linear mixed model with an interaction term is used between genotype and each stage/organ/tissue/cell type to detect association between eQTLs and stage/organ/tissue/cell type. However, afterwards, the authors describe another analysis performed to detect developmental stage, tissue and cell type-specific eQTLs, i.e. that samples were divided into two groups (one type vs. all others) and eQTL association was tested between each of these two groups and genotype, using linear regression analysis. For cell type-specific eQTLs, they obtained the samples in the top and bottom quartiles based on their deconvoluted cell type populations for each cell type. Can the authors explain why they performed these two approaches and how they compared between the two.
- Are there multiple developmental stages in the fetal RNA-seq dataset? It could be helpful for the reader to include a table that provides a breakdown of the various types of tissues, developmental stages, study name and number of samples per type.
- It would be interesting to compare the pathways in which the target genes of cell type specific eQTLs are enriched relative to cell type-shared eQTLs or eQTLs that are specific to early development vs. adult tissue-specific eQTLs.
- The authors found that that 47 pairs of genes and their antisense RNA share the same eQTL signal. I assume this means that the eQTLs acts in the same direction on both the gene and its antisense (which could lead to opposite regulation of the gene). Can the authors comment on whether they found separate eQTLs acting on a gene and its antisense gene in opposite directions, which would lead to accentuation of the gene's eQTL effect.
- The authors performed colocalization analysis between GWAS loci associated with five heart related complex diseases or traits and overlapping eQTLs in the cardiac tissues and cell types. They used the coloc method that assumes one causal variant per locus. Since we know that there is allelic heterogeneity in many GWAS and eQTL loci (as the authors showed in Figure 1a), it would be more

powerful to use other Bayesian colocalization methods, such as eCAVIAR (Hormozdiari et al., AJHG 2016) that can consider multiple causal variants in a locus, or CAFEH (Arvanitis et al., AJHG 2022, PMID: 35085493) that also accounts for tissue specificity of eQTL effects that may be well suited for this study.

Minor comments:

- Can the authors provide more details on how many variants were found common between the two studies, and how many remained after quality control (QC).
- In the first paragraph of the Results section "By regressing out the genotype of each lead variant, we observed that, on average, each eGene had 1.54 eQTLs (range: 1-6)," for clarification, I would add the word "independent" to "1.54 independent eQTLs".
- I think there is a typo in the sentence: " γ_n is the effect size of the n th genotype principal component," I think 'n' is supposed to be 'm'.
- In the Methods section under "Colocalization between GWAS and eQTL signals", I would recommend the authors add the word "GWAS" before "summary statistics": "We obtained summary statistics from five cardiac traits from the pan-UKBB repository (<https://pan.ukbb.broadinstitute.org/>)."

Reviewer #2 (Remarks to the Author):

D'Antonio et al describe SNP associations of gene expression (eQTLs) in heart tissues at different development stages or from different sites, and compared tissue/cell-specific eQTLs with snATAC-Seq signals. The authors colocalize eQTLs with GWAS candidates of cardiac traits and disease and hypothesize a few eQTLs as causal variants underlying these GWAS signals. The results suggest GWAS candidates of cardiac traits are likely caused by development stage- and/or tissue-specific eSNPs. The study has a potential to be a useful resource for studying genetics of cardiac disease. My main concern is that the authors proposed many causal eQTLs for GWAS candidates of cardiac traits and disease but have not done any experiments to validate the causal effects beyond the colocalization.

Here are some questions of technical details:

- 1) For expression traits with multiple eQTLs, what are the criteria for identifying additional associations?
- 2) Page 16, the formula for detecting cell type, tissue or stage specific does not seem right, XijZij may be better XijZik.
- 3). For each expression-SNP pair, how many interactions are tested? Are the same criteria used for identifying interaction signals (stage-, tissue-, and cell type-specific associations)?

Reviewer #3 (Remarks to the Author):

The manuscript by D'Antonio et al. describes a comprehensive eQTL analysis of samples relevant for the cardiovascular system. The authors re-analyze large numbers of published RNA-seq samples. The key contribution is the systematic analysis of interaction effects for different contexts, such as developmental stage, organ, tissue and cell type. The paper is well written and addresses an important problem by providing results with the potential to improve the follow up analysis of GWAS hits.

Major comments:

- 1) My main concern is that the manuscript is not always clearly distinguishing between eQTL that can only be detected when the context information is available and eQTL that can be detected using 'traditional' separate eQTL analysis of individual developmental stages and tissues. It is reasonable to assume that there will be context specific eQTL, but as it stands now there is no clear distinction

between those eQTL that can only be found when considering the context information and those that can be found by other approaches as well. It would be important to quantify this and then show the added value of these context specific eQTL in terms of how they improve the downstream analyses, such as overlap with GWAS.

2) While the statistical approaches overall seem to be appropriate, the methods are sometimes not described with sufficient detail (see the following specific comments) to be able to fully judge all results.

Specific comments:

3) What makes it sometimes a bit difficult to follow this paper is a lack of clarity as to which set of eQTL results (global or 'spatiotemporally regulated') is being used in the different analyses. Please specify this explicitly for each analysis.

4) In the GWAS analysis the authors should demonstrate the added value of using their 'spatiotemporally regulated' eQTL instead of 'regular' eQTL from GTEx (for the tissues considered here) for the interpretation of GWAS loci. How many loci are there, for which only the 'spatiotemporally regulated' eQTL colocalize?

5) The enrichment of GWAS SNP among 'spatiotemporally regulated' eQTL is interesting. The authors distinguish enrichment for the different interacting variables (contexts). The way this is currently described (e.g. 'Pulse pressure was enriched for fetal-like iPSC-CVPC-, arteria-, aorta-, smooth muscle-, endocardial- and immune- eQTLs') implies, that these eQTL are specific to the respective context. This is however not really the case, as the interaction term would also be significant if the eQTL was active in all contexts but the one tested for. Can the authors dissect the context specific contributions, as this would be particularly interesting for understanding the disease processes.

6) In the fine mapping analysis the authors suggest that fine mapping 'reduces the number of candidate causal variants to only a handful in the majority of loci and provides a spatiotemporal molecular mechanism underpinning the association between genetic variation and cardiac traits.' Again it would be important to (systematically) show here the added value of using their 'spatiotemporally regulated' eQTL instead of 'regular' eQTL from GTEx (for the tissues considered here). How much smaller do the credible sets become and how many more loci can be resolved?

7) Is the definition of the credible sets based on colocalization a standard approach? It seems that the 95% credible sets are usually defined based only on the GWAS signal. How do the results based on these two definitions compare to each other?

8) Observing a mismatch between the GWAS lead SNPs and the SNPs with highest coloc posterior might also indicate that the trait and gene expression are actually driven by different causal variants. The authors should also discuss this possibility in the light of this preprint:
<https://www.biorxiv.org/content/10.1101/2022.05.07.491045v1>

9) The authors report that SNPs that are eQTL for multiple genes are more likely to be 'spatiotemporally regulated'. It is unclear which set of eQTL results is used for the initial analysis to identify the SNPs that are eQTL for multiple genes. In addition, please clarify how the statistical tests supporting this statement reported for stage, organ, tissue and cell type are computed.

10) The comparison of eQTL for sense and corresponding antisense transcripts is quite interesting. Please add more details: how many pairs are being tested? are the effect sizes correlated or anti-correlated? Are there any enrichments for regulatory elements overlapping with these SNPs that could explain these effects?

11) Another interesting aspect concerns the pairs of genes and antisense transcripts that do not share eQTL. Is there any difference in terms of annotations between genes that share eQTL with their antisense transcript and those that do not?

12) In general the first section on annotation and comparison is not so novel and rather a QC and comparison to previous findings. As such, it may be moved into a supplementary section and only discussed very briefly in the results?

13) Additional validation of the cell type specific eQTL should be shown for the cardiomyocytes by comparison of overlap and correlation of effect sizes from the GTEx iQTL analysis (Kim-Hellmuth Science 2020).

14) The conclusion that "Different mechanisms underlie eQTLs for genes and isoforms ,, may be a bit overstated, as 56% of genes have actually a shared underlying causal variant.

15) Annotation of isoform and gene level eQTL could be a bit more fine grained. Please also compare to the findings in GTEx or the Geuvadis paper.

16) In the global eQTL analysis it is not clearly stated whether this has been done for each tissue separately or using the joint model.

17) Using 285 PEER factors seems very high. In the literature smaller numbers are typically used. It would be important to check if these factors still capture meaningful variation in the data and to make sure that the fitting actually converged.

18) Many different eQTL pipelines exist but it seems that a consensus pipeline is slowly emerging. The authors used eigenMT for gene wise multiple testing adjustment. It would be important to justify why and potentially also to compare to the GTEx pipeline (fastqtl).

19) Currently, there is no information on how many repeated measurements (same person, multiple tissues) are present in the data and how these are treated in the linear mixed model. I would assume that this is encoded in the kinship matrix. Please clarify.

20) In the results the interaction analysis of stage, organ or tissue cites ref 46 (Casale, F.P., Rakitsch, B., Lippert, C. & Stegle, O. Efficient set tests for the genetic analysis of correlated traits. Nat Methods 12, 755-8 (2015)). This paper describes a multi trait set test but no interaction analysis. Please provide more details on how this test is performed exactly.

21) The methods on detecting stage, organ, tissue or cell type specific eQTLs need to be clarified more. From the results section, it seems that separate linear models ($\text{expr} \sim \text{genotype}$) are computed for groups of samples depending on tissue or cell type proportion. How are these then compared according to which criteria? Please add this to the methods.

22) Please clarify for Fig 1E whether the test is actually contrasting SNPs that are eQTL for genes vs SNPs that are eQTL for isoforms or if the results show enrichment of SNP that are eQTL for genes vs SNPs that are not eQTL with positive sign and eQTL for isoforms vs SNPs that are not eQTL with negative sign.

23) The abbreviation PPA for reporting the results of the coloc analysis is only introduced in the GWAS section but used already before.

24) It is quite surprising to find such a large difference between the number of colocalized eQTL for the different traits. Is there any difference between these GWAS studies (sample size, number of cases, number of GWAS loci)?

RESPONSE TO REVIEWER COMMENTS

Reviewer #1 (Remarks to the Author):

The manuscript “Fine mapping spatiotemporal mechanisms of genetic variants underlying cardiac traits and disease” by D’Antonio et al., characterizes for the first time genetic regulation of gene and isoform expression (eQTLs) in fetal cardiac tissues versus adult heart and artery tissues. They identify fetal, adult and cell type specific eQTLs in these tissues and use the eQTLs to investigate underlying causal regulatory mechanisms for genomic loci associated with five cardiac traits and diseases. To do this, the authors combined RNA-seq data from 966 fetal-like and adult cardiac tissues with whole genome sequencing of the donors’ DNA from two studies (iPSCORE and GTE_x) and used a linear regression mixed effects model to compute eQTLs. The fetal like tissues are induced pluripotent stem cell (iPSC) differentiated to cardiovascular precursor cells (iPSC-CVPCs), including cardiomyocytes and epicardium-derived cells. To identify eQTLs specific to early developmental stages versus adult cardiac (atrium, ventricle, aorta and coronary artery) tissues, or that are tissue or cell type specific the authors added an interaction term between genotype and each developmental stage, tissue or cell type (estimated with deconvolution analysis) to the regression model. The authors investigated the sharing of regulation between genes and isoforms and found that adult eQTLs are more likely to affect multiple genes than fetal eQTLs. They further performed colocalization analysis between known GWAS loci for five cardiac traits and co-occurring cardiac eQTLs and fine-mapping analysis which led to identification of eQTLs and 99% credible sets of variants likely to be causal to 210 GWAS loci. This work proposed fetal-, adult tissue-, and cell type-specific eQTLs that may contribute to cardiac traits, such as fetal- and smooth muscle-specific eQTLs for pulse pressure or cardiac muscle- eQTLs and atrial fibrillation.

While this is a commendable and important effort, I have several major comments on the methodologies chosen for the analyses presented in the study that may not be optimal for comparison of genetic regulation across different tissues and developmental stages, or for dealing with allelic heterogeneity in complex trait loci, that may affect the results and conclusions of this work.

Major comments:

1. The authors combined 966 RNA-seq samples and WGS data from 180 fetal-like iPSC-CVPC samples from 139 subjects from the iPSCORE Collection and 786 adult cardiac samples from 2 heart and 2 artery tissues from 352 GTE_x subjects, and performed eQTL analysis on the combined dataset using a mixed effects model to account for overlap in donors between different subsets of tissues. Tissue, organ, developmental stage and cell type specific eQTLs were assessed by adding a genotype by tissue/cell type interaction term to the model. I think there are more suitable and better powered approaches for comparing genetic regulation of gene expression between tissues, which account for differences in sample size, direction of effect of the eVariants, and allelic heterogeneity in the QTL loci and across tissues, such as the multi-variate adaptive shrinkage (mash) method (Urbut et al., Nature Genetics 2019, PMID: 30478440). In the least I would recommend performing eQTL analysis on each tissue separately and comparing results to those generated based on the combined analysis.

Action: We thank the reviewer for this comment, as it made us realize the we needed to validate the enrichment of tissue-specific eQTLs and GWAS signals. We also note that Reviewer #3 has similar comments: #1 (R3C1) and #4 (R3C4).

Multiple studies have shown that eQTLs can be shared across multiple tissues¹⁻⁵, but there is no consensus on a specific approach to identify shared eQTLs versus eQTLs that are specific or associated with a specific tissue. The simplest approach, introduced by GTE_x², includes performing eQTL analyses independently in each tissue and then determining whether an eQTL is observed in one or multiple tissues. More advanced methods, such as multivariate adaptive shrinkage (mash)³, compare effect sizes of the same variants across multiple tissues. The major advantage of mash is improving the power of eQTL detection in all tissues by combining the effect sizes of the same variant across tissues⁴. As input, mash requires effect sizes (β) and their standard errors of genes obtained by conducting condition-by-condition analyses. Only genes with available effect sizes for all conditions are considered. Combined method approaches, such as the one used in this study and FastGxC⁵, distinguish between tissue-specific and tissue-shared eQTLs using linear mixed models. These methods are performed in two steps: in the first-step a combined eQTL analysis is conducted using all tissues and then in the second-step context-specific eQTLs are identified using an interaction test between genotype and each of the contexts. Of note, while mash’s advantage is improved power of detecting eQTLs, it can also be used to identify tissue-specific eQTLs⁴; on

the contrary, the second-step in the combined eQTL methods is specifically aimed at identifying tissue-specific eQTLs. Additionally, mash can only work for categorical features (such as developmental stage, organ or tissue: each sample can be described as belonging to only one of each developmental stage, organ or tissue), but not continuous (cell type proportions) features; whereas the combined analysis approach can work for both categorical and continuous features.

To respond to the Reviewer's concern, we proceeded as follows to validate enrichment of tissue-specific eQTLs in the cardiac GWAS signals:

Performing mash

Mash requires effect sizes and their standard errors in each tissue as input. To have a direct comparison between mash and our outputs for validation, we initially performed an eQTL analysis on each tissue separately using limix and the same covariates as our combined eQTL approach (see Methods section “Validation of the enrichment of tissue-associated eQTLs for GWAS cardiac traits”). We then selected from our combined eQTL analysis the lead variant of each of the 18,030 primary and conditional gene expression eQTLs (11,692 primary and 6,338 conditional, Figure 1A), obtained their effect sizes and standard errors for each tissue from the single-tissue eQTL analyses, and used these values to perform mash with default parameters. We annotated each eQTL signal as active in a tissue if the local false sign rate (LFSR) < 0.05 (Table S10).

Comparing eQTLs obtained with different methods

Using the single-tissue eQTL analyses we found varying numbers of eGenes and eQTLs across tissues: 1) aorta: 9,812 eGenes, and 13,029 eQTL signals (3,217 conditional eQTLs); atrium: 7,148 eGenes, and 8,770 eQTL signals (1,622 conditional eQTLs); ventricle: 5,951 eGenes, and 7,353 eQTL signals (1,402 conditional eQTLs); coronary artery: 5,458 eGenes, and 6,143 eQTL signals (685 conditional eQTLs); and iPSC-CVPC: 3,393 eGenes, and 3,682 eQTL signals (289 conditional eQTLs, Table S9). The observed differences in the number of eQTLs in each tissue are likely due to power differences across the tissues from sample numbers and tissue heterogeneity.

We next examined the mash results obtained using the eQTLs (for eGenes, i.e. not for eIsoforms) from the first-step of our combined approach coupled with their effect sizes and standard errors for each tissue from the single-tissue eQTL analyses as input (Table S10). Mash improved the detection of eQTLs active in each tissue compared with the single-tissue eQTL analysis for all tissues but aorta (1.45X times for atrium, 1.63X for ventricle, 1.89X for coronary artery and 2.40X for iPSC-CVPCs, Figure S13). Interestingly, the least-powered tissues (i.e. with the smallest number of unrelated samples: iPSC-CVPCs and coronary artery) had the largest increase in the number of eQTL signals detected, whereas the tissue with the largest number of tested samples (aorta) had a comparable number of eQTLs detected with the two methods. These results demonstrate that performing one combined eQTL analysis using all tissue samples and then identifying tissue associations results in increased power, compared with performing single-tissue eQTL analyses, as was previously suggested^{1,3,4}.

Of note, neither the traditional tissue-by-tissue eQTL analyses, nor the mash approach is able to detect cell type-associations, which can only be identified by using our interaction approach (described in the Methods section “Spatiotemporal eQTL mapping”). Therefore, while mash is useful to determine the tissue context of each eQTL, it cannot fully substitute for our combined analysis approach, as it cannot be used to test cell type associations.

We have added a detailed “Supplemental Note 2” Figure S13, Table S9 and Table S10 to explain the differences between these approaches for identifying tissue-associated eQTLs.

Figure S13: Comparison of the number of tissue-associated eQTLs using two methods: single-tissue eQTLs versus mash

Figure S13: Barplots showing the number of eQTLs detected per tissue performing single-tissue eQTL analysis versus using mash on the eQTLs (for eGenes, ie, not for eIsoforms) identified in the first-step of the combined analysis (This study eQTLs) with their effect sizes and standard errors for each tissue from the single-tissue eQTL analyses as input.

Using mash results to validate enrichment of tissue-specific eQTLs and GWAS signals

We sought to validate enrichments of the tissue-specific eQTLs and GWAS signals shown in Figure 5E-G and Figure S9. Using the eQTLs detected by mash for each tissue, we performed enrichment tests across the five cardiac GWAS traits. We used varying LFSR thresholds to define whether a gene was expressed in a given tissue, and observed similar enrichments to those obtained using our combined analysis approach (Figure S14):

- Myocardial infarction does not have strong associations with any tissue;
- QRS duration does not have strong associations with any tissue;
- Atrial fibrillation is enriched for eQTLs active in atrium and, at stringent LFSR thresholds, with ventricle ($LFSR \leq 1 \times 10^{-5}$) and with iPSC-CVPC ($LFSR \leq 1 \times 10^{-10}$);
- Pulse rate is enriched for eQTLs active in atrium and ventricle at all LFSR thresholds, and with aorta at most thresholds;
- Pulse pressure is enriched for eQTLs active in aorta at all thresholds, iPSC-CVPC at stringent LFSR thresholds ($LFSR \leq 1 \times 10^{-10}$), and coronary artery at loose LFSR thresholds ($LFSR \geq 0.05$).

In our “Supplemental Note 2” we describe the above validation and have included Figure S14 and Table S10.

We have also added the following to the Results section “Cardiac traits enriched for spatiotemporal eQTLs that function in specific contexts”:

“To validate enrichments of the spatiotemporal eQTLs and GWAS signals, we determined the tissue context of each eQTL signal using an independent method, multivariate adaptive shrinkage (mash)³, and observed very similar enrichments (See Supplemental Note 2, Figure S13, Figure S14, Table S9, Table S10)”.

Additionally, we have expanded the Methods by adding the section “Validation of the enrichment of tissue-associated eQTLs for GWAS cardiac traits” to describe these analyses:

“To validate enrichment of tissue-specific eQTLs in the cardiac GWAS signals we used the multi-variate adaptive shrinkage (mash) method⁴ – see Supplemental Note 2 for details on this method. We initially performed independent tissue-specific eQTL analyses on five tissues (iPSC-CVPC, arteria, aorta, ventricle, and atrium) using a linear mixed model (limix) and the same covariates as our combined eQTL approach: sex, normalized number of RNA-seq reads, % of reads mapping to autosomes or sex chromosome, % of mitochondrial reads, 20 genotype PCs, and kinship matrix, and recalculated the number of optimal PEER factors. Gene expression was quantile-normalized across all samples within each of the five datasets, and the number of optimal PEER factors was determined using the same method described for the combined eQTL analysis: we obtained 200 random genes and performed eQTL analyses using a range of PEER factors and selected the number of PEER factors for each dataset associated with the maximum number of detected eQTLs for these 200 genes. We used the following number of PEER factors: 40 for iPSC-CVPC, 60 for atrium, 70 for ventricle, 70 for aorta, and 30 for coronary artery. The

kinship matrix was included as a random effect. Conditional eQTL analysis and FDR correction was performed using the same method as our combined eQTL approach (Table S9).

To have a direct comparison between mash and our outputs for validation, we next examined the mash results obtained using the eQTLs from the first-step of our combined approach (for eGenes, not for eIsoforms; Figure 1A) coupled with their effect sizes and standard errors for each tissue from the single-tissue eQTL analyses as input – see Supplemental Note for details. We ran mash with default parameters and obtained the local false sign rate (LFSR)^{3,4}. We annotated each eQTL signal as associated with a tissue if the LFSR < 0.05 (Table S10). We also filtered at more stringent thresholds (up to 1×10^{-30}). At the least stringent LFSR threshold (LFSR = 0.05), we observed < 11,500 eQTLs in each of the four adult tissues and 8,856 in iPSC-CVPCs (Figure S13). Using the eQTLs detected by mash for each tissue, we performed enrichment tests across the five cardiac GWAS traits, as described in the previous section (Figure S14).”

2. Given that samples are from two different studies, it would be good to see how the samples from the two studies cluster in PCA space based on their gene expression profiles across all genes before and after PEER factor correction to make sure that batch effects are properly corrected for between the iPSCORE and GTEx samples. The authors chose to include 285 PEERs (inferred covariates of non-specific gene expression variation) in the eQTL regression model based on the largest eGene discovery of a test set of 200 genes as a function of number of PEERs (up to 300 PEERs). This number (N=285) seems very large for the given sample size. Can the authors please show the plot of number of identified eQTLs or eGenes as a function of number of PEERS to support this choice.

Action: We thank the reviewer for these suggestions and have updated the manuscript to better describe how we selected 285 PEER factors and show that the PEER factors account for batch effects.

With regard to the number of selected PEER factors, we divided the 19,986 expressed genes into ten deciles based on their average TPM, and selected 20 random genes from each decile. We next performed eQTL analysis on each of the resulting 200 genes using 5, 10, 20 and increments of 20 PEER factors between 20 and 300. Having identified the maximum number of eQTLs using 280 PEER factors, we performed a second eQTL analysis on the same 200 genes using 270, 275, 285 and 290 PEER factors. We chose 285 PEER factors because it resulted in maximal eQTL discovery and any additional PEERs may decrease the power, as the reviewer has stated. While 285 may seem high, anything lower than 100 was not optimal and resulted in many fewer eQTLs (Figure S18A). We and others have used this method for selecting the number of PEER factors in previously published studies^{2,6,7}.

To confirm that the PEER factors captured batch effects between the iPSCORE and GTEx samples, we plotted the samples in the PEER1 vs PEER2 space and observed that the samples were distinctly separated (Figure S18B). We also examined the correlation between the first 10 PEER factors and other covariates, and observed that the PEER factors were distinctly correlated with iPSCORE and GTEx datasets, as well as other technical covariates (Figure S18C). These analyses show that batch effects were properly corrected for between the iPSCORE and GTEx datasets.

In our updated manuscript, we have added Figure S18 and rewritten the Methods section “Covariates for eQTL mapping” to describe how we selected 285 PEER factors and how PEER factors account for batch effects:

“We sought to determine the optimal number of PEER factors⁸ for obtaining the maximum number of eQTLs. We initially calculated PEER factors on the 10,000 expressed genes with the largest variance across all samples. To limit biases due to the expression levels of each gene, we divided the 19,986 expressed genes into ten deciles based on their average TPM, and selected 20 random genes for each decile, for a total of 200 genes. We next performed eQTL analysis on each of the 200 genes using the following numbers of PEER factors: 5, 10, 20 and increments of 20 between 20 and 300. These eQTL analyses of the 200 genes to determine the optimal number of PEER factors was performed using the same covariates as for the combined eQTL analysis: 1) sex; 2) normalized number of RNA-seq reads; 3) % of reads mapping to autosomes or sex chromosome; 4) % of mitochondrial reads; 5) 20 genotype principal components to account for global ancestry; 6) the selected number of PEER factors to account for transcriptome variability; and 7) kinship matrix. We observed the maximum number of eQTLs using 280 PEER factors. To further refine the optimal number of PEER factors, we performed a second eQTL analysis using 270, 275, 285 and 290 PEER factors, which determined that 285 PEER factors were optimal (Figure S18A). For isoform eQTLs, we used 80 PEER factors as a compromise between obtaining the largest number of eQTLs and computational burden. To confirm that the PEER factors were accounting for batch effects between the iPSCORE and GTEx samples, we examined the distribution of the samples in PEER space and the correlation of the top 10 PEER factors with other covariates (Figure S18B). We also examined the correlation between the first 10 PEER factors and other covariates and observed that the PEER factors were distinctly correlated with iPSCORE and GTEx datasets, as well as other technical

covariates (Figure S18C). These analyses show that batch effects were properly corrected for between the iPSCORE and GTEx datasets”.

Figure S18: Assessing the optimal number of PEER factors

Figure S18: (A) Proportion of eGenes (Y axis) using different combinations of PEER factors (X axis: 5-300 PEER factors). The maximum number of eGenes is reached at 285 PEER factors. (B) Plot showing the PEER factor coordinates (PEER 1 and PEER 2) of each of the 966 samples: the samples cluster by tissue, as expected. (C) Corplot showing the correlation between all the covariates used to detect eQTLs, including PEER factors 1-10.

- Given that samples come from two different studies, it would be good to inspect the ancestral distribution of the iPSCORE samples compared to the GTEx samples based on the top genotype principal components (PCs) (e.g. plot PC1 vs PC2, PC1 vs PC3, PC2 vs PC3 for all samples colorcoding points by study). To correct for potential differences in population background between samples and studies, in particular given that the samples are from two different studies (iPSCORE and GTEx), the authors included the top 20 genotype PCs in their eQTL regression model. A smaller number of genotype PCs might suffice (as adding more covariates to the model might lead to loss of power). I would recommend inspecting/plotting the variance explained by each PC (eigenvalue divided by the sum of all eigenvalues across all PCs, where n=total number of samples), and choosing the number of genotype

PCs based on those components with the highest relative variance explained, that fall before the infection point. Can the authors present...

Action: We thank the Reviewer for this comment and agree that the ancestral distributions of iPSCORE and GTEx samples based on the genotype PCs is relevant for understanding if these two different studies can be combined and analyzed together. We have recently published a study⁹ that used the same 966 iPSCORE and GTEx samples and showed the sex and ancestral distributions of the samples. Specifically, in **Figure S1B** (shown below), we showed each individual plotted on the genotype PC space.

Recent published studies have used different numbers of PCs for their analyses: Liu et al. used only 4, 10 and 12 PCs¹⁰, whereas C Johnson et al. used 20¹¹ and Sinnott-Armstrong et al. used both 20 and 40¹². Moreover, studies have shown that the number of genotype PCs used in a genetic association study does not have major effects on power: Mogil et al. tested a range of genotype PCs (0,3,5,10) and found that the number of eQTL discovered was nearly the same¹³; and, The Pan UK BioBank Methodologies (<https://pan.ukbb.broadinstitute.org/docs/qc/index.html>) confirmed that including different numbers of PCs in their GWAS does not change ancestry assignments. Based on these published studies, we decided to use the top 20 genotype PCs in our regression model.

In our updated manuscript, we have inserted an explanation about the genotype PCA analysis in the Methods section “Genotype Principal Component Analysis”

“We previously performed principal component analysis (PCA) on WGS variants to determine the global ancestry of each individual in this study⁹. Briefly, we used the genotypes of 1,634,010 SNPs that had allele frequencies between 30% and 60% in the 1000 Genomes Phase 3 Project and genotyped in both iPSCORE and GTEx. We merged the VCF files from 1000 Genomes, iPSCORE, and GTEx, and performed PCA using the *pca* function in *plink 1.90b3x*¹⁴. We showed that iPSCORE and GTEx individuals overlap their associated 1000 Genome superpopulations, and that iPSCORE includes more individuals of East Asian ancestry than GTEx. The top 20 genotype principal components used to account for global ancestry are available for all 491 individuals (139 iPSCORE and 352 GTEx) in the expanded **Table S1** in Figshare”.

Figure S1. Sex and ancestry distributions⁹

(A) Sex distribution between iPSCORE individuals (blue) and GTEx individuals (maroon).

(B) PCA showing the ancestry of all subjects. Colored circles represent ancestry of individuals from the 1000 Genomes Project. GTEx and iPSCORE individuals are represented by pink and blue “X”, respectively.

- The authors used a set of 1,634,010 SNPs with an allele frequency between 0.3-0.6 in the 1000 Genomes Phase 3 project and genotyped in both GTEx and iPSCORE to generate genotype PCs. Did the authors perform LD pruning between these variants to ensure that the SNPs are genetically independent to each other? This usually yields on the order of 10^5 SNPs in WGS data. This is important to do prior to computing genotype PCs to properly represent the genetic variation between population backgrounds.

Action: We recognize that LD pruning is a standard approach. In a previous publication¹⁵, we used an approach similar to the one described in this study. Per the reviewer’s suggestion, we re-calculated the PCs with LD-pruned variants (using 200 kb sliding windows of r^2 at 0.2 with step size of 100 SNPs), which resulted in 95,853 SNPs, and compared these PCs with the PCs used in the current manuscript. As shown below, we found that the first 4 PCs had near perfect correlation ($R \sim 1.00$). We also examined the distribution of the samples in the genotype PCA space with and without LD-pruning and found

that the placements of the samples were nearly identical (see Figure below). Based on these results, we are confident that our analyses properly accounted for global ancestry between the samples.

5. Can the authors specify in the Methods section what method was used to reconstruct and quantify the isoforms? Was it using RSEM or another method?

Action: We apologize for the confusion here. We had referenced our recent study⁹, which describes the processing steps in detail. However, we agree that these details should be described in the Methods for the current manuscript as well.

In our updated manuscript, we have inserted the following in the Methods section “RNA-seq data”:

“RNA-seq data for all 966 cardiac samples (180 iPSC-CVPC and 786 adult) was obtained from dbGaP (phs000924 and phs000424) and processed as previously described in detail^{16,17}. Briefly, FASTQ files were aligned to the hg19 reference genome using STAR 2.5.0a¹⁸ and Gencode V.34lift37¹⁹ with parameters *outFilterMultimapNmax 20, -outFilterMismatchNmax 999, -alignIntronMin 20, -alignIntronMax 1000000, -alignMates-GapMax 1000000*. We sorted the BAM files using Sambamba 0.6.7²⁰ and marked duplicates using biobambam2 (2.0.95) *bammarkduplicates*²¹. To quantify TPM gene expression and relative isoform abundance, we used RSEM²² with options *rsem-calculate-expression—bam—num-threads 16—no-bam-output—seed 3272015—estimate-rspd—paired-end—forward-prob 0*. Using this pipeline, we determined the expression of 62,492 genes and their corresponding 229,835 isoforms. Only 19,586 autosomal genes with TPM ≥ 1 in at least 10% of the 966 samples were considered as expressed and used for eQTL analysis. Likewise, 37,032 isoforms (TPM ≥ 1 and usage $>10\%$ in at least 10% of the 966 samples) from 10,337 expressed genes were used for isoform eQTL analysis”.

6. What normalization was performed on the number of RNA-seq reads per gene or isoform? For example, in GTEx v8 TMM normalization with EdgeR is applied followed by standard normalization.

Action: We apologize for not including this information. We performed quantile normalization across the samples per gene and isoform.

This information is now inserted in the Methods section “Data processing”:

“To normalize gene and isoform expression and usage across samples, we performed quantile-normalization using *normalize.quantiles* (preprocessCore) and *qnorm* functions in R, in order to obtain mean expression = 0 and standard deviation = 1, as we previously described^{9,23}”.

7. Did the authors include sex in eQTL regression model? It seems that they added sex only for estimating the optimal number of PEERs in the eQTL analysis. Can the authors clarify this point.

Action: Sex was included as a covariate in eQTL mapping, but, in our original submission, we did not describe the covariates used to determine the optimal number of PEER factors. Every eQTL analysis described in the manuscript used the same covariates.

We have added the following in the Methods section “Covariates for eQTL mapping”:

“These eQTL analyses of the 200 genes to determine the optimal number of PEER factors was performed using the same covariates as for the combined eQTL analysis: 1) sex; 2) normalized number of RNA-seq reads; 3) % of reads mapping to autosomes or sex chromosome; 4) % of mitochondrial reads; 5) 20 genotype principal components to account for global ancestry; 6) the selected number of PEER factors to account for transcriptome variability; and 7) kinship matrix”.

8. To test enrichment of eQTLs in different functional genomic annotations, the authors inspected the overlap of the lead variant per eGene or eIsoform different genomic categories, including intergenic regions, introns, promoters, UTRs, splice sites, and exons. However, since the lead variant might not be the causal variant but rather tagging the causal variant via LD, I would recommend doing this analysis using all the significant variant-gene pairs per eGene/eIsoform or at least include the variants in LD (e.g., $r^2 > 0.8$) with the lead variant. Also, can the authors more clearly explain the enrichment test that was performed here. Another method that could be fitting to apply here is TORUS (<https://github.com/xqwen/torus>) that is a logistic regression model that estimates enrichment of eQTL variants in functional annotations.

Action: We agree with the Reviewer that lead variants may not be the causal variant at each locus. Therefore, for each eGene or eIsoform, we performed fine mapping using coloc, obtained the fine-mapped variants with PPA > 0.01 and used them for the enrichment analysis between eGene and eIsoform signals. We apologize that this was not clear in the original manuscript.

To improve clarity, we describe the definition of 99% credible sets in the Methods section “Testing enrichment of eQTL signals in different genomic annotations”:

“To test the enrichment of eQTLs in intergenic regions, introns, promoters, UTRs, splice donor sites, splice acceptor sites and exons, we obtained fine-mapped variants for all eQTL signals for both eGenes and eIsoforms.

Determining the 99% credible sets for each eQTL signal: To define a credible set of candidate causal variants for each eQTL association, we performed genetic fine mapping using the *finemap.abf* function in the *coloc* R package²⁴. This Bayesian method converts p-values of all the variants tested for a specific gene to posterior probabilities of association (PPA).

Testing enrichment of eQTL signals in different genomic annotations: We obtained fine-mapped variants for all eQTL signals for both eGenes and eIsoforms with PPA > 0.01. We determined the overlap of each fine mapped variant and each genomic annotation using *bedtools intersect* and compared the proportions of variants overlapping each annotation between eGenes and eIsoforms using Fisher’s Exact Test. Promoters were defined as the 2,000 bp upstream of the transcription start site. Exons and UTRs were obtained from *Gencode V.34lift37*. For the intronic variants, we calculated their distance from the closest exon to determine their overlap with splice sites”.

9. Can the authors please clarify how they determined developmental stage-, organ-, tissue-, and cell type-specific eQTLs. At first in the Methods section, it is stated that a linear mixed model with an interaction term is used between genotype and each stage/organ/tissue/cell type to detect association between eQTLs and stage/organ/tissue/cell type. However, afterwards, the authors describe another analysis performed to detect developmental stage, tissue and cell type-specific eQTLs, i.e. that samples were divided into two groups (one type vs. all others) and eQTL association was tested between each of these two groups and genotype, using linear regression analysis. For cell type-specific eQTLs, they obtained the samples in the top and bottom quartiles based on to their deconvoluted cell type

populations for each cell type. Can the authors explain why they performed these two approaches and how they compared between the two.

Action: We apologize for the lack of clarity in this analysis. As Reviewer #3 pointed out (R3C5), an interaction can still be considered significant if the eQTL is active in all other samples but the context being tested (for example: if we are testing the interaction with “left ventricle”, the eQTLs that are significant in all the other tissues but left ventricle can have a significant p-value). In our original submission, we had added a second filtering step to address this potential issue (referred to by the Reviewer as “samples were divided into two groups”), but we realize that it was not thoroughly described. In this second filtering step, for binary contexts (developmental stage, tissue and organ), we divided all samples into two sets corresponding to the two binary values (for example: 1) left ventricle; 2) all the other samples). For cell types, we divided all samples into quartiles and analyzed the top and bottom quartiles. We performed an additional eQTL analysis on each of the two sets and classified the interactions based on the following criteria:

- a. If the eQTL is significant only for the tested context (for example: left ventricle), the eQTL signal is considered context-specific (left ventricle-specific);
- b. If the eQTL is significant only for all the other samples, the eQTL is not considered context-specific and thus not used in downstream analysis;
- c. If the eQTL is significant for both sets but its effect size is stronger (larger absolute value) in the tested context, the eQTL is context-associated (left ventricle-associated);
- d. If the eQTL is significant for both sets but its effect size is stronger (larger absolute value) in all the other samples, the eQTL is not considered as context-specific or context-associated.

In Supplementary Figures, we show examples of eQTLs with significant interactions with stage (Figure S1A), tissue (Figure S2), and cell type (Figure S3).”

We have updated the Methods section “Spatiotemporal eQTL mapping” to describe the second filtering step more clearly:

“Additional filtering steps to determine context-specific and context-associated eQTLs: While the interaction LMM described above identifies context-specific eQTLs, an interaction would also be considered significant if the eQTL is active in all other samples but the context being tested (for example: if we are testing the interaction with “left ventricle”, the eQTLs that are significant in all the other tissues but left ventricle would have a significant p-value). To avoid counting these associations as “context-specific” we included the following filtering step for all spatiotemporal eQTLs (i.e., those with a significant interaction term in the above analysis). For binary contexts (developmental stage, tissue and organ), we divided all samples into two sets corresponding to the two binary values (for example: 1) left ventricle; 2) all the other samples). For cell types, we divided all samples into quartiles and analyzed the top and bottom quartiles. We performed an additional standard eQTL analysis on each of the two sets. P-values were corrected across all interactions using Benjamini-Hochberg’s method. Associations with adjusted p-value < 0.05 were considered significant. We considered the following scenarios:

1. eQTL is significant only for the tested context (for example: left ventricle): the eQTL signal is considered as context-specific (left ventricle-specific);
2. eQTL is significant only for “all the other samples”: the eQTL is not considered as context-specific;
3. eQTL is significant for both sets but its effect size is stronger (larger absolute value) in the tested context: the eQTL is context-associated (left ventricle-associated);
4. eQTL is significant for both sets but its effect size is stronger (larger absolute value) in “all the other samples”: the eQTL is not considered as context-specific or context-associated”.

10. Are there multiple developmental stages in the fetal RNA-seq dataset? It could be helpful for the reader to include a table that provides a breakdown of the various types of tissues, developmental stages, study name and number of samples per type.

Action: Thank you for this question, we agree that this point was not described with enough detail.

We have updated the Methods section “Data processing” to clarify this point:

“IPSCORE iPSC-CVPCs were derived using a standardized protocol followed by lactate purification^{16,25,26}. All 180 samples were collected at day 25 differentiation and represent fetal cardiac cells, as we and others have previously shown^{16,25-27}”.

11. It would be interesting to compare the pathways in which the target genes of cell type specific eQTLs are enriched relative to cell type-shared eQTLs or eQTLs that are specific to early development vs. adult tissue-specific eQTLs.

Action: This is an interesting suggestion. To respond to the Reviewer’s comment, we performed Gene Ontology enrichment analyses on eGenes associated with each developmental stage, organ, tissue and cell type using a Fisher’s Exact Test per the reviewer’s suggestion and observed no enrichment (after multiple test correction with Benjamini-Hochberg’s method) between spatiotemporal eGenes versus all the other eGenes. We believe that the lack of enrichment may be due to the fact that eQTLs are in general neutral, otherwise they would be negatively selected (if deleterious) or fixed (if they provide fitness advantages) in the population. Therefore, it would be expected that no pathways should be enriched for tissue- or developmental stage-specific eQTLs.

12. The authors found that that 47 pairs of genes and their antisense RNA share the same eQTL signal. I assume this means that the eQTLs acts in the same direction on both the gene and its antisense (which could lead to opposite regulation of the gene). Can the authors comment on whether they found separate eQTLs acting on a gene and its antisense gene in opposite directions, which would lead to accentuation of the gene’s eQTL effect.

Action: The Reviewer has made an excellent point. Per the Reviewer’s suggestion, we examined the effect sizes of each pair of genes and their antisense RNA. Of the 47 pairs, we found that 12 acted in opposite directions (Figure S7). We then investigated each of these 12 pairs (Figure S8) and found literature evidence of a negative association between the expression of a gene and its associated antisense RNA in two cases (*TRAPPC12/TRAPPC12-AS1* and *ABCF2/ABCF2-AS1*). These findings suggest that the eQTLs may result in stronger effects, because they are simultaneously associated with changes in the expression of an antisense RNA and changes in the opposite direction for its target gene.

We have added Supplemental Note 1, Figure S6, Figure S7 and Figure S8:

“Of the 47 pairs that shared the same eQTL signal, we found 12 whose eQTL acted in opposite directions on the expression of the eGene and its antisense RNA (Figure S7, Figure S8). We next tested whether the eQTLs for correlated and anticorrelated eGene/antisense pairs were associated with different types of regulatory elements, and found that correlated pairs were enriched for intergenic regions while anti-correlated pairs were enriched for splice acceptor and donor sites and exonic regions (Figure S6B). The 12 pairs with opposite effects included two genes whose antisense RNA is known to be involved in the negative regulation of gene expression: *TRAPPC12/TRAPPC12-AS1* and *ABCF2/ABCF2-AS1*^{28,29}. These results suggest that, for these 12 pairs, the eQTLs may increase the regulatory effect of the antisense RNA, whose change in expression (increase/decrease) results in the opposite expression change (decrease/increase) on their target gene.”

13. The authors performed colocalization analysis between GWAS loci associated with five heart related complex diseases or traits and overlapping eQTLs in the cardiac tissues and cell types. They used the coloc method that assumes one causal variant per locus. Since we know that there is allelic heterogeneity in many GWAS and eQTL loci (as the authors showed in Figure 1a), it would be more powerful to use other Bayesian colocalization methods, such as eCAVIAR (Hormozdiari et al., AJHG 2016) that can consider multiple causal variants in a locus, or CAFEH (Arvanitis et al., AJHG 2022, PMID: 35085493) that also accounts for tissue specificity of eQTL effects that may be well suited for this study.

Action: To circumvent the problem of multiple causal variants, we have also performed conditional eQTL analysis, which identifies additional independent eQTLs in the locus (up to five conditional signals/eGenes, Figure 1A).

We have updated the Results to describe the presence of multiple causal variants for each eGene: “An eGene may have multiple different eQTL signals, each associated with a different underlying causal variant, which can be detected by regressing out the genotype of the lead variant of the primary and subsequent conditional eQTLs. We obtained conditional eQTLs for 4,394 eGenes (37.6% of all eGenes), including 1,315 with two conditional eQTLs, 395 with three, 160 with four and 74 with five (Figure 1A), which is in line with what has recently been reported by GTEX⁷.”

We also update the section “Colocalization identifies potential molecular mechanisms underlying GWAS signals”: “We focused on 1,444 eGenes and 919 eIsoforms that overlapped or were in close proximity (<500 kb) with genome wide-significant GWAS signals; and to account for the potential presence of multiple independent causal variants at the same locus, performed colocalization using both primary and conditional eQTL signals. We found that the eQTLs for 206 eGenes and 125 eIsoforms (including 296 primary and 35 conditional eQTLs) colocalized with high PPA with at least one GWAS signal (PPA ≥ 0.8 , Figure 4, Table S7)”.

Minor comments:

1. Can the authors provide more details on how many variants were found common between the two studies, and how many remained after quality control (QC).

Action: We obtained all variants with MAF >1% in each study, that were in Hardy-Weinberg equilibrium in GTEx ($p > 1 \times 10^{-6}$) and that were within 500 kb of any expressed gene. This resulted in 4,962,200 total tested variants for 19,586 genes.

We have added the following text in the Methods sections “WGS data” to specify this information:

“VCF files from WGS data were obtained from dbGaP (phs001325 and phs000424). We retained variants with MAF >1% in both studies, that were in Hardy-Weinberg equilibrium ($p > 10^{-6}$) and that were within 500 kb of any expressed gene. Specifically, we expanded the coordinates of each of the 19,586 expressed genes (500 kb upstream and downstream) and extracted all variants in these regions using *bcftools view* with parameters *--f PASS -q 0.01:minor*. Next, we merged the resulting VCF files (*bcftools merge -m none*), normalized indels and split multiallelic variants (*bcftools norm -m-*) and removed variants genotyped in fewer than 99% of samples (*bcftools filter -i 'F_PASS(GT!="mis") > 0.99*). Finally, we converted the resulting VCF files to text using *bcftools query* and converted the genotypes from character strings (0/0, 0/1 and 1/1) to numeric (0, 0.5 and 1, respectively). This resulted in 4,962,200 total variants that were in common between the two studies (iPSCORE and GTEx) and used for eQTL mapping”.

2. In the first paragraph of the Results section “By regressing out the genotype of each lead variant, we observed that, on average, each eGene had 1.54 eQTLs (range: 1-6),” for clarification, I would add the word “independent” to “1.54 independent eQTLs”.

Action: Thank you for the suggestion, we have inserted the word “independent”: “An eGene may have multiple different eQTL signals, each associated with a different underlying causal variant, which can be detected by regressing out the genotype of the lead variant of the primary and subsequent conditional eQTLs. We obtained conditional eQTLs for 4,394 eGenes (37.6% of all eGenes), including 1,315 with two conditional eQTLs, 395 with three, 160 with four and 74 with five (1.54 independent eQTL signals per gene, Figure 1A), which is in line with what has recently been reported by GTEx”.

3. I think there is a typo in the sentence: “ γ_n is the effect size of the n th genotype principal component,” I think ‘n’ is supposed to be ‘m’.

Action: We thank the reviewer for their in depth reading of our manuscript. We have revised the typo to “m”.

4. In the Methods section under “Colocalization between GWAS and eQTL signals”, I would recommend the authors add the word “GWAS” before “summary statistics”: “We obtained summary statistics from five cardiac traits from the pan-UKBB repository (<https://pan.ukbb.broadinstitute.org/>).”

Action: We agree with the reviewer. We have added the word “GWAS” before “summary statistics”.

Reviewer #2 (Remarks to the Author):

D’Antonio et al describe SNP associations of gene expression (eQTLs) in heart tissues at different development stages or from different sites, and compared tissue/cell-specific eQTLs with snATAC-Seq signals. The authors colocalize eQTLs with GWAS candidates of cardiac traits and disease and hypothesize a few eQTLs as causal variants underlying these GWAS signals. The results suggest GWAS candidates of cardiac traits are likely caused by development stage- and/or tissue-specific eSNPs. The study has a potential to be a useful resource for studying genetics of cardiac disease. My main

concern is that the authors proposed many causal eQTLs for GWAS candidates of cardiac traits and disease but have not done any experiments to validate the causal effects beyond the colocalization

We agree with the reviewer that independent confirmation of the functional effects of the putative causal variants is important. In the original manuscript we determined if the causal variants identified by our colocalization analysis correspond to lead index GWAS SNPs that have previously been reported as associated with the same trait in the GWAS Catalog. We determined that 43 (~20%) of the putative causal variants had previously been described as lead index variants for the same trait and that another ~20% were in LD with such a lead variant. These data are essentially a replication of a substantial number of the putative causal variants associated with the 210 GWAS loci that we identified. To further address the reviewer's comment we conducted two additional analyses to characterize the functional effects of the putative causal variants:

- 1) First, we tested if the putative causal variants were enriched for being in fetal and adult heart chromatin states. Specifically, we obtained Roadmap Epigenome Data for fetal heart, adult right atrium, adult left ventricle, and adult right ventricle. We observed that variants with high PPA are more likely to reside in active chromatin states, such as enhancer regions in fetal heart ($p = 1.53 \times 10^{-29}$) and transcriptional start sites in adult heart (right atrium: $p = 1.14 \times 10^{-5}$; left ventricle: $p=1.41 \times 10^{-13}$; and right ventricle: $p=1.43 \times 10^{-7}$, Figure S16).
- 2) Second, we tested the association between the strength of the PPA for a putative causal SNP and the predicted impact of the SNP on disease using DeepSea^{30,31}. Using linear regression, we observed that SNPs with higher PPAs are more likely to impact disease compared to SNPs with lower PPAs ($p = 6.08 \times 10^{-14}$, Figure S17).

We added the following text to the Results section “Fine mapping using colocalization data identifies putative causal variants for hundreds of loci” and generated Figure S16 and Figure S17.

“To examine the functional effects of the putative causal variants in the 210 credible sets, we performed two enrichment analyses. First, we investigated the overlap of candidate causal variants with fetal and adult heart chromatin states. Specifically, we obtained Roadmap Epigenome Data for fetal heart, adult right atrium, adult left ventricle, and adult right ventricle and observed that variants with high PPA are more likely to reside in active chromatin states, such as enhancer regions in fetal heart ($p = 1.53 \times 10^{-29}$) and transcriptional start sites in adult heart (right atrium: $p = 1.14 \times 10^{-5}$; left ventricle: $p=1.41 \times 10^{-13}$; and right ventricle: $p=1.43 \times 10^{-7}$, Figure S16). Second, we tested the association between the strength of the PPA for a putative causal SNP and the predicted impact of the SNP on disease using DeepSea^{30,31}. Using linear regression, we observed that SNPs with higher PPAs are more likely to impact disease compared to SNPs with lower PPAs ($p = 6.08 \times 10^{-14}$, Figure S17). These results show that the variants with high PPA are more likely to have functional impacts, confirming their putative causal role for the associated cardiac traits”.

We also added the details of these analyses in the Methods sections “Colocalization between GWAS and eQTL signals”:
“**Enrichment of GWAS causal variants in heart chromatin states:** Across all 210 fine-mapped GWAS loci, we obtained 2,157 SNPs with $PPA \geq 0.01$ and tested their enrichment in Roadmap Chromatin States for fetal heart, adult right atrium, adult left ventricle, and adult right ventricle using a linear regression model with the *lm* function in R where binary state annotation was modeled as a function of SNP PPA. Enrichments with P-values < 0.05 were considered significant.
Prediction of GWAS causal variants for impacting disease: We provided a VCF of all 2,157 SNPs with $PPA \geq 0.01$ in the 99% credible sets of the 210 fine-mapped GWAS loci as input into DeepSea^{30,31}, which is a deep-learning based framework that predicts the chromatin effects of each SNP. One of the outputs of DeepSea (version “Beluga”) is the disease impact score for each SNP, which was computed using a logistic regression model that prioritizes candidate disease-associated SNPs based on their predicted effects on transcriptional or post-transcriptional regulation. To determine whether GWAS candidate causal eQTLs are likely to impact disease, we used a linear regression model to test the association between SNP PPA and disease impact score”.

Here are some questions of technical details:

1. For expression traits with multiple eQTLs, what are the criteria for identifying additional associations?

Action: By “multiple eQTLs”, we believe the reviewer was referring to conditional eQTLs. To identify these eQTLs, we performed a stepwise regression analysis where the genotype of lead variant was included as a covariate. We repeated this analysis to identify up to 5 independent associations. After multiple test correction across all independent associations using eigenMT (all variants tested for the same gene) and Benjamini-Hochberg's correction (for all tested genes), we considered

the conditional eQTL to be significant if $q\text{-value} < 0.05$. This pipeline is similar to the methods described in previous studies^{32,33}.

We have updated the Methods section “Conditional eQTL signals”:

“To identify additional independent eQTL associations for a gene (i.e. conditional eQTLs), we performed a stepwise regression analysis in which we re-performed eQTL analysis and included the genotype of the lead eQTL as a covariate. We repeated the analysis to discover up to five conditional associations. For each iteration, we performed the two-step procedure described above and considered conditional eQTLs with $q\text{-values} < 0.05$ as significant”.

1. Page 16, the formula for detecting cell type, tissue or stage specific does not seem right, $X_{ij}Z_{ij}$ may be better $X_{ij}Z_{ik}$.

Action: We thank the reviewer for their in depth reading of our manuscript and apologize for this typo. We have updated this annotation accordingly.

2. For each expression-SNP pair, how many interactions are tested? Are the same criteria used for identifying interaction signals (stage-, tissue-, and cell type-specific associations)?

Action: We tested 16 interactions, one for each context: two organs (arteria and heart), four tissues (atrium, ventricle, aorta, and coronary artery), eight cell types (cardiac muscle, smooth muscle, endothelial, fibroblast, immune, cardiac neuron, endocardial, and myofibroblast), and two developmental stages (fetal-like iPSC-CVPC and adult). This information is described in **Figure 2A,B** as well as **Table S4** in Column 6.

We have inserted the following text in the Methods section “Spatiotemporal eQTL mapping” to make this information clearer:

“To detect eQTL associations with cell types, tissue or developmental stage, we obtained the lead variant for each significant eQTL from the combined analysis for each eGene and eIsoform and used the same linear mixed model described above and included an interaction between genotype and context. We considered a total of 16 different contexts: two organs (arteria and heart), four tissues (atrium, ventricle, aorta, and coronary artery), eight cell types (cardiac muscle, smooth muscle, endothelial, fibroblast, immune, cardiac neuron, endocardial, and myofibroblast), and two developmental stages (fetal-like iPSC-CVPC and adult, **Table S4**)”.

Reviewer #3 (Remarks to the Author):

The manuscript by D’Antonio et al. describes a comprehensive eQTL analysis of samples relevant for the cardiovascular system. The authors re-analyze large numbers of published RNA-seq samples. The key contribution is the systematic analysis of interaction effects for different contexts, such as developmental stage, organ, tissue and cell type. The paper is well written and addresses an important problem by providing results with the potential to improve the follow up analysis of GWAS hits.

Major comments:

1. My main concern is that the manuscript is not always clearly distinguishing between eQTL that can only be detected when the context information is available and eQTL that can be detected using ‘traditional’ separate eQTL analysis of individual developmental stages and tissues. It is reasonable to assume that there will be context specific eQTL, but as it stands now there is no clear distinction between those eQTL that can only be found when considering the context information and those that can be found by other approaches as well. It would be important to quantify this and then show the added value of these context specific eQTL in terms of how they improve the downstream analyses, such as overlap with GWAS.

Action: We agree with the reviewer that we should justify why our approach is suitable and provides improved results compared with available methods. To validate our combined eQTL results, we used a well-established method (multivariate adaptive shrinkage, mash)^{3,4}, which deals with large numbers of tests in many (e.g. dozens) different “conditions” (intended as “different tissues” in our study). By combining effect sizes and their standard errors across multiple tissues, mash improves the power to detect eQTLs and provides a tissue-context for each eQTL signal. However, mash can only work for

categorical features (such as developmental stage, organ or tissue: each sample can be described as belonging to only one of each developmental stage, organ or tissue), but not continuous (cell type proportions) features, rendering our approach the only currently available that can estimate cell type associations.

To respond to the Reviewer's concern, we proceeded as follows to validate enrichment of tissue-specific eQTLs in the cardiac GWAS signals:

Performing mash

Mash requires effect sizes and their standard errors in each tissue as input. To have a direct comparison between mash and our outputs for validation, we initially performed an eQTL analysis on each tissue separately using limix and the same covariates as our combined eQTL approach (see Methods section “Validation of tissue-associated eQTLs”). We then selected from our combined eQTL analysis the lead variant of each of the 18,030 primary and conditional gene expression eQTLs (11,692 primary and 6,338 conditional, Figure 1A), obtained their effect sizes and standard errors for each tissue from the single-tissue eQTL analyses, and used these values to perform mash with default parameters. We annotated each eQTL signal as active in a tissue if the local false sign rate (LFSR) < 0.05.

Comparing eQTLs obtained with different methods

Using the single-tissue eQTL analyses we found varying numbers of eGenes and eQTLs across tissues: 1) aorta: 9,812 eGenes, and 13,029 eQTL signals (3,217 conditional eQTLs); atrium: 7,148 eGenes, and 8,770 eQTL signals (1,622 conditional eQTLs); ventricle: 5,951 eGenes, and 7,353 eQTL signals (1,402 conditional eQTLs); coronary artery: 5,458 eGenes, and 6,143 eQTL signals (685 conditional eQTLs); and iPSC-CVPC: 3,393 eGenes, and 3,682 eQTL signals (289 conditional eQTLs, Table S9). The observed differences in the number of eQTLs in each tissue are likely due to power differences across the tissues from sample numbers and tissue heterogeneity.

We next examined the mash results obtained using the eQTLs (for eGenes) from the first-step of our combined approach coupled with their effect sizes and standard errors for each tissue from the single-tissue eQTL analyses as input. Mash improved the detection of eQTLs active in each tissue compared with the single-tissue eQTL analysis for all tissues but aorta (1.45X times for atrium, 1.63X for ventricle, 1.89X for coronary artery and 2.40X for iPSC-CVPCs, Figure S13). Interestingly, the least-powered tissues (i.e. with the smallest number of unrelated samples: iPSC-CVPCs and coronary artery) had the largest increase in the number of eQTL signals detected, whereas the tissue with the largest number of tested samples (aorta) had a comparable number of eQTLs detected with the two methods. These results demonstrate that performing one combined eQTL analysis using all tissue samples and then identifying tissue associations results in increased power, compared with performing single-tissue eQTL analyses, as was previously suggested^{1,3,4}.

Of note, neither the traditional tissue-by-tissue eQTL analyses, nor the mash approach is able to detect cell type-associations, which can only be identified by using our interaction approach (described in the Methods section “Spatiotemporal eQTL mapping”). Therefore, while mash is useful to determine the tissue context of each eQTL, it cannot fully substitute for our combined analysis approach, as it cannot be used to test cell type associations.

Figure S13: Comparison of the number of tissue-associated eQTLs using two methods: single-tissue eQTLs versus mash

Figure S13: Barplots showing the number of eQTLs detected per tissue performing single-tissue eQTL analysis versus using mash on the eQTLs (for eGenes, ie, not for eIsoforms) identified in the first-step of the combined analysis (This study eQTLs) with their effect sizes and standard errors for each tissue from the single-tissue eQTL analyses as input.

We have added a detailed “Supplemental Note 2” and the following to the Results section “Cardiac traits enriched for eQTLs that function in specific spatiotemporal contexts”:

“To validate enrichments of the tissue-specific eQTLs and GWAS signals, we determine the tissue context of each eQTL signal using an independent method, multivariate adaptive shrinkage (mash)³. We observed very similar enrichments to those obtained using our combined analysis approach (Figure S13, Figure S14, Table S9, Table S10, Supplemental Note 1)”.

We have expanded the Methods by adding the section “Validation of the enrichment of tissue-associated eQTLs for GWAS cardiac traits” to describe these analyses:

“To validate enrichment of tissue-specific eQTLs in the cardiac GWAS signals we used the multi-variate adaptive shrinkage (mash) method⁴ – see Supplemental Note 1 for details on this method. We initially performed independent tissue-specific eQTL analyses on five tissues (iPSC-CVPC, arteria, aorta, ventricle, and atrium) using a linear mixed model (limix) and the same covariates as our combined eQTL approach: sex, normalized number of RNA-seq reads, % of reads mapping to autosomes or sex chromosome, % of mitochondrial reads, 20 genotype PCs, and kinship matrix, and recalculated the number of optimal PEER factors. Gene expression was quantile-normalized across all samples within each of the five datasets, and the number of optimal PEER factors was determined using the same method described for the combined eQTL analysis: we obtained 200 random genes and performed eQTL analyses using a range of PEER factors and selected the number of PEER factors for each dataset associated with the maximum number of detected eQTLs for these 200 genes. We used the following number of PEER factors: 40 for iPSC-CVPC, 60 for atrium, 70 for ventricle, 70 for aorta, and 30 for coronary artery. The kinship matrix was included as a random effect. Conditional eQTL analysis and FDR correction was performed using the same method as our combined eQTL approach (Table S9).

To have a direct comparison between mash and our outputs for validation, we next examined the mash results obtained using the eQTLs from the first-step of our combined approach (for eGenes, not for eIsoforms; Figure 1A) coupled with their effect sizes and standard errors for each tissue from the single-tissue eQTL analyses as input – see Supplemental Note 1 for details. We ran mash with default parameters and obtained the local false sign rate (LFSR)^{3,4}. We annotated each eQTL signal as associated with a tissue if the LFSR < 0.05 (Table S10). We also filtered at more stringent thresholds (up to 1×10^{-30}). At the least stringent LFSR threshold (LFSR = 0.05), we observed < 11,500 eQTLs in each of the four adult tissues and 8,856 in iPSC-CVPCs (Figure S13). Using the eQTLs detected by mash for each tissue, we performed enrichment tests across the five cardiac GWAS traits, as described in the previous section (Figure S14)”.

2. While the statistical approaches overall seem to be appropriate, the methods are sometimes not described with sufficient detail (see the following specific comments) to be able to fully judge all results.

Action: We apologize for the lack of details in our Methods. In response to this and other reviewers’ comments, we have substantially reorganized and updated the Methods to describe our approaches in greater detail. To provide more clarity about the analyses, we divided the Methods into four sections: 1) Data processing; 2) Combined eQTL analysis; 3)

Covariates for eQTL mapping; 4) Testing enrichment of eQTL signals in different genomic annotations; 5) Colocalization between eQTL signals; 6) spatiotemporal eQTL mapping; and 7) colocalization between GWAS and eQTL signals.

We provide below several updates that we believe are relevant to this reviewer's concerns:

1. Reviewer 1 comment 2 (R1C2): we updated the description of PEER factors (“Covariates for eQTL mapping”)
2. R1C3: we updated the description of the genotype PCA and kinship matrix generation (“Covariates for eQTL mapping”)
3. R1C7: we provide a more detailed description of the covariates used for the eQTL mapping (“Covariates for eQTL mapping”);
4. R1C9, R2C2, R3C5 and R3C6: we expanded the description of the detection of spatiotemporal eQTLs (“Spatiotemporal eQTL mapping”);
5. R2C1: we expanded the description of conditional eQTLs (“Combined eQTL analysis”);
6. R3C8: we improved the description of the filtering of colocalization by posterior probability (“Colocalization between GWAS and eQTL signals”)
7. We updated the Methods section “Data processing” to address the following concerns:
 - a. R1C5: description of the methods used to quantify isoform usage;
 - b. R1C6: normalization of the gene expression levels for eQTL detection;
 - c. R1C10: description of the iPSC-CVPC differentiation protocol.

Specific comments:

3. What makes it sometimes a bit difficult to follow this paper is a lack of clarity as to which set of eQTL results (global or ‘spatiotemporally regulated’) is being used in the different analyses. Please specify this explicitly for each analysis.

Action: We apologize for the confusion on this. To avoid any misunderstanding, for any analyses using spatiotemporal eQTLs, we explicitly state “spatiotemporal eQTLs”. We refer to the “global” eQTLs from the combined analysis simply as eQTLs. **Table S2** contained all eQTLs that were identified in this study, and **Table S4** provides spatiotemporal information for each of the identified eQTLs.

4. In the GWAS analysis the authors should demonstrate the added value of using their ‘spatiotemporally regulated’ eQTL instead of ‘regular’ eQTL from GTEx (for the tissues considered here) for the interpretation of GWAS loci. How many loci are there, for which only the ‘spatiotemporally regulated’ eQTL colocalize?

Action: Combined method approaches, such as the one used in this study and FastGxC⁵, distinguish between tissue-specific and tissue-shared eQTLs using linear mixed models. These methods are performed in two steps: in the first-step a combined eQTL analysis is conducted using all tissues and then in the second-step context-specific eQTLs are identified using an interaction test between genotype and each of the contexts. Hence, the spatiotemporal eQTLs are a subset of all the eQTLs that we identified in this study. Therefore, the added value of using spatiotemporal eQTLs is in the enrichments that we observe within each spatiotemporal context.

We sought to validate enrichments of the tissue-specific eQTLs and GWAS signals shown in **Figure 5E-G** and **Figure S9**. Using the eQTLs detected by mash for each tissue, we performed enrichment tests across the five cardiac GWAS traits. We used varying LFSR thresholds to define whether a gene was expressed in a given tissue, and observed similar enrichments to those obtained using our combined analysis approach (**Figure S14**):

- Myocardial infarction does not have strong associations with any tissue;
- QRS duration does not have strong associations with any tissue;
- Atrial fibrillation is enriched for eQTLs active in atrium and, at stringent LFSR thresholds, with ventricle ($LFSR \leq 1 \times 10^{-5}$) and with iPSC-CVPC ($LFSR \leq 1 \times 10^{-10}$);
- Pulse rate is enriched for eQTLs active in atrium and ventricle at all LFSR thresholds, and with aorta at most thresholds;
- Pulse pressure is enriched for eQTLs active in aorta at all thresholds, iPSC-CVPC at stringent LFSR thresholds ($LFSR \leq 1 \times 10^{-10}$), and coronary artery at loose LFSR thresholds ($LFSR \geq 0.05$).

We have added a detailed “Supplemental Note 2” and expanded the Methods by adding the section “Validation of the enrichment of tissue-associated eQTLs for GWAS cardiac traits” to describe these analyses:

“To have a direct comparison between mash and our outputs for validation, we next examined the mash results obtained using the eQTLs from the first-step of our combined approach (for eGenes, not for eIsoforms; Figure 1A) coupled with their effect sizes and standard errors for each tissue from the single-tissue eQTL analyses as input – see Supplemental Note for details. We ran mash with default parameters and obtained the local false sign rate (LFSR)^{3,4}. We annotated each eQTL signal as associated with a tissue if the LFSR < 0.05 (Table S10). We also filtered at more stringent thresholds (up to 1×10^{-30}). At the least stringent LFSR threshold (LFSR = 0.05), we observed < 11,500 eQTLs in each of the four adult tissues and 8,856 in iPSC-CVPCs (Figure S13). Using the eQTLs detected by mash for each tissue, we performed enrichment tests across the five cardiac GWAS traits, as described in the previous section (Figure S14)”.

5. The enrichment of GWAS SNP among ‘spatiotemporally regulated’ eQTL is interesting. The authors distinguish enrichment for the different interacting variables (contexts). The way this is currently described (e.g. ‘Pulse pressure was enriched for fetal-like iPSC-CVPC-, arteria-, aorta-, smooth muscle-, endocardial- and immune- eQTLs’) implies, that these eQTL are specific to the respective context. This is however not really the case, as the interaction term would also be significant if the eQTL was active in all contexts but the one tested for. Can the authors dissect the context specific contributions, as this would be particularly interesting for understanding the disease processes.

Action: The reviewer has made an excellent point. We agree that an interaction can be significant if the eQTL is active in all other contexts except for the one tested for. However, the two-step method that we used to identify context-specific eQTLs addresses this issue. In our original submission, we had added the second filtering step to address the potential issue described by the Reviewer, but we realize that it was not thoroughly described. In this second filtering step, for binary contexts (developmental stage, tissue and organ), we divided all samples into two sets corresponding to the two binary values (for example: 1) left ventricle; 2) all the other samples). For cell types, we divided all samples into quartiles and analyzed the top and bottom quartiles. We performed an additional standard eQTL analysis on each of the two sets. P-values were corrected across all interactions using Benjamini-Hochberg’s method. Associations with adjusted p-value < 0.05 were considered significant. We considered the following scenarios:

- e. If the eQTL is significant only for the tested context (for example: left ventricle), the eQTL signal is considered context-specific (left ventricle-specific);
- f. If the eQTL is significant only for all the other samples, the eQTL is not considered context-specific and thus not used in downstream analysis;
- g. If the eQTL is significant for both sets but its effect size is stronger (larger absolute value) in the tested context, the eQTL is context-associated (left ventricle-associated);
- h. If the eQTL is significant for both sets but its effect size is stronger (larger absolute value) in all the other samples, the eQTL is not considered as context-specific or context-associated.

In Supplementary Figures, we show examples of eQTLs with significant interactions with stage (Figure S1A), tissue (Figure S2), and cell type (Figure S3).”

We have updated the Methods section “Additional filtering steps to determine context-specific and context-associated eQTLs” to describe the second filtering step more clearly:

“While the interaction LMM described above identifies context-specific eQTLs, an interaction would also be considered significant if the eQTL is active in all other samples but the context being tested (for example: if we are testing the interaction with “left ventricle”, the eQTLs that are significant in all the other tissues but left ventricle would have a significant p-value). To avoid counting these associations as “context-specific” we included the following filtering step for all spatiotemporal eQTLs (i.e., those with a significant interaction term in the above analysis). For binary contexts (developmental stage, tissue and organ), we divided all samples into two sets corresponding to the two binary values (for example: 1) left ventricle; 2) all the other samples). For cell types, we divided all samples into quartiles and analyzed the top and bottom quartiles. We performed an additional standard eQTL analysis on each of the two sets. P-values were corrected across all interactions using Benjamini-Hochberg’s method. Associations with adjusted p-value < 0.05 were considered significant. We considered the following scenarios:

1. eQTL is significant only for the tested context (for example: left ventricle): the eQTL signal is considered as context-specific (left ventricle-specific);
2. eQTL is significant only for “all the other samples”: the eQTL is not considered as context-specific;

3. eQTL is significant for both sets but its effect size is stronger (larger absolute value) in the tested context: the eQTL is context-associated (left ventricle-associated);
4. eQTL is significant for both sets but its effect size is stronger (larger absolute value) in “all the other samples”: the eQTL is not considered as context-specific or context-associated”.
6. In the fine mapping analysis the authors suggest that fine mapping ‘reduces the number of candidate causal variants to only a handful in the majority of loci and provides a spatiotemporal molecular mechanism underpinning the association between genetic variation and cardiac traits.’ Again it would be important to (systematically) show here the added value of using their ‘spatiotemporally regulated’ eQTL instead of ‘regular’ eQTL from GTEx (for the tissues considered here). How much smaller do the credible sets become and how many more loci can be resolved?

Action: We apologize for the confusion: to identify spatiotemporally regulated eQTLs, we only tested the lead variant for each eQTL signal and annotated the eQTL signal according to whether its lead variant has a spatiotemporal association. Therefore, we are not proposing that fine mapping “reduces the number of candidate causal variants to only a handful in the majority of loci and provides a spatiotemporal molecular mechanism underpinning the association between genetic variation and cardiac traits”, but rather that genetic fine mapping in general reduces the number of potential causal variants at many GWAS loci, and we provide a framework to annotate the signals that colocalize between eQTLs and GWAS with a spatiotemporal context.

We have updated the Results section “**Fine mapping using colocalization data identifies putative causal variants for hundreds of loci**”:

“These results show that genetic fine mapping GWAS loci by colocalization with eQTL signals reduces the number of candidate causal variants to only a handful in the majority of loci”.

We have also updated the Methods section “**Spatiotemporal eQTL mapping**”:

“Using the *iscan* function in *limix*, we implemented the following linear mixed model, which we applied to the lead variant of each eQTL signal (primary and conditional)”.

7. Is the definition of the credible sets based on colocalization a standard approach? It seems that the 95% credible sets are usually defined based only on the GWAS signal. How do the results based on these two definitions compare to each other?

Action: By definition, an $X\%$ credible set is a list of variants that cumulatively contribute to $>X\%$ probability to contain the causal variant. There are multiple approaches for “fine mapping” of GWAS loci: 1) only using GWAS signals, originally termed “genetic fine mapping” in the definition provided by Mahajan et al.³⁴, which simply converts p-values to posterior probabilities of association; 2) using a GWAS signal in combination with functional features, termed “functional fine mapping”³⁴ - methods such as FGWAS³⁵ are based on this approach; and 3) colocalization²⁴ between GWAS and eQTLs also referred to as “genetic fine mapping”, which results from the assumption that colocalization can discriminate between genetic signals that have the same or different underlying causal variants. To address the Reviewer’s concern, we performed a standard GWAS genetic fine mapping and compared it with our colocalization approach. We found that our colocalization approach results in stronger posterior probabilities for the lead variants ($p = 2.07 \times 10^{-7}$, paired t-test) and smaller credible set sizes ($p = 1.15 \times 10^{-13}$, paired t-test, **Figure S15**).

We have added this analysis to the Results section “**Fine mapping using colocalization data identifies putative causal variants for hundreds of loci**”:

“To compare the results from fine mapping by colocalization with eQTLs with standard genetic fine mapping, for each of the 210 loci we performed a standard genetic fine mapping using the GWAS signals alone^{24,34}, and observed that the colocalization approach resulted in smaller credible set sizes ($p = 2.07 \times 10^{-7}$, paired t-test, **Figure S15**) and stronger posterior probabilities for the lead variants ($p = 1.15 \times 10^{-13}$).”;

And to the Methods section “Fine-mapping and obtaining 99% credible sets for GWAS signals”: “Standard genetic fine mapping³⁴, which converts the p-value of each variant at each of the 210 GWAS loci to PPA, without considering the colocalization with eQTLs. We used the *finemap.abf* function from the coloc package in R²⁴ on the GWAS summary statistics at each of the 210 loci. 99% credible sets describe the smallest set of variants that cumulatively contribute to \geq 99% of the PPA in each locus”; and added **Figure S15**.

8. Observing a mismatch between the GWAS lead SNPs and the SNPs with highest coloc posterior might also indicate that the trait and gene expression are actually driven by different causal variants. The authors should also discuss this possibility in the light of this preprint: <https://www.biorxiv.org/content/10.1101/2022.05.07.491045v1>

Action: Thank you for these interesting suggestions. After reading the suggested paper, we agree that there are limitations to using eQTLs for the characterization of GWAS associations that should be discussed. Therefore, we have now inserted the following text in the Discussion to emphasize this point: “Of note, there remains hundreds of GWAS signals that did not colocalize with eQTLs, indicating that there is still a substantial fraction of GWAS that could not be explained by eQTLs alone. This is consistent with recent findings that eQTL and GWAS studies are biased towards different types of variants in aspects of regulatory properties and genomic enrichments³⁶. Therefore, future studies on the functional characterization of genetic variants will likely require integration with other functional information, such as epigenomic data to identify candidate causal variants”.

The other concern by the Reviewer (mismatches between GWAS and eQTL signals describe different causal variants) is taken care of by colocalization, which provides a posterior probability for each of five hypotheses: 1) H0: neither trait has a significant association at the tested locus; 2) H1: only the first trait is associated; 3) H2: only the second trait is associated; 4) H3: both traits are associated but the underlying variants are different; and 5) H4: both traits are associated and share the same underlying variants. As we describe in the Methods section “Colocalization between GWAS and eQTL signals”, “we considered a trait to colocalize with a gene if the GWAS signal colocalized (PP-H4 > 0.8) with the eGene or any of its eIsoforms, while we discarded all the colocalizations with high PP-H3”.

9. The authors report that SNPs that are eQTL for multiple genes are more likely to be ‘spatiotemporally regulated’. It is unclear which set of eQTL results is used for the initial analysis to identify the SNPs that are eQTL for multiple genes. In addition, please clarify how the statistical tests supporting this statement reported for stage, organ, tissue and cell type are computed.

Action: We apologize for the lack of clarity in our analyses and have added a new Methods sub-section (“Validation of tissue-associated eQTLs – Enrichment of spatiotemporally regulated eQTLs in polygenic eQTLs”) to describe the associations between colocalizing eGenes and their spatiotemporal contexts (described in **Figure 3A,B**):

“To determine if two eGenes or eIsoforms had the same underlying eQTL signal, we used an approach similar to the one described in the section “Colocalization between eQTL signals”. Briefly, we obtained all pairs of eGenes within 500 kb of each other and performed colocalization between each combination of primary and conditional eQTL signals for both eGenes. Two eGenes or eIsoforms colocalized if they had PP-H4 > 0.8. Next, we tested the association between the number of genes that share the same eQTL signal and the spatiotemporal context of the eQTL signal (coded as a binary variable: 1 = eGenes are associated with the spatiotemporal context; 0 otherwise) using linear regression. To confirm that eQTL signals associated with multiple eGenes were more likely than expected to be associated with the same spatiotemporal context, we also performed a permutation analysis. We counted the number of occurrences where the same spatiotemporal context was shared between two colocalizing eGenes and compared it with the distribution of 100 permutations of each eGene-spatiotemporal context association. We calculated Z-scores and converted them to p-values using the *pnorm* function in R”.

10. The comparison of eQTL for sense and corresponding antisense transcripts is quite interesting. Please add more details: how many pairs are being tested? are the effect sizes correlated or anti-correlated? Are there any enrichments for regulatory elements overlapping with these SNPs that could explain these effects?

Action: Thank you for these questions. We tested 163 pairs of eGenes and their antisense RNAs.

We have updated the Results section “Cardiac genes and their paired antisense RNA enriched for sharing eQTL signals.

“In total, we tested 163 pairs of genes and their associated antisense RNAs, and observed that 47 pairs shared the same eQTL signal”

We have updated the Methods section “Colocalization between eQTL signals”:

“To determine whether paired genes and antisense RNAs were enriched for sharing an eQTL signal, we obtained the gene symbols of all eGenes from Gencode. We identified 163 gene/antisense pairs where both the gene and its antisense RNA were eGenes and whose gene symbols were “Gene A” and “Gene-A-AS1”, respectively. We found that 47 of these pairs shared the same eQTL signal (PP-H4 > 0.8) and tested whether the proportion of colocalized eQTL was greater than expected between any pair of eGenes that were within 500 kb of each other (146,472 pairs) using a Fisher’s Exact Test”.

We investigated the regulatory function of eQTLs associated with correlated and anticorrelated eGene/antisense pairs, and have added Supplemental Note 1, Figure S6, Figure S7, and Figure S8:

“In total, we tested 163 pairs of genes and their associated antisense RNAs and observed that 47 pairs shared the same eQTL signal. We examined the enrichment of genes that share an eQTL signal with their antisense RNA compared to those that do not. We found that pairs that share an eQTL were enriched for splice acceptor sites and promoter regions while those that have distinct signals were enriched in intronic and splice donor regions (Figure S6A). This indicates that as previously observed^{37,38}, there is a coordinated regulatory mechanism of promoters shared by eGenes-antisense pairs of genes.

Of the 47 pairs, we found 12 whose eQTL acted in opposite directions on the expression of the eGene and its antisense RNA (Figure S7, Figure S8). We next tested whether the eQTLs for correlated and anticorrelated eGene/antisense pairs were associated with different types of regulatory elements and found that correlated pairs were enriched for intergenic regions while anti-correlated pairs were enriched for splice acceptor and donor sites and exonic regions (Figure S6B). The 12 pairs with opposite effects included two genes whose antisense RNA is known to be involved in the negative regulation of gene expression: *TRAPPC12/TRAPPC12-AS1* and *ABCF2/ABCF2-AS1*^{28,29}. These results suggest that, for these 12 pairs, the eQTLs may increase the regulatory effect of the antisense RNA, whose change in expression (increase/decrease) results in the opposite expression change (decrease/increase) on their target gene”.

updated the Methods section “Colocalization between eQTL signals”:

“To determine if eQTLs associated with gene/antisense pairs that act in the same versus opposite directions have differential enrichments, we obtained all fine-mapped variants with PPA > 0.01 (from colocalization) and annotated each variant by its location relative to both the gene and antisense (intergenic, introns, promoters, UTRs, splice donor sites, splice acceptor sites, and exons). We then used a Fisher’s Exact Test to calculate the enrichment by comparing the proportion of SNPs within the genomic region between correlated and anti-correlated gene/antisense pairs. P-values were adjusted using Benjamini-Hochberg’s method. Enrichments with FDR < 0.05 were considered significant. This analysis was repeated for the comparison between pairs that shared the same eQTL signal (PP-H4 > 0.8) and pairs that had distinct signals (PP-H3 > 0.8). For the gene/antisense pairs with distinct signals (PP-H3 > 0.8), we used all fine-mapped (from *finemap.abf*) variants for both genes and antisense with PPA > 0.01”.

11. Another interesting aspect concerns the pairs of genes and antisense transcripts that do not share eQTL. Is there any difference in terms of annotations between genes that share eQTL with their antisense transcript and those that do not?

Action: As the reviewer suggested, we examined the enrichment of genes that share eQTLs with their antisense RNA compared to those that do not. We found that pairs that share an eQTL were enriched for splice acceptor sites and promoter regions while those that have distinct signals were enriched in intronic and splice donor regions. This enrichment of shared signals in promoter regions is consistent with previous observations that antisense and gene regulation are dependent on each other.

See response to Comment 10 including Supplemental Note 1.

12. In general the first section on annotation and comparison is not so novel and rather a QC and comparison to previous findings. As such, it may be moved into a supplementary section and only discussed very briefly in the results?

Action: We kindly thank the reviewer for this suggestion. However, although many other studies have performed gene and isoform eQTL analyses, we believe that this is an important finding we need to highlight as it has not yet been shown, at least widely-established, that isoform eQTLs computed based on isoform usage are associated with alternative splicing properties. Therefore, we believe that keeping this figure as a main figure would be highly beneficial for understanding other results in the paper.

13. Additional validation of the cell type specific eQTL should be shown for the cardiomyocytes by comparison of overlap and correlation of effect sizes from the GTEx iQTL analysis (Kim-Hellmuth Science 2020).

Action: We thank the reviewer for this suggestion. We compared the correlation of effect sizes of GTEx myocyte-associated iQTLs for heart left ventricle with all of the interactions we tested in our study. We found that there was a positive correlation between GTEx myocyte-associated iQTLs and cardiac muscle-, heart-, and heart ventricle-associated eQTLs.

We have added the following to the Results section “Mapping spatiotemporal cardiovascular eQTLs”:

“Furthermore, we observed that cardiac muscle-, heart- and left ventricle-specific eQTLs were strongly correlated with the set of myocyte eQTLs that were described by GTEx ³⁹ (Figure S5)”

and to the Methods section “Validation of cell type-associated eQTLs (comparison with GTEx cell-type associated eQTLs)”:

“To validate the eQTLs associated with spatiotemporal contexts we also compared the correlation of effect sizes with cell type-associated eQTLs identified by the GTEx Consortium ³⁹. We downloaded myocyte interaction eQTLs for heart left ventricle from the GTEx Data Portal and tested the correlation between effect sizes for matching eGenes using the cor.test function in R. We observed positive associations between GTEx iQTL for myocytes in heart left ventricle and cardiac muscle-specific/associated, heart-specific/associated eQTLs, and heart ventricle-specific/associated eQTLs (Figure S5); and added Figure S5.

14. The conclusion that “Different mechanisms underlie eQTLs for genes and isoforms, may be a bit overstated, as 56% of genes have actually a shared underlying causal variant.

Action: We now have changed our section title to “Mechanisms Underlying eQTLs for Genes and Isoforms”.

15. Annotation of isoform and gene level eQTL could be a bit more fine grained. Please also compare to the findings in GTEx or the Geuvadis paper.

Action: We thank the reviewer for this suggestion. We have compared our results to the GTEx paper (GTEx et al., Science, 2020) and found that our results were similar to their findings. Isoform-level eQTLs were enriched in transcribed regions while gene-level eQTLs were enriched in transcriptional regulatory elements.

We have inserted the following text in the Discussion to further expand our analyses on gene and isoform eQTLs: “These results were consistent with previous studies, which described an enrichment in regulatory elements for variants affecting gene expression ^{2,40} and transcribed regions for variants affecting alternative splicing ^{2,41}”.

16. In the global eQTL analysis it is not clearly stated whether this has been done for each tissue separately or using the joint model.

Action: We used a single test combining all 966 samples, using an LMM to identify eQTL associations for each tissue. We have inserted the following text in the Methods to explicitly state this information: “To identify genetic variants that are associated with cardiac gene expression, we performed a joint eQTL analysis using all 966 samples”. We have also added a heading in the Results section: “Combined eQTL analysis”, and the following sentence in this section: To map the regulatory effects of genetic variants on these fetal-like and adult cardiac tissues, we performed a combined eQTL analysis on all 966 RNA-seq samples using a linear mixed model (LMM) including a kinship matrix to account for genetic relatedness between samples.

17. Using 285 PEER factors seems very high. In the literature smaller numbers are typically used. It would be important to check if these factors still capture meaningful variation in the data and to make sure that the fitting actually converged.

Action: We thank the reviewer for this comment and refer the reviewer to our response to Reviewer #1's Comment #2 that expressed similar concerns. Briefly, we plotted the number of eQTLs discovered as a function of the number of PEER factors used in the model and found that using 285 PEER factors resulted in maximal eQTL discovery. We reasoned that using more would lose power. Furthermore, anything fewer than 100 PEER factors was not optimal for our joint eQTL analysis, as shown in the plot. This method of selecting PEER factors was similar to the methods used in previous studies^{2,6,7}. Below, **Figure S18A** shows that using 285 PEER factors results in the largest number of detected eQTLs.

Figure S18: Assessing the optimal number of PEER factors

Figure S18A: Proportion of eGenes (Y axis) using different combinations of PEER factors (X axis: 5-300 PEER factors). The maximum number of eGenes is reached at 285 PEER factors.

In our updated manuscript, we included **Figure S18** and updated the Methods section “Covariates for eQTL mapping” to detail these points:

“We performed eQTL mapping using the following covariates (**Table S1**): 1) sex; 2) normalized number of RNA-seq reads; 3) % of reads mapping to autosomes or sex chromosome; 4) % of mitochondrial reads; 5) 20 genotype principal components to account for global ancestry; 6) 285 PEER factors to account for transcriptome variability; and 7) kinship matrix.

Genotype Principal Component Analysis: We previously performed principal component analysis (PCA) on WGS variants to determine the global ancestry of each individual in this study⁹. Briefly, we used the genotypes of 1,634,010 SNPs that had allele frequencies between 30% and 60% in the 1000 Genomes Phase 3 Project and genotyped in both iPSCORE and GTEx. We merged the VCF files from 1000 Genomes, iPSCORE, and GTEx, and performed PCA using the *pca* function in *plink 1.90b3x*¹⁴. We showed that iPSCORE and GTEx individuals overlap their associated 1000 superpopulations, and that iPSCORE includes more individuals of East Asian ancestry than GTEx. The top 20 genotype principal components used to account for global ancestry are available for all 491 individuals (139 iPSCORE and 352 GTEx) in the expanded **Table S1** in Figshare.

PEER factors: We sought to determine the optimal number of PEER factors⁸ for obtaining the maximum number of eQTLs. We initially calculated PEER factors on the 10,000 expressed genes with the largest variance across all samples. To limit biases due to the expression levels of each gene, we divided the 19,586 expressed genes into ten deciles based on their average TPM, and selected 20 random genes for each decile, for a total of 200 genes. We next performed eQTL analysis on each of the 200 genes using the following numbers of PEER factors: 5, 10, 20 and increments of 20 between 20 and 300. These eQTL analyses of the 200 genes to determine the optimal number of PEER factors was performed using the same covariates as for the combined eQTL analysis: 1) sex; 2) normalized number of RNA-seq reads; 3) % of reads mapping to autosomes or sex chromosome; 4) % of mitochondrial reads; 5) 20 genotype principal components to account for global

ancestry; 6) the selected number of PEER factors to account for transcriptome variability; and 7) kinship matrix. We observed the maximum number of eQTLs using 280 PEER factors. To further refine the optimal number of PEER factors, we performed a second eQTL analysis using 270, 275, 285 and 290 PEER factors, which determined that 285 PEER factors were optimal (Figure S18A). For isoform eQTLs, we used 80 PEER factors as a compromise between obtaining the largest number of eQTLs and computational burden. To confirm that the PEER factors were accounting for batch effects between the iPSCORE and GTEx samples, we examined the distribution of the samples in PEER space and the correlation of the top 10 PEER factors with other covariates (Figure S18B). We also examined the correlation between the first 10 PEER factors and other covariates and observed that the PEER factors were distinctly correlated with iPSCORE and GTEx datasets, as well as other technical covariates (Figure S18C). These analyses show that batch effects were properly corrected for between the iPSCORE and GTEx datasets.

Kinship Matrix: The random effects term captures relatedness between samples and are described in a kinship matrix. To construct the kinship matrix, we used the kinship function in plink 1.90b3x⁴² using the same set of 1,634,010 SNPs employed in the genotype PCA”.

18. Many different eQTL pipelines exist but it seems that a consensus pipeline is slowly emerging. The authors used eigenMT for gene wise multiple testing adjustment. It would be important to justify why and potentially also to compare to the GTEx pipeline (fastqtl).

Action: Our cohort contains individuals who are genetically related to each other, as well as multiple samples from the same individual. Therefore, a standard linear regression, such as fastqtl, is not compatible with our study design as it would not account for relatedness. Linear mixed models can adjust for this by incorporating the kinship matrix. We used limix in our previous study⁶. EigenMT is a widely used method that performs multiple test correction at the gene level while accounting for LD between variants. This method has been used by previous studies, including GTEx^{2,7,43}. Our two-step FDR correction (1: all variants tested for the same gene, using eigenMT; and 2: lead variants for all the 19,586 tested genes and 37,032 isoforms) has been previously proposed by Huang et al.⁴⁴ to account for both local and global multiple tests.

We have inserted the following text to further describe why we used eigenMT and a linear mixed model as opposed to fastQTL in the Methods section, “**Combined eQTL analysis**”:

“To account for genetic relatedness between samples, we performed eQTL mapping using a linear mixed model (LMM) with limix v.3.0.4⁴⁵ (*scan* function), which incorporates a kinship matrix as a random effect” and “FDR correction: To perform FDR correction, we used a two-step procedure similar to the one described in Huang et al.⁴⁴ that first corrects at the gene level and then at the genome-wide level. 1) For each gene, p-values were FDR-corrected using eigenMT, which takes into account the LD structure of the tested variants; and 2) across the lead variants for all the 19,586 tested genes and 37,032 isoforms, p-values were corrected for false discovery rate (FDR) using Benjamini-Hochberg’s correction and considered only eQTLs with q-values < 0.05 as significant”.

19. Currently, there is no information on how many repeated measurements (same person, multiple tissues) are present in the data and how these are treated in the linear mixed model. I would assume that this is encoded in the kinship matrix. Please clarify.

Action: The reviewer is correct in that this information is encoded in the kinship matrix. In our study, we have more than one repeated measurement for 23 iPSCORE individuals (range 2-6 per individual) and 228 GTEx individuals (range 2-4 per individual).

This information was provided in Table S1, and we have clarified this information in the Methods section “**Data processing**”:

“23 iPSCORE subjects had more than one iPSC-CVPC samples (range: 2-6), whereas 228 GTEx individuals had samples for at least two of the four adult cardiac tissues (range: 2-4)”

20. In the results the interaction analysis of stage, organ or tissue cites ref 46 (Casale, F.P., Rakitsch, B., Lippert, C. & Stegle, O. Efficient set tests for the genetic analysis of correlated traits. Nat Methods 12, 755-8 (2015)). This paper describes a multi trait set test but no interaction analysis. Please provide more details on how this test is performed exactly.

Action: limix has been updated since publication. We refer the reviewer to the GitHub page of the software (<https://github.com/limix/limix>). We also updated the reference to cite this webpage.

21. The methods on detecting stage, organ, tissue or cell type specific eQTLs need to be clarified more. From the results section, it seems that separate linear models ($\text{expr} \sim \text{genotype}$) are computed for groups of samples depending on tissue or cell type proportion. How are these then compared according to which criteria? Please add this to the methods.

Action: We apologize for not making this analysis clear. We performed the second linear mixed regression to identify eQTLs that are active in the context being tested. As the Reviewer stated in their previous comment (#5), an interaction can be significant if the eQTL is active in all other contexts except the one being tested. To ensure we do not retain these interactions, we performed a filtering step that included two linear regressions on gene expression \sim genotype: one for the “positive” samples (the tested spatiotemporal context or the samples in the top quartile of the tested cell type); and the other on the “negative” samples (all the other samples or the samples in the bottom quartile of the tested cell type). We only considered interactions in which there is a significant eQTL association for the “positive” samples.

We have modified the first sentence in the Results section “**Mapping spatiotemporal cardiovascular eQTLs**”:

“To identify eQTLs for eGenes and eIsoforms that function in a spatiotemporal manner, we used an interaction term to determine if the lead eVariants identified in the combined analysis were associated with samples at one stage but not the other (iPSC-CVPC or adult) or samples annotated as a specific organ (arteria and heart) or tissue (atrium, ventricle, aorta and coronary artery)”

We have inserted the following text in the Methods section “**Additional filtering steps to determine context-specific and context-associated eQTLs**” to prevent further ambiguity:

“While the interaction LMM described above identifies context-specific eQTLs, an interaction would also be considered significant if the eQTL is active in all other samples but the context being tested (for example: if we are testing the interaction with “left ventricle”, the eQTLs that are significant in all the other tissues but left ventricle would have a significant p-value). To avoid counting these associations as “context-specific” we included the following filtering step for all Spatiotemporal eQTLs (i.e., those with a significant interaction term in the above analysis). For binary contexts (developmental stage, tissue and organ), we divided all samples into two sets corresponding to the two binary values (for example: 1) left ventricle; 2) all the other samples). For cell types, we divided all samples into quartiles and analyzed the top and bottom quartiles. We performed an additional standard eQTL analysis on each of the two sets. P-values were corrected across all interactions using Benjamini-Hochberg’s method. Associations with adjusted p-value < 0.05 were considered significant. We considered the following scenarios:

1. eQTL is significant only for the tested context (for example: left ventricle): the eQTL signal is considered as context-specific (left ventricle-specific);
2. eQTL is significant only for “all the other samples”: the eQTL is not considered as context-specific;
3. eQTL is significant for both sets but its effect size is stronger (larger absolute value) in the tested context: the eQTL is context-associated (left ventricle-associated);
4. eQTL is significant for both sets but its effect size is stronger (larger absolute value) in “all the other samples”: the eQTL is not considered as context-specific or context-associated.

In Supplementary Figures, we show examples of eQTLs with significant interactions with stage (**Figure S1A**), tissue (**Figure S2**), and cell type (**Figure S3**).”

22. Please clarify for Fig 1E whether the test is actually contrasting SNPs that are eQTL for genes vs SNPs that are eQTL for isoforms or if the results show enrichment of SNP that are eQTL for genes vs SNPs that are not eQTL with positive sign and eQTL for isoforms vs SNPs that are not eQTL with negative sign.

Action: We apologize for not making this analysis clear. We tested the enrichment of eQTLs using a Fisher’s Exact Test comparing the proportion of fine mapped variants with PPA >0.01 for eGenes versus the fine mapped variants associated with isoform usage. In response to R1C8, we updated this analysis to compare the proportion of fine-mapped variants of eQTLs associated with gene expression versus eQTLs associated with isoform usage.

We have updated the Methods section “Testing enrichment of eQTL signals in different genomic annotations”:

“To test the enrichment of eQTLs in intergenic regions, introns, promoters, UTRs, splice donor sites, splice acceptor sites and exons, we obtained fine-mapped variants for all eQTL signals for both eGenes and eIsoforms.

Determining the 99% credible sets for each eQTL signal: To define a credible set of candidate causal variants for each eQTL association, we performed genetic fine mapping using the *finemap.abf* function in the *coloc* R package ²⁴. This Bayesian method converts p-values of all the variants tested for a specific gene to posterior probabilities of association (PPA).

Testing enrichment of eQTL signals in different genomic annotations: We obtained fine-mapped variants for all eQTL signals for both eGenes and eIsoforms with PPA > 0.01. We determined the overlap of each fine mapped variant and each genomic annotation using *bedtools intersect* and compared the proportions of variants overlapping each annotation between eGenes and eIsoforms using Fisher’s Exact Test. Promoters were defined as the 2,000 bp upstream of the transcription start site. Exons and UTRs were obtained from *Gencode V.34lift37*. For the intronic variants, we calculated their distance from the closest exon, in order to determine their overlap with splice sites”.

23. The abbreviation PPA for reporting the results of the coloc analysis is only introduced in the GWAS section but used already before.

Action: Thank you very much for this observation. We updated the Manuscript to move the abbreviation to the first mentioning of PPA.

24. It is quite surprising to find such a large difference between the number of colocalized eQTL for the different traits. Is there any difference between these GWAS studies (sample size, number of cases, number of GWAS loci)?

Action: As the reviewer proposed, the varying numbers of colocalized eQTLs for different traits is due to the fact that the total number of genome-wide significant loci is different between traits. Pulse rate and pulse pressure are known to have hundreds of genome-wide significant loci, whereas recent studies have identified <100 atrial fibrillation- or myocardial infarction-associated loci ^{46,47}. We show the Manhattan Plots for each trait in **Figure 4**. Below we provide the GWAS references from independent studies, which show the differences in the numbers of genome-wide significant loci for each of the five traits:

- 1) Pulse rate: Eppinga et al. ⁴⁸
- 2) Pulse pressure: Giri et al. ⁴⁹
- 3) QRS duration: Sotoodehnia et al. ⁵⁰
- 4) Atrial Fibrillation: Roselli et al. ⁴⁶
- 5) Myocardial Infarction: Hartiala et al. ⁴⁷

We added the following to the Results section “Colocalization identifies potential molecular mechanisms underlying GWAS signals”:

“This finding is consistent with the fact that pulse rate and pulse pressure are known to have hundreds of genome-wide significant loci, whereas less than 100 GWAS loci have been associated with atrial fibrillation- or myocardial infarction ⁴⁶⁻⁵⁰”.

References

1. Lu, A. *et al.* Fast and powerful statistical method for context-specific QTL mapping in multi-context genomic studies. *bioRxiv*, 2021.06.17.448889 (2021).
2. Consortium, G.T. The GTEx Consortium atlas of genetic regulatory effects across human tissues. *Science* **369**, 1318-1330 (2020).
3. Stephens, M. False discovery rates: a new deal. *Biostatistics* **18**, 275-294 (2017).
4. Urbut, S.M., Wang, G., Carbonetto, P. & Stephens, M. Flexible statistical methods for estimating and testing effects in genomic studies with multiple conditions. *Nat Genet* **51**, 187-195 (2019).
5. He, Y. *et al.* sn-spMF: matrix factorization informs tissue-specific genetic regulation of gene expression. *Genome Biol* **21**, 235 (2020).
6. Bonder, M.J. *et al.* Identification of rare and common regulatory variants in pluripotent cells using population-scale transcriptomics. *Nat Genet* **53**, 313-321 (2021).
7. Consortium, G.T. *et al.* Genetic effects on gene expression across human tissues. *Nature* **550**, 204-213 (2017).
8. Stegle, O., Parts, L., Piipari, M., Winn, J. & Durbin, R. Using probabilistic estimation of expression residuals (PEER) to obtain increased power and interpretability of gene expression analyses. *Nat Protoc* **7**, 500-7 (2012).
9. D'Antonio, M. *et al.* In heart failure reactivation of RNA-binding proteins is associated with the expression of 1,523 fetal-specific isoforms. *PLoS Comput Biol* **18**, e1009918 (2022).
10. Liu, W., Johansson, A., Rask-Andersen, H. & Rask-Andersen, M. A combined genome-wide association and molecular study of age-related hearing loss in *H. sapiens*. *BMC Med* **19**, 302 (2021).
11. Johnson, E.C. *et al.* A large-scale genome-wide association study meta-analysis of cannabis use disorder. *Lancet Psychiatry* **7**, 1032-1045 (2020).
12. Sinnott-Armstrong, N., Naqvi, S., Rivas, M. & Pritchard, J.K. GWAS of three molecular traits highlights core genes and pathways alongside a highly polygenic background. *Elife* **10**(2021).
13. Mogil, L.S. *et al.* Genetic architecture of gene expression traits across diverse populations. *PLoS Genet* **14**, e1007586 (2018).
14. Purcell, S. *et al.* PLINK: a tool set for whole-genome association and population-based linkage analyses. *Am J Hum Genet* **81**, 559-75 (2007).
15. Panopoulos, A.D. *et al.* iPSCORE: A Resource of 222 iPSC Lines Enabling Functional Characterization of Genetic Variation across a Variety of Cell Types. *Stem Cell Reports* **8**, 1086-1100 (2017).
16. D'Antonio-Chronowska, A., D'Antonio, M. & Frazer, K.A. In vitro Differentiation of Human iPSC-derived Cardiovascular Progenitor Cells (iPSC-CVPCs) *Bio-Protocol* **10**(2020).
17. DeBoever, C. *et al.* Large-Scale Profiling Reveals the Influence of Genetic Variation on Gene Expression in Human Induced Pluripotent Stem Cells. *Cell Stem Cell* **20**, 533-546 e7 (2017).
18. Dobin, A. *et al.* STAR: ultrafast universal RNA-seq aligner. *Bioinformatics* **29**, 15-21 (2013).
19. Frankish, A. *et al.* GENCODE reference annotation for the human and mouse genomes. *Nucleic Acids Res* **47**, D766-D773 (2019).
20. Tarasov, A., Vilella, A.J., Cuppen, E., Nijman, I.J. & Prins, P. Sambamba: fast processing of NGS alignment formats. *Bioinformatics* **31**, 2032-4 (2015).
21. Tischler, G. & Leonard, S. biobambam: tools for read pair collation based algorithms on BAM files. *Source Code for Biology and Medicine* **9**, 13 (2014).
22. Li, B. & Dewey, C.N. RSEM: accurate transcript quantification from RNA-Seq data with or without a reference genome. *BMC Bioinformatics* **12**, 323 (2011).
23. Nariai, N., Greenwald, W.W., DeBoever, C., Li, H. & Frazer, K.A. Efficient Prioritization of Multiple Causal eQTL Variants via Sparse Polygenic Modeling. *Genetics* **207**, 1301-1312 (2017).
24. Giambartolomei, C. *et al.* Bayesian test for colocalisation between pairs of genetic association studies using summary statistics. *PLoS Genet* **10**, e1004383 (2014).

25. Burridge, P.W. *et al.* Chemically defined generation of human cardiomyocytes. *Nat Methods* **11**, 855-60 (2014).
26. Lian, X. *et al.* Directed cardiomyocyte differentiation from human pluripotent stem cells by modulating Wnt/beta-catenin signaling under fully defined conditions. *Nat Protoc* **8**, 162-75 (2013).
27. Benaglio, P. *et al.* Allele-specific NKX2-5 binding underlies multiple genetic associations with human electrocardiographic traits. *Nat Genet* **51**, 1506-1517 (2019).
28. Chung, J. *et al.* Genome-wide pleiotropy analysis of neuropathological traits related to Alzheimer's disease. *Alzheimers Res Ther* **10**, 22 (2018).
29. Skuodas, S. *et al.* The ABCF gene family facilitates disaggregation during animal development. *Mol Biol Cell* **31**, 1324-1345 (2020).
30. Zhou, J. & Troyanskaya, O.G. Predicting effects of noncoding variants with deep learning-based sequence model. *Nat Methods* **12**, 931-4 (2015).
31. Zhou, J. *et al.* Deep learning sequence-based ab initio prediction of variant effects on expression and disease risk. *Nat Genet* **50**, 1171-1179 (2018).
32. Vinuela, A. *et al.* Genetic variant effects on gene expression in human pancreatic islets and their implications for T2D. *Nat Commun* **11**, 4912 (2020).
33. Brown, A.A. *et al.* Predicting causal variants affecting expression by using whole-genome sequencing and RNA-seq from multiple human tissues. *Nat Genet* **49**, 1747-1751 (2017).
34. Mahajan, A. *et al.* Fine-mapping type 2 diabetes loci to single-variant resolution using high-density imputation and islet-specific epigenome maps. *Nat Genet* **50**, 1505-1513 (2018).
35. Pickrell, J.K. Joint analysis of functional genomic data and genome-wide association studies of 18 human traits. *Am J Hum Genet* **94**, 559-73 (2014).
36. Mostafavi, H., Spence, J.P., Naqvi, S. & Pritchard, J.K. Limited overlap of eQTLs and GWAS hits due to systematic differences in discovery. *bioRxiv*, 2022.05.07.491045 (2022).
37. Engstrom, P.G. *et al.* Complex Loci in human and mouse genomes. *PLoS Genet* **2**, e47 (2006).
38. Yelin, R. *et al.* Widespread occurrence of antisense transcription in the human genome. *Nat Biotechnol* **21**, 379-86 (2003).
39. Kim-Hellmuth, S. *et al.* Cell type-specific genetic regulation of gene expression across human tissues. *Science* **369**(2020).
40. Wen, X., Luca, F. & Pique-Regi, R. Cross-population joint analysis of eQTLs: fine mapping and functional annotation. *PLoS Genet* **11**, e1005176 (2015).
41. Garrido-Martin, D., Borsari, B., Calvo, M., Reverter, F. & Guigo, R. Identification and analysis of splicing quantitative trait loci across multiple tissues in the human genome. *Nat Commun* **12**, 727 (2021).
42. Chang, C.C. *et al.* Second-generation PLINK: rising to the challenge of larger and richer datasets. *Gigascience* **4**, 7 (2015).
43. Schwartzenruber, J. *et al.* Molecular and functional variation in iPSC-derived sensory neurons. *Nat Genet* **50**, 54-61 (2018).
44. Huang, Q.Q., Ritchie, S.C., Brozynska, M. & Inouye, M. Power, false discovery rate and Winner's Curse in eQTL studies. *Nucleic Acids Res* **46**, e133 (2018).
45. Lippert, C., Horta, D., Casale, F.P. & Stegle, O. <https://github.com/limix/limix>.
46. Roselli, C. *et al.* Multi-ethnic genome-wide association study for atrial fibrillation. *Nat Genet* **50**, 1225-1233 (2018).
47. Hartiala, J.A. *et al.* Genome-wide analysis identifies novel susceptibility loci for myocardial infarction. *Eur Heart J* **42**, 919-933 (2021).
48. Eppinga, R.N. *et al.* Identification of genomic loci associated with resting heart rate and shared genetic predictors with all-cause mortality. *Nat Genet* **48**, 1557-1563 (2016).
49. Giri, A. *et al.* Trans-ethnic association study of blood pressure determinants in over 750,000 individuals. *Nat Genet* **51**, 51-62 (2019).
50. Sotoodehnia, N. *et al.* Common variants in 22 loci are associated with QRS duration and cardiac ventricular conduction. *Nat Genet* **42**, 1068-76 (2010).

REVIEWER COMMENTS

Reviewer #1 (Remarks to the Author):

The authors have done a lot of work to sufficiently address most of my comments and those of the other reviewers. I have only a few remaining comments after which I think the paper is suitable for publication:

1) The authors compare the number of eQTLs detected per tissue based on single-tissue eQTL analysis, the multi-tissue mash method, and their combined method using a mixed effects model, using barplots. This only compares the number of discovered eQTLs. It would be informative to know how many eGenes overlap between the methods. Can the authors add a venn diagram showing the overlap (number/fraction) of eGenes or eQTLs discovered between the different methods, in particular between the combined approach and mash.

2) The authors in this paper seemed to have focused on the most significant (lead) eQTL per eGene or eIsoform (results in Table S2). Do the authors make all significant variant-gene pairs found per eGene or eIsoform available, which is usually on the order of 10^5 or 10^6 variant-gene pairs per tissue based on GTEx. This would be useful for different types of downstream analyses by other users.

3) In the Methods section on page 22, significance of enrichment of GWAS causal variants in heart chromatin states in 210 fine-mapped GWAS loci was determined with a nominal enrichment P-value cutoff of < 0.05 . I think the authors should apply a multiple hypothesis correct p-value cutoff.

4) For the plots in Fig. 1C,D, Fig3C-H, Fig. 5B-D, Fig. 6G-Q, Figure S11, S12 it would be informative to colorcode the points based on LD to the lead or reference variant, as done in LocusZoom or LocusCompare plots, using a relevant reference panel.

5) In the write up in the Methods section "Validation of cell type-associated eQTLs (comparison with GTEx cell-type associated eQTLs)", the authors should specify if they computed a Pearson or Spearman's correlation coefficient. It would be good to also decrease the font of the colorbar in Figure S5, as the numbers overlap.

6) For colocalization analysis between GWAS and eQTL signals, the authors state that they only performed colocalization on the genome-wide significant GWAS variants, based on the Methods section "Colocalization between GWAS and eQTL signals" on page 21: "We performed colocalization between each significant eQTL signal and all genome-wide significant GWAS signals at each locus using the coloc.abf function in R". Can the authors comment on whether multiple variants per GWAS locus were considered in the colocalization analysis or only the lead GWAS variant? It could be more powerful to consider all the significant variant-gene pairs tested per eQTL that fall in the LD window around a lead GWAS variant, since the lead GWAS variant is not always that variant with the highest colocalization posterior probability, but it would be good if the authors could at least clarify how many variants were tested per locus.

7) To this paragraph the authors added to the Discussion following comment 8 from Reviewer #3, the authors can add 'trans-eQTLs' as another potential causal mechanism underlying GWAS associations (See in bold), as it has been shown to be enriched for heritability of complex traits to a larger extent than cis-eQTLs (though most studies are still under-powered to detect trans eQTLs): "Therefore, future studies on the functional characterization of genetic variants will likely require integration with other functional information, such as epigenomic data or trans-eQTLs to identify candidate causal variants".

8) The authors added the term "polygenic eQTLs" to the manuscript to denote eQTLs acting on more than one gene. I would recommend using a different term because "polygenic" is used in the

genetics/genomics field to denote contribution of multiple variants to a given phenotype or disease, but here we are referring to the effect on one variant on multiple genes. Perhaps the authors can use instead "multigenic eQTLs".

Reviewer #2 (Remarks to the Author):

The authors adequately addressed my concerns.

Reviewer #3 (Remarks to the Author):

I would like to thank the authors for clarifying most of the points. One think that struck me, is that the coloc analyses only seem to be performed on the global eQTL results and not on the spatiotemporally resolved eQTL. There is no reason, why not to perform the coloc and also enrichment analyses on results from the interaction model. In fact, this might actually lead to smaller credible sets and higher enrichments of functional annotations. I would strongly recommend to do run these analysis not only on the lead variants, but on the spatiotemporla eQTL results across all SNPs in each of the loci.

Specific comments in relation to responses:

Re comment 6:

It would really be interesting to show whether the spatio-temporal eQTL do reduce the size for the credible sets. This would really demonstrate that taking this information into account can increase the certainty in fine-mapping. The authors should run the analysis of credible sets based on the spatio temporal eQTL analysis applied to all SNPs in a locus (and also use these for coloc).

Re6 also affects results in previous comment 9. Why is coloc only done on the global eQTL results?

Re comment 1:

The new results show that the method used by the authors produces more tissue associated eQTL than the mash and traditional approach. In addition, the authors separately show enrichment of GWAS traits among mash results and their own results. It is difficult to compare these enrichment results. Which approach is showing a clearer signal for phenotypes?

Re comment 4:

The spatiotemporal analysis could in principle also be applied without first running the combined global eQTL analysis. This might reveal additional eQTL that have previously not been possible to detect because the context information is neglected. To get a full picture of these spatiotemporal eQTL, this would be a great addition. If no results in this direction are provided, at least this option should be discussed.

Related to previous comment 5:

p6, l140 'associated' should be explained briefly in the text

p7, 158-160: please make clear whether the cell type proportion that goes into the initial interaction analysis is a quantitative variable or discretized (into quintiles).

Re9 and 10: why do you use different statistical test to assess very similar situations for context dependency and overlap of anti-sense genes?

Re 17: I still find this number of PEER factors extremely high (as reviewer 1) - only looking at the discovery might be a bit misleading here. Could the authors try to establish what these factors actually capture? How much variance is there on these factors and do they correlate with any

measured covariates?

New minor comment:

The term polygenic eQTL seems not to be accurate: traditionally polygenic would describe those eGenes that have many independent eQTL. When the say locus affects multiple genes, it should be called pleiotropic instead.

Reviewer #1 (Remarks to the Author):

The authors have done a lot of work to sufficiently address most of my comments and those of the other reviewers. I have only a few remaining comments after which I think the paper is suitable for publication:

1) The authors compare the number of eQTLs detected per tissue based on single-tissue eQTL analysis, the multi-tissue mash method, and their combined method using a mixed effects model, using barplots. This only compares the number of discovered eQTLs. It would be informative to know how many eGenes overlap between the methods. Can the authors add a venn diagram showing the overlap (number/fraction) of eGenes or eQTLs discovered between the different methods, in particular between the combined approach and mash.

Our combined eQTL analysis jointly identifies eQTLs and eGenes across all tissues. We then perform a second step using an interaction term to identify eQTLs for eGenes and eIsoforms that function in a spatiotemporal manner. On the other hand, mash requires effect sizes and their standard errors in each tissue as input. To have a direct comparison between mash and our outputs for validation, we initially performed an eQTL analysis on each tissue separately using limix and the same covariates as our combined eQTL approach (see Table S9 and Methods section “Validation of tissue-associated eQTLs”). We then selected from our combined eQTL analysis the lead variant of each of the 18,030 eQTLs (11,692 primary and 6,338 conditional, Figure 1A), obtained their effect sizes and standard errors for each tissue from the single-tissue eQTL analyses, and used these values to perform mash with default parameters. We annotated each eQTL signal as active in a tissue if the local false sign rate (LFSR) < 0.05 (Table S10). Therefore, for each tissue, the eQTLs, and hence eGenes, identified by mash will be a subset of those that we identified in combined eQTL approach.

To address the Reviewer’s comment, we have added the requested Venn diagrams in Figure S13:

“B-F) Venn diagrams showing the overlap of the total 11,692 eGenes identified in the combined analysis with the eGenes identified as associated by mash (lfsr < 0.05) in each of the five tissues. To determine the overlap between these two sets, we intersected the 11,692 eGenes from the combined analysis with the eGenes identified as significant by mash (lfsr < 0.05) for the specific tissue. For example, for aorta (Panel B), mash identified 4,251 of the original 11,692 eGenes as being tissue-associated while the remaining 7,441 were not.”

2) The authors in this paper seemed to have focused on the most significant (lead) eQTL per eGene or eIsoform (results in Table S2). Do the authors make all significant variant-gene pairs found per eGene or eIsoform available, which is usually on the order of 10^5 or 10^6 variant-gene pairs per tissue based on GTEx. This would be useful for different types of downstream analyses by other users.

We agree that this was not clear in the main text. In the Methods at the end of the “Combined eQTL analysis” section we had stated, “Summary statistics for all eQTLs are reported in Figshare”. In the main text we have now added, “Summary Statistics reported in Figshare¹” in parenthesis.

3) In the Methods section on page 22, significance of enrichment of GWAS causal variants in heart chromatin states in 210 fine-mapped GWAS loci was determined with a nominal enrichment P-value cutoff of < 0.05 . I think the authors should apply a multiple hypothesis correct p-value cutoff.

We thank the reviewer for their careful review of our manuscript. We have: 1) modified the text in the methods to: **Enrichment p-values were FDR-corrected (Benjamini-Hochberg)**; 2) changed Figure S16 to show the FDR-corrected p-values (Benjamini-Hochberg); 3) changed the corresponding figure legend: **Filled circles represent FDR-corrected p-values (Benjamini-Hochberg). Variants with high PPA are more likely to reside in active chromatin states, such as enhancer (Enh) regions in fetal heart ($q = 2.29 \times 10^{-28}$) and transcriptional start sites (TssA) in adult heart (right atrium: $q = 3.34 \times 10^{-31}$; left ventricle: $q = 2.75 \times 10^{-38}$; and right ventricle: $q = 3.9 \times 10^{-45}$);** and 4) changed the text on page 10 in the Results section to reflect the updated p values.

4) For the plots in Fig. 1C,D, Fig3C-H, Fig. 5B-D, Fig. 6G-Q, Figure S11, S12 it would be informative to colorcode the points based on LD to the lead or reference variant, as done in LocusZoom or LocusCompare plots, using a relevant reference panel.

Thank you for the suggestion. We have modified the figures suggested by the Reviewer.

5) In the write up in the Methods section “Validation of cell type-associated eQTLs (comparison with GTEx cell-type associated eQTLs)”, the authors should specify if they computed a Pearson or Spearman’s correlation coefficient. It would be good to also decrease the font of the colorbar in Figure S5, as the numbers overlap.

We have updated the colorbar in Figure S5 and changed the following sentence in the Methods section: “**We downloaded myocyte interaction eQTLs for heart left ventricle from the GTEx Data Portal and calculated the Pearson correlation coefficient between effect sizes for matching eGenes using the *cor.test* function in R**”.

6) For colocalization analysis between GWAS and eQTL signals, the authors state that they only performed colocalization on the genome-wide significant GWAS variants, based on the Methods section “Colocalization between GWAS and eQTL signals” on page 21: “We performed colocalization between each significant eQTL signal and all genome-wide significant GWAS signals at each locus using the *coloc.abf* function in R”. Can the authors comment on whether multiple variants per GWAS locus were considered in the colocalization analysis or only the lead GWAS variant? It could be more powerful to consider all the significant variant-gene pairs tested per eQTL that fall in the LD window around a lead GWAS variant, since the lead GWAS variant is not always that variant with the highest colocalization posterior probability, but it would be good if the authors could at least clarify how many variants were tested per locus.

We apologize for the confusion: we tested each genome-wide significant GWAS signal against all eQTL signals within 500 kb. Colocalization compares the effect sizes and p-values of all the variants tested in both studies, therefore we compared

all the genotyped variants in both eQTL and GWAS for each locus that had at least one genome-significant variant for both eQTLs and GWAS.

We have modified and/or added the following sentences: “We found 1,444 eGenes and 919 eIsoforms overlapped or were in close proximity (<500 kb) with genome-wide significant GWAS variants. We performed colocalization between each significant eQTL signal and all genome-wide significant GWAS signals within the same locus (defined as GWAS signal and corresponding eGene or eIsoform <500 kb apart) using the *coloc.abf* function in R². Colocalization tests if two genetic association signals (such as eQTL and GWAS) have the same underlying causal variant by comparing the effect sizes of all the variants genotyped at the locus of interest. Colocalization was performed only on loci that included at least one genome-wide significant eVariant and one genome-wide significant GWAS variant”.

7) To this paragraph the authors added to the Discussion following comment 8 from Reviewer #3, the authors can add ‘trans-eQTLs’ as another potential causal mechanism underlying GWAS associations (See in bold), as it has been shown to be enriched for heritability of complex traits to a larger extent than cis-eQTLs (though most studies are still under-powered to detect trans eQTLs): “Therefore, future studies on the functional characterization of genetic variants will likely require integration with other functional information, such as epigenomic data or trans-eQTLs to identify candidate causal variants”.

We have updated this sentence as suggested by the Reviewer.

8) The authors added the term “polygenic eQTLs” to the manuscript to denote eQTLs acting on more than one gene. I would recommend using a different term because “polygenic” is used in the genetics/genomics field to denote contribution of multiple variants to a given phenotype or disease, but here we are referring to the effect on one variant on multiple genes. Perhaps the authors can use instead “multigenic eQTLs”.

We agree that “polygenic eQTLs may be misleading, and have changed it to “multigenic eQTLs”. We thank the reviewer for this suggestion.

Reviewer #2 (Remarks to the Author):

The authors adequately addressed my concerns.

Thank you.

Reviewer #3 (Remarks to the Author):

I would like to thank the authors for clarifying most of the points. One think that struck me, is that the coloc analyses only seem to be performed on the global eQTL results and not on the spatiotemporally resolved eQTL. There is no reason, why not to perform the coloc

and also enrichment analyses on results from the interaction model. In fact, this might actually lead to smaller credible sets and higher enrichments of functional annotations. I would strongly recommend to do run these analyses not only on the lead variants, but on the spatiotemporal eQTL results across all SNPs in each of the loci.

We apologize for the confusion: We performed colocalization between each significant eQTL signal and all genome-wide significant GWAS signals within the same locus (defined as GWAS signal and corresponding eGene or eIsoform <500 kb apart) using the *coloc.abf* function in R ². Colocalization tests if two genetic association signals (such as eQTL and GWAS) have the same underlying causal variant by comparing the effect sizes of all the variants genotyped at the locus of interest. Colocalization was performed only on loci that included at least one genome-wide significant eVariant and one genome-wide significant GWAS variant. Hence the spatiotemporally resolved eQTLs were included in the coloc analysis and in the Results we describe that, “Fifteen of the 79 credible sets with five or fewer SNPs were associated with spatiotemporal eQTL signals, suggesting that the association between these genetic variants and cardiac traits likely occurs at specific developmental stages, tissues or cell types”.

To improve the clarity of this analysis, we have modified and/or added the following sentences to the Methods on page 21: “We found 1,444 eGenes and 919 eIsoforms overlapped or were in close proximity (<500 kb) with genome-wide significant GWAS variants. We performed colocalization between each significant eQTL signal and all genome-wide significant GWAS signals within the same locus (defined as GWAS signal and corresponding eGene or eIsoform <500 kb apart) using the *coloc.abf* function in R ². Colocalization tests if two genetic association signals (such as eQTL and GWAS) have the same underlying causal variant by comparing the effect sizes of all the variants genotyped at the locus of interest. Colocalization was performed only on loci that included at least one genome-wide significant eVariant and one genome-wide significant GWAS variant.”.

Specific comments in relation to responses:

Re comment 6: It would really be interesting to show whether the spatiotemporal eQTL do reduce the size for the credible sets. This would really demonstrate that taking this information into account can increase the certainty in fine-mapping. The authors should run the analysis of credible sets based on the spatiotemporal eQTL analysis applied to all SNPs in a locus (and also use these for coloc).

We apologize for the confusion: performing spatiotemporal eQTL analysis does not change the effect sizes of the eVariants but only describes if the eQTL signal is associated with developmental stage, tissue, organ or cell type. Hence, the spatiotemporal eQTLs are not expected to improve fine-mapping. As described in response to the reviewers’ comment above, Colocalization compares the effect sizes and p-values of all the variants tested in both studies, therefore we compared all the genotyped variants in both eQTL and GWAS for each locus that had at least one genome-significant variant for both eQTLs and GWAS.

We tested the spatiotemporal context of eQTLs by adding the genotype by context interaction only to the lead variant of each eQTL signal. Since each eQTL signal underlies one single causal variant, the spatiotemporal context of the lead variant is sufficient to annotate but not modify the whole eQTL signal.

To make this clearer, we added the following to the “Spatiotemporal eQTL mapping” section in the Methods: “Since each eQTL signal underlies one single causal variant, the spatiotemporal context of the lead variant is sufficient to annotate the whole eQTL signal”.

Re6 also affects results in previous comment 9. Why is coloc only done on the global eQTL results?

Since colocalization is performed using all genotyped variants in both eQTL and GWAS studies, we performed it using the global eQTL results (see response to first comment by R3 above). To test the spatiotemporal context of each eQTL, we tested the genotype by context interaction only on the lead variant.

Re comment 1: The new results show that the method used by the authors produces more tissue associated eQTL than the mash and traditional approach. In addition, the authors separately show enrichment of GWAS traits among mash results and their own results. It is difficult to compare these enrichment results. Which approach is showing a clearer signal for phenotypes?

Our method and MASH provide different information: we focused on identifying eQTL signals that are tissue-specific and, to do so, we developed a very stringent approach, which likely results in a large number of false negatives (i.e. tissue-specific eQTLs that we do not classify as such); on the other hand, MASH uses information from other tissues to improve power to detect eQTLs in a specific tissue. Therefore, MASH identifies eQTLs that are “active” in a tissue because they occur in multiple other tissues, rather than describing tissue-specific eQTLs.

In general, at local false sign rate (LFSR) < 0.05 , MASH results in more associations with cardiac traits than our method, because it relies on thousands of eQTL signals, rather than a few hundred. At more stringent LFSR, enrichments are more consistent. For example, with our tissue-specific eQTL method, we found that pulse rate is associated with atrium, adult heart, adult eQTLs and, to a lower extent, arteria, whereas MASH at LFSR < 0.05 identified associations with all adult tissues. At the most stringent threshold (LFSR $< 10^{-20}$) MASH identified associations with atrium, ventricle and aorta.

To make this clearer, we have updated the Methods section “Validation of the enrichment of tissue-associated eQTLs for GWAS cardiac traits”: “While mash is useful to validate the tissue-associations that we identified, our method and mash provide different information: we focused on identifying eQTL signals that are tissue-specific and, to do so, we developed a very stringent approach, which likely results in a large number of false negatives (i.e. tissue-specific eQTLs that we do not classify as such); on the other hand, mash uses information from other tissues to improve power to detect eQTLs in a specific tissue. Therefore, mash is able to identify eQTLs that are “active” in a tissue because they occur in multiple other tissues, rather than describing tissue-specific eQTLs”.

We have also added the following to the legend of Figure S14: “In general, at local false sign rate (LFSR) < 0.05 , MASH results in more associations with cardiac traits than our method, because it relies on thousands of eQTL signals, rather than

a few hundred. At more stringent LFSR, enrichments are more consistent. For example, with our tissue-specific eQTL method, we found that pulse rate is associated with atrium, adult heart, adult eQTLs and, to a lower extent, arteria, whereas MASH at $LFSR < 0.05$ identified associations with all adult tissues. At the most stringent threshold ($LFSR < 10^{-20}$) MASH identified associations with atrium, ventricle and aorta”.

Re comment 4: The spatiotemporal analysis could in principle also be applied without first running the combined global eQTL analysis. This might reveal additional eQTL that have previously not been possible to detect because the context information is neglected. To get a full picture of these spatiotemporal eQTL, this would be a great addition. If no results in this direction are provided, at least this option should be discussed.

We thank the Reviewer for this comment. By performing conditional eQTL analysis results in the detection of independent signals for each gene or isoform, the first conditional eQTL is detected by regressing out the genotype of the lead variant for the primary eQTL, the second conditional eQTL is detected by regressing out the genotype of the lead variants for the primary and first conditional eQTLs, and so on. This procedure is performed to remove all the eQTL signals for each variant in LD with each lead variant. By adding the genotype by context interaction, this would not be changed, therefore it is extremely unlikely that new signals would be detected. So, while an interesting idea, we feel that the suggested analysis is out of the scope of this study.

Related to previous comment 5:

p6, 1140 'associated' should be explained briefly in the text p7, 158-160: please make clear whether the cell type proportion that goes into the initial interaction analysis is a quantitative variable or discretized (into quintiles).

We thank the reviewer for carefully reading our manuscript and apologize that this was not clear. The interaction that the Reviewer is referring to (the linear mixed model described in the Methods section “Spatiotemporal eQTL mapping”) uses the cell type proportion values. However, we perform a second step in the interaction analysis in order determine the association between the tested eQTL and the top and bottom quartiles. Four scenarios may occur:

1. eQTL is significant only for the tested cell type (i.e. the top quartile): the eQTL signal is considered tissue-specific;
2. eQTL is significant only for the bottom quartile: the eQTL is not considered as cell type-specific;
3. eQTL is significant for both the top and bottom quartiles but its effect size is stronger (larger absolute value) in the top quartile: the eQTL is cell type-associated;
4. eQTL is significant for both the top and bottom quartiles but its effect size is stronger (larger absolute value) in the bottom quartile: the eQTL is not considered as cell type-specific or cell type-associated.

We updated the Methods section “Spatiotemporal eQTL mapping”: “When analyzing cell types, the following four scenarios may occur:

1. eQTL is significant only for the tested cell type (i.e. the top quartile): the eQTL signal is considered tissue-specific;
2. eQTL is significant only for the bottom quartile: the eQTL is not considered as cell type-specific;
3. eQTL is significant for both the top and bottom quartiles but its effect size is stronger (larger absolute value) in the top quartile: the eQTL is cell type-associated;
4. eQTL is significant for both sets the top and bottom quartiles but its effect size is stronger (larger absolute value) in the bottom quartile: the eQTL is not considered as cell type-specific or cell type-associated”.

Re9 and 10: why do you use different statistical test to assess very similar situations for context dependency and overlap of anti-sense genes?

To test for context dependency, we had to use a permutation test because some eGenes and/or eIsoforms colocalized with multiple other eGenes or eIsoforms, while some eGenes and/or eIsoforms colocalized only with one other eGene or eIsoform. A permutation test was required in order to ensure that one or few context-specific eGenes did not drive the entire enrichment signal. On the other hand, in the gene/antisense analysis all tests were paired, therefore the Fisher’s Exact Test was appropriate.

We added the following text to the Methods section “Enrichment of spatiotemporally regulated eQTLs in multigenic eQTLs”: “To confirm that eQTL signals associated with multiple eGenes were more likely than expected to be associated with the same spatiotemporal context and to ensure that enrichments were not due to a small number of signals shared by many eGenes (certain eGenes may have a stronger weight on the overall analysis than other), we also performed a permutation analysis”.

Re 17: I still find this number of PEER factors extremely high (as reviewer 1) - only looking at the discovery might be a bit misleading here. Could the authors try to establish what these factors actually capture? How much variance is there on these factors and do they correlate with any measured covariates?

We understand that the number of PEER factors may seem high, but we have utilized the approach recently published by Oliver Stegle’s lab ^{3,4}, who developed PEER factors in the first place. Figure S18C shows how the top 10 PEER factors correlate with all the other covariates used for the eQTL analysis. As expected, the top PEER factors are strongly correlated with developmental stage.

New minor comment: The term polygenic eQTL seems not to be accurate: traditionally polygenic would describe those eGenes that have many independent eQTL. When the say locus affects multiple genes, it should be called pleiotropic instead.

We agree with the Reviewer that the term “polygenic eQTLs” may not be accurate. We followed the suggestion by Reviewer 1 (R1C8) and changed it to “multigenic eQTLs”.

References

1. D'Antonio, M. Fine mapping spatiotemporal mechanisms of genetic variants underlying cardiac traits and disease. *figshare*, <https://doi.org/10.6084/m9.figshare.c.5594121> (2021).
2. Giambartolomei, C. *et al.* Bayesian test for colocalisation between pairs of genetic association studies using summary statistics. *PLoS Genet* **10**, e1004383 (2014).
3. Bonder, M.J. *et al.* Identification of rare and common regulatory variants in pluripotent cells using population-scale transcriptomics. *Nat Genet* **53**, 313-321 (2021).
4. Stegle, O., Parts, L., Piipari, M., Winn, J. & Durbin, R. Using probabilistic estimation of expression residuals (PEER) to obtain increased power and interpretability of gene expression analyses. *Nat Protoc* **7**, 500-7 (2012).